# A within-host infection model to explore tolerance and resistance

**David Duneau[1,2]*†, Pierre DM Lafont[1,3]†, Christine Lauzeral[1], Nathalie Parthuisot[1], Christian Faucher[1], Xuerong Jin[4], Nicolas Buchon[4], Jean-Baptiste Ferdy[1]***

[1]Centre de Recherche sur la Biodiversit´e et l'Environnement, Universit´e Paul Sabatier, Toulouse, France; [2]Centre for Cardiovascular Sciences, Queen's Medical Research Institute, University of Edinburgh, Edinburgh, United Kingdom; [3]Institute of Evolutionary Biology, School of Biological Sciences, University of Edinburgh, Edinburgh, United Kingdom; [4]Cornell Institute of Host-Microbe Interactions and Disease, Department of entomology, Cornell university, New York, United States

**Abstract** How are some individuals surviving infections while others die? The answer lies in how infected individuals invest into controlling pathogen proliferation and mitigating damage, two strategies respectively called resistance and disease tolerance. Pathogen within-host dynamics (WHD), influenced by resistance, and its connection to host survival, determined by tolerance, decide the infection outcome. To grasp these intricate effects of resistance and tolerance, we used a deterministic theoretical model where pathogens interact with the immune system of a host. The model describes the positive and negative regulation of the immune response, consider the way damage accumulate during the infection and predicts WHD. When chronic, infections stabilize at a Set-Point Pathogen Load (SPPL). Our model predicts that this situation can be transient, the SPPL being then a predictor of life span which depends on initial condition (e.g. inoculum). When stable, the SPPL is rather diagnostic of non-lethal chronic infections. In lethal infections, hosts die at a Pathogen Load Upon Death (PLUD) which is almost independent from the initial conditions. As the SPPL, the PLUD is affected by both resistance and tolerance but we demonstrate that it can be used in conjunction with mortality measurement to distinguish the effect of disease tolerance from that of resistance. We validate empirically this new approach, using *Drosophila melanogaster* and the pathogen *Providencia rettgeri*. We found that, as predicted by the model, hosts that were wounded or deficient of key antimicrobial peptides had a higher PLUD, while Catalase mutant hosts, likely to have a default in disease tolerance, had a lower PLUD.

***For correspondence:**
david.duneau@gmail.com (DD);
jean-baptiste.ferdy@univ-tlse3.fr
(J-BaptisteF)

†These authors contributed equally to this work

**Competing interest:** The authors declare that no competing interests exist.

## Editor's evaluation

Duneau et al. provide an extensive effort to model parameters of infection, an important topic in disease management. The theoretical findings of this study are important and will be of interest to mathematical biologists to model infection. The empirical data support the arguments, but may be incomplete, and more could be done in experiment design to shore up the robustness of these findings. This study helps us to better understand the complex course of infection.

## Introduction

Infectious diseases produce a vast array of symptoms, some leading to death whereas others are easily overcome. This heterogeneity of outcomes reflects, in part, the diversity of pathogens. However, even typically benign pathogens can occasionally be fatal, while the deadliest pathogens rarely cause 100% mortality (e.g. even in Ebola infections, 40% of infected people survive, *Shultz et al., 2016*).

Predicting who is at greater risk of dying from an infection is central in medicine, because accurate predictions can help tailoring public health efforts. A common approach to this problem relies on detecting statistical associations between patients' conditions and fatality rates (e.g. *Barda et al., 2020* in the case of COVID-19). Although essential and efficient, this approach is not meant to understand the ultimate causes of death.

What actually makes an infected patient succumb is that it did not control pathogen proliferation and could not sustain the damage imposed by the infection (*Colaço et al., 2021*; *Duneau and Ferdy, 2022*; *Jackson et al., 2014*; *Soares et al., 2017*). Overall, the hosts can, therefore, survive an infection by strategies which combine resisting to pathogens by producing immune defense and tolerating the consequences of infection (in animals *Ayres and Schneider, 2008*; *Råberg et al., 2007*; *Read et al., 2008*, but also in plants *Kover and Schaal, 2002*). Hosts which survive because they tolerate infections can either sustain a high level of damage, (i.e. disease tolerance stricto sensu *Schneider, 2021*) or repair them efficiently to maintain homeostasis, (i.e. resilience sensu *Ferrandon, 2013*). Forecasting the outcome of an infection for one individual patient requires that we quantify the relative investment in each of these strategies (*Balard and Heitlinger, 2022*).

In principle, separating the effects of disease resistance from those of disease tolerance should be relatively easy, as resistance directly impacts the Within Host Dynamics (WHD) of pathogens, while tolerance affects survival but should have no effect on WHD. The task is in fact challenging because both resistance and tolerance have indirect effects which intermingle. For example, immune defense is a double-edged sword which can save the host's life but sometimes cause pathologies and reduce lifespan (*Critchlow et al., 2023*; *Lin et al., 2018*; *Petkau et al., 2017*). An increase in resistance could, therefore, come with an apparent decrease in disease tolerance. Reciprocally, disease tolerance and the damage mitigation it relies on could allow a host to sustain higher levels of defense, and indirectly increase its resistance to disease. These intricate effects, added to the fact that some regulatory genes affect both resistance and tolerance, led some authors to consider that they should be seen as two finely co-regulated aspects of the immune response to infection (*Martins et al., 2019*; *Pucillo and Vitale, 2020*). Studying experimentally the WHD and its connection to survival is probably still the only way to understand the mechanisms which underlie resistance and tolerance, but theoretical studies are needed to unravel their effects and provide testable predictions (*Lazzaro and Tate, 2022*).

We present here an effort to disentangle the action of disease tolerance and resistance using a mathematical model of WHD. Our model aims at being general, while taking advantage of recent empirical descriptions of WHD in *Drosophila melanogaster* (*Duneau et al., 2017a*). Invertebrates have proven to be good alternative experimental models to understand the relation between WHD and

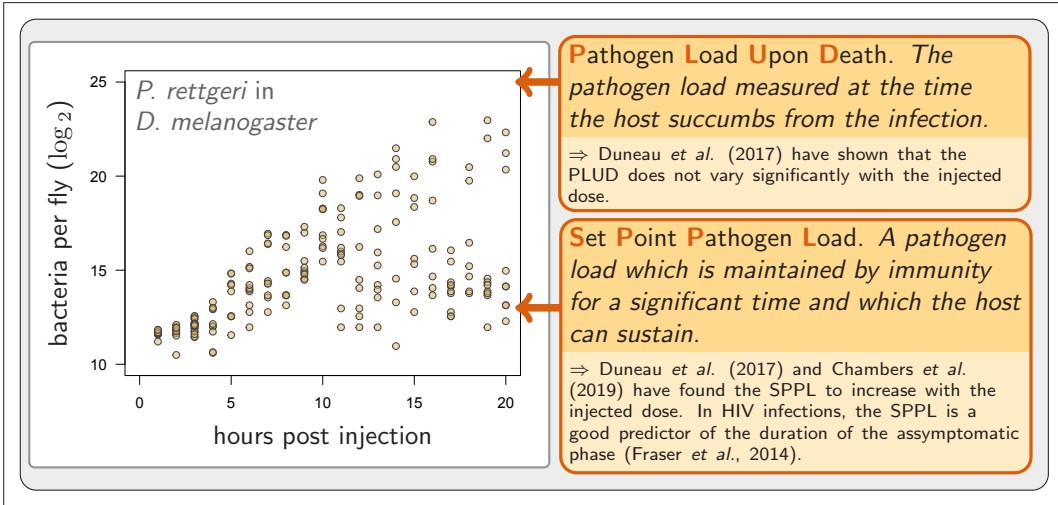

**Figure 1.** Example of experimental within-host bacterial dynamic using *Drosophila*. Adapted from *Duneau et al., 2017a*. Each point represents the bacterial load estimated from a single killed male fly (*Drosophila melanogaster*) injected with a suspension of the bacterium *Providencia rettgeri* containing *ca.* 2000 bacterial cells. Twenty hours after injection, some flies maintain a moderate load and will live on for several days, while others have reached high loads and will die rapidly.

infection outcomes. First, they provide the advantage that the total pathogen load can be quantified on large numbers of animals, regardless of the pathogen tropism, so that the WHD can be studied on a wide range of disease types (*Duneau et al., 2017a*). Second, they share with mammals a large part of the mechanisms underlying their innate immune system, as shown in *Drosophila melanogaster* (*Buchon et al., 2014*). Our purpose was twofold: first, we aimed to designing a model which could reproduce the different situations documented by *Duneau et al., 2017a* and in other studies; second we used it to explore how experimental proxies of resistance and disease tolerance actually connect to infection parameters. *Figure 1* presents an example of WHD in *Drosophila* which illustrates one of the behaviors we wanted our model to be able to reproduce, and defines the PLUD and the SPPL, which we propose as a generalization of SPVL and SPBL to pathogens other than viruses and bacteria, the two quantities which we have investigated most. Finally, we used our model to design methods which could help to experimentally distinguish the effects of disease tolerance from that of resistance.

We found that only models with complex, non-linear regulations of immunity can satisfactorily reproduce infections which range from chronic to acute and from benign to lethal. We further demonstrated that the model adequately predicts both the existence and the properties of the SPPL if defense production is assumed to decrease with accumulating damage. We show that the SPPL can then be transient and does, as documented in HIV infections, predict the host's lifespan. We further demonstrated that SPPLs with entirely different properties can be measured in other situations where the pathogen stably associates to the host, causing little damage. Our model also predicts that the host dies at a PLUD which is virtually independent of inoculum size (as observed by *Duneau et al., 2017a*), and should therefore reflect the genetic characteristics of interacting pathogens and hosts. We finally propose that the PLUD could be used in combination with a mortality measurement (such as the Hazard Ratio) to experimentally distinguish the effects of disease tolerance from that of resistance. We validated this theory using experimental infections of *Drosophila melanogaster* by the pathogenic bacterium *Providencia rettgeri*.

## Methods and results
### A model of host-pathogen interactions

One purpose of our work was to develop a mathematical model capable of reproducing the wide variety of WHD documented to date. Our model aims at being general, but we used the recent empirical work of *Duneau et al., 2017a* on *Drosophila melanogaster* to facilitate explanations and interpretations. This previous work documents chronic and acute diseases, some benign, while others are lethal. But most importantly, it also depicts situations where an infection initiated in carefully controlled conditions can bifurcate, with some of the hosts developing a form of chronic infection when others die within a few days (as illustrated in *Figure 1*).

Such bifurcations in dynamical systems are most often by-products of bistability. The few available examples of bistable WHD models in the literature are all modifications of the Lotka-Volterra predator-prey model (*Pujol et al., 2009*; *Souto-Maior et al., 2018*). *Mayer et al., 1995* demonstrated that in such models non-linear functional responses for both pathogen dynamics and host defense regulation could yield bifurcations between chronic infections with high and low equilibrium pathogen load.

Other models have proven that the non-linearities that create bifurcations could originate from feedback between pathogen proliferation and immune defense production. *van Leeuwen et al., 2019*, for example, demonstrated that a model that explicitly describes the amount of resource intestinal pathogens diverted from their host can be bistable, a result later confirmed by *Yu et al., 2021*. *Ellner et al., 2021* showed that situations where pathogens actively destroy immune defenses and also create bistability in disease outcome. All these models have in common that they predict bistability where some infections are cleared (or almost cleared, in the case of *Ellner et al., 2021* conceptual model). In that sense, they do not reproduce the situations documented by *Duneau et al., 2017a*, where some infections remain chronic while others are lethal.

In our research, we amalgamated *Mayer et al., 1995*'s concept of nonlinear modulation in defense production with the hypothesis that pathogens cause damage that can hinder the host's ability to combat infection. We customized our model to not only forecast pathogen dynamics, but also estimate the host's survival prospects.

Let $x$ be the pathogen load (e.g. bacteria), $y$ the level of defense (e.g. anti-microbial peptides) produced by the host immune system, and $z$ the damage caused by the infection (see *Figure 2A*). Whether the host suffers from severe infection, chronic yet mild infection, or clears pathogens, depends on its genetically determined capacity to fight or tolerate pathogens, but also on the inoculum size and on the host physiological state at the start of the infection. In our model, the inoculum size is the initial value of the pathogen load ($x_0$). The physiological state of the host at the start of the infection is described by the initial level of defense ($y_0$) and of damage ($z_0$). The initial level of defense can be set by constitutive immunity, when a healthy host is at its homeostatic state (i.e. $y_0 = \tilde{y}_h$). It can also be lowered, when the host is weakened by environmental challenges. Similarly, before the infection has started, the host suffers from an initial level of damage which origin is not the studied infection. This level is $\tilde{z}_h$ when the host is at its homeostatic state, but can be increased in case of environmental challenges (e.g. infection starting from a wound).

A set of parameters (described in *Figure 2A*) characterizes the genetic interaction between a host and a pathogen. These parameters reflect both the genetically determined capacity of the host to fight an infection and the genetically determined aggressiveness of the pathogen it is fighting against. These parameters govern the dynamics of $(x, y, z)$ over time $t$, as described by the following differential equations:

$$
\begin{cases}
\dfrac{dx}{dt} = & x\left(1 - x\right) - \delta xy \\[2mm]
\dfrac{dy}{dt} = & F(x)G(y, z) - \varphi y \\[2mm]
\dfrac{dz}{dt} = & \omega x + \eta y - \xi z \,.
\end{cases}
\tag{1}
$$

For the sake of simplicity, the equations of system (*Equation 1*) are written dimensionless, $x$ being a fraction of the pathogen's carrying capacity within the host and time units corresponding to the pathogen's generation time. The equation governing the dynamics of pathogens is that of the evolution of a prey population in a classic Lotka-Volterra model (*Lotka, 1923*; *Volterra, 1928*). The pathogens are destroyed by the defense at a rate $\delta y$, with $\delta$ representing the efficacy of immune molecules or cells at killing pathogens.

We considered that damage should increase with pathogen load, but that it might also increase with the level of defense that is produced (i.e. immunopathology, *González-González and Wayne, 2020*; *Graham et al., 2005*; *Rommelaere et al., 2024*). We assumed that damage are caused by pathogens at a rate $\omega$ and by defense at a rate $\eta$. We further reasoned that damage are repaired at a constant rate $\xi$. It is now well established that repairing mechanisms are in fact modulated during the course of an infection (*Martins et al., 2019*; *Pucillo and Vitale, 2020*). For simplicity, and because the details of how this regulation responds to damage and defense production are still unknown, we deliberately neglected this modulation. We also made the simplifying assumption that damage caused by pathogens and by immune responses is of the same nature, allowing it to be repaired at the same rate. Finally, we postulated that the host cannot sustain infinite damage, resulting in death if $z$ exceeds the threshold value $z_d$.

We assumed that the time variation in the level of defense is set by the balance between the production of new molecules or cells ($F(x)G(y, z)$) and the spontaneous decay of already produced ones ($\varphi y$, with defense persisting longer in the host when $\varphi$ is decreased). The functions $F$ and $G$ describe the modulation of defense production, with $F$ modeling how production is activated upon pathogen detection, and $G$ describing negative regulation. Following *Mayer et al., 1995*, we wrote both $F$ and $G$ as non-linear functional responses and investigated how the shapes of these two functions influence the outcome of the infection.

The function $F$ is given by

$$
F(x) = \gamma + \alpha \frac{x^u}{1 + \beta x^v} \,.
\tag{2}
$$

We posited that $F$ must be an increasing function of pathogen load ($u \geq v$ and $\beta > 0$). In the absence of pathogens and with no negative regulation, the production of defense equals $\gamma$. The parameter $\gamma$, therefore, sets the maximum constitutive production of defense. Upon infection, $F$ increases with

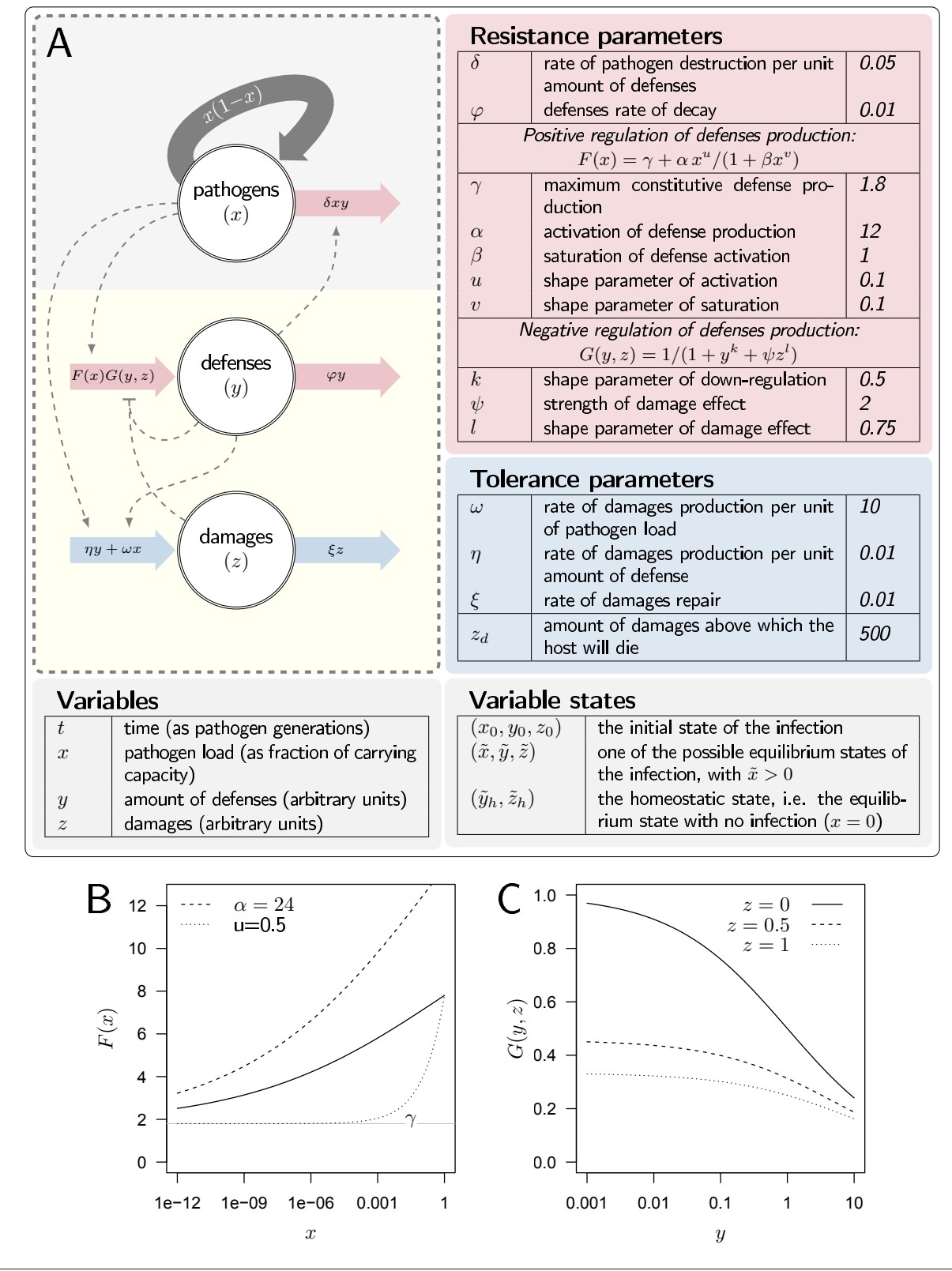

**Figure 2.** A model of Within Host Dynamics (WHD). (**A**) A description of the model. Large arrows indicate fluxes which make pathogen load ($x$), level of defense ($y$), and damage ($z$) vary over time. Dashed arrows indicate how each variable influences, negatively (flat arrowhead) or positively, these fluxes. Tables list parameters and their default values, separating those which determine the production or efficacy of defense (resistance parameters) from those which determine the production and the repair of damage (tolerance parameters). (**B**). The activation of defense as described by the functional

*Figure 2 continued on next page*

*Figure 2 continued*

response $F(x)$ gives the level of defense production that is reached in the absence of negative regulation. It increases with pathogen load $x$, $F(0) = \gamma$ (the horizontal gray line) being the maximum constitutive defense production, reached when the host is not infected. $\alpha$ controls how fast defense expression increases with load (dashed curve: $\alpha = 24$ instead of 12 for the plain curve). Increasing $u$, (dotted curve: $u = 0.75$ instead of 0.5) makes $F$ increase slower when the pathogen load is low. (**C**) The negative regulation of defense production as described by the functional response $G(y, z)$. Down-regulation of defense production increases with both defense ($y$) and damage ($z$) levels. Note, however, that $\psi = 0$ or $l = 0$ would make $G$ independent of damage level.

---

pathogen load (because we assume $\alpha > 0$), and reaches its maximum feasible value when $x$ reaches the carrying capacity (i.e. when $x = 1$ with then $F(1) = \gamma + \alpha/(1 + \beta)$). **Figure 2B** illustrates that higher values of α, i.e., more sensitive defense activation, make this increase steeper. Increasing $u$ above $v$ makes $F$ sigmoid, which reproduces cases where the production of defense is barely activated unless the pathogen load becomes significant (as **Figure 2B** illustrates). Overall, the parametrization of $F$ allowed us to vary the constitutive level of defense expression, the speed at which defenses are activated upon infection, and the load at which pathogens are detected.

The function $G$ corresponds to the negative regulation of the production of defense, with

$$G(y, z) = \frac{1}{1 + y^k + \psi z^l}. \tag{3}$$

Here, we considered that the regulation of the production of defense must incorporate a negative effect of immune defense ($y$) on defense production itself. This was motivated by the negative feedback loops that regulate immune responses in mammals and insects. In *Drosophila*, for example, the *Diedel* and *WntD* genes have such a function (**Lamiable et al., 2016a**; **Lamiable et al., 2016b**). Similarly, *PGRP-LB*, which has mammalian orthologs, is activated by the immune deficiency (Imd) pathway, one of the signaling cascades which regulates the expression of most antimicrobial peptides. But *PGRP-LB* also down-regulates this same Imd pathway, and thus provides a clear example of negative regulation of immune response (**Kleino and Silverman, 2014**).

Our model also aims to incorporate how damage accumulation modifies the regulation of defense production. This can conceptually be separated into two distinct phases. Early in the infection, damage-associated molecular patterns (DAMPs) have been proposed as universally used signals that trigger the activation of an immune response (**Matzinger, 1994**; **Seong et al., 2021**). These DAMPs can come from a wound, that would be systematically associated to the infection. We have assumed that in this case, DAMPs would trigger an instantaneous increase in both defense and damage, which we can model by increasing both $y_0$ and $z_0$. DAMPs could also originate from the infection itself, but they should then strongly correlate with bacterial load, as damage and load are tightly coupled during early infection. Therefore, we assumed that the positive regulation of defense production upon the detection of DAMPs that originate from the infection is incorporated in function $F$.

Later in the infection, the accumulation of damage caused by the infection could disrupt immune system homeostasis by altering the host's health condition (**Stoecklein et al., 2012**). This may occur because the energy spent repairing damage is no longer available for defense production. Additionally, the tissues producing defenses may themselves be damaged by the infection, impairing the production of immune molecules or cells. In our model, these effects are represented in function $G$ by the negative effect of the damage ($z$) on defense production (see **Figure 2C**), the importance of which is controlled by the parameters $\psi$ and $l$.

As should be clear from **Equations 2 and 3**, we purposely made $F$ and $G$ versatile so that our model can reproduce various types of host-pathogen interactions. This came at the cost of a large number of parameters. **Figure 2** represents the model, lists its parameters and gives the parameter values we have used in our computations, unless otherwise specified. Parameters can be separated in two lists, with disease tolerance parameters that control damage production or repair on one side, and resistance parameters, which control the efficacy and level of immune defense on the other (see **Figure 2**). This clear-cut distinction, though, ignores potential indirect effects. For example, increasing the host capacity to repair damage (i.e. increasing $\xi$) makes it more tolerant to the infection. But damage hinders defense production, so that increasing $\xi$ indirectly allows the host to produce more defense, hence making it more resistant. Similarly, increased defense production causes additional damage and should, therefore, reduce the host's tolerance to disease. One of our aims was, therefore,

to describe how the combined direct and indirect effects of each parameter impact both resistance and tolerance to infections.

Finally, due to the complexity of functions $F$ and $G$, the equilibria of our model are intractable. We, therefore, relied on numerical methods to compute both the homeostatic state in the absence of pathogens $(\tilde{y}_h, \tilde{z}_h)$ and other equilibria $(\tilde{x}, \tilde{y}, \tilde{z})$ with pathogens present. This, as most of the numerical analysis detailed here, has been performed using the GSL library (*Galassi et al., 2009*) and functions provided in (*R Developmental Core Team, 2018*). We were nevertheless able to derive the stability conditions for some of the equilibria (see Appendix 1) and the relation between equilibrium load and parameters (see Appendix 2).

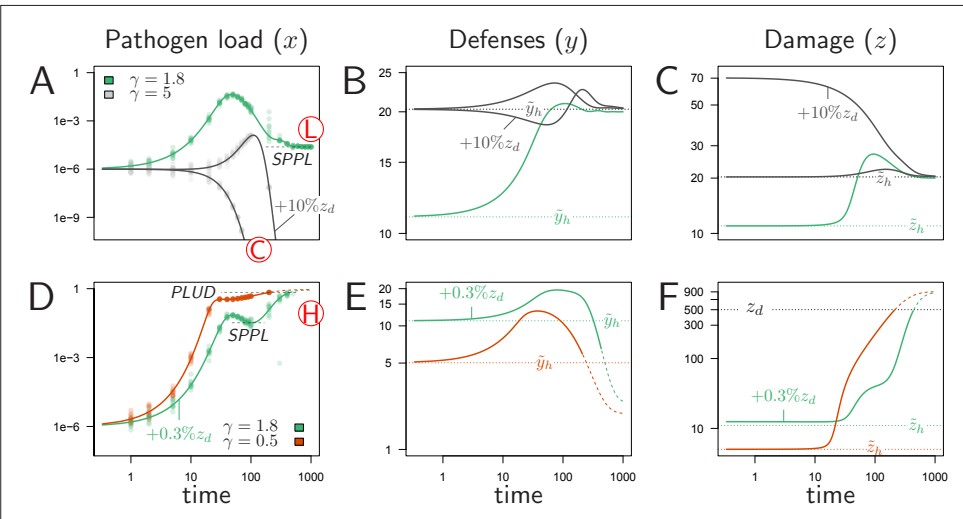

**Figure 3.** Illustrative cases of infection dynamics. Each column corresponds to one of the three variables of the model ($x$, pathogen load; $y$, defense; $z$, damage). In all cases, the initial pathogen load ($x_0$) is $10^{-6}$ and, unless otherwise specified, $y_0$ and $z_0$ are set at their homeostatic state $\tilde{y}_h$ and $\tilde{z}_h$ (indicated by horizontal dashed line). Red letters in circles indicate the outcome of the infection: C, clearance; L, low pathogen load; H, high pathogen load. Dots correspond to results of stochastic simulations where the initial pathogen load is randomly drawn from a log-normal distribution with an average in $\log(x_0)$ set to $\log(10^{-6})$ and variance 0.15, and then the deterministic equations of the model are solved numerically. (**A-C**) illustrate cases where the immune defense eventually controls the infection, either by clearing pathogens (gray curves, $\gamma = 5$) or by making it chronic (green curves, $\gamma = 1.8$). With $\gamma = 5$, the constitutive expression of defense is strong and hence the level of defense before infection ($\tilde{y}_h$) is very high. This level slightly increases upon infection (which leads to a corresponding increase in damage, as defense induce damage when $\eta > 0$) and returns to the homeostatic state once pathogens have been cleared. With $\gamma = 1.8$ (green curve), the initial level of defense is much lower and the infection stabilizes at the Set Point Pathogen Load (SPPL, see *Figure 1*). Defense and damage both peak following infection and, because pathogen are not cleared, they eventually stabilize at a level above the homeostatic state. (**D-F**) illustrates cases where the immune response does not suffice to control the infection (green curves, $\gamma = 1.8$, red curves, $\gamma = 0.5$). In both cases, the peak in defense production induced by infection is not sufficient to stop pathogen proliferation. This, combined to the side effects of defense, provokes a sharp increase in damage. The accumulation of damage hinders defense production (because $\psi > 0$ and $l > 0$) which in turn makes the level of defense rapidly decrease. The infection is then out of control and eventually kills the host, when the accumulated damage $z$ exceeds the level the host can sustain ($z > z_d$). Note that the two hosts die at very similar Pathogen Load Upon Death (PLUD, see *Figure 1*) although at very different times. Note also that bacteria continue proliferating after host death, so that the PLUD is below the carrying capacity at the time of death (which is fixed to one) and likely much below the load that could be reached in a cadaver. When $\gamma = 1.8$ (green curve), the immune response is strong enough to curb pathogen proliferation. This maintains the population of pathogens at a SPPL, which differs from that of the green curve in figures **A-C** because it is transient. These two green curves have been obtained with identical parameters; they only differ in the initial level of damage $z_0$ (which, in D-F, is increased by 0.3% of $z_d$). Their differences, therefore, demonstrate that the model can be bistable: with fixed parameters, the outcome of an infection might depend on initial conditions. These simulations can be reproduced using a dedicated R Shiny web application (*Chang et al., 2021*), where parameter values can be changed to explore their impact on the dynamics (see https://plafont.shinyapps.io/WHD_app/, DOI: 10.5281/zenodo.13309653).

## The different possible outcomes of an infection

The entanglement between pathogen proliferation, defense, and damage production is too complex to predict the outcome of an infection from their separate effects. But numerically integrating *Equations 1* allowed us to simulate infections, and thus to investigate how variations in parameters (e.g. the maximum level of constitutive production of defense, $\gamma$) or initial variable values (e.g. the initial level of damage, $z_0$) impact WHD and eventually determine the survival of the host.

By presenting a few chosen simulations, we first demonstrate that our model can reproduce situations where the host succeeds in controlling the infection (*Figure 3A–C*), either by clearing pathogens ($\gamma = 5$) or by maintaining pathogen load, and therefore damage, at a tolerable level ($\gamma = 1.8$). In the latter case, we can consider that the population of pathogens eventually reaches a Set Point Pathogen Load (SPPL, as defined in *Figure 1*) and will stay there for the whole host's life.

In the three simulations of *Figure 3A–C*, the level of defense was initially set at its homeostatic state, the immune state of a healthy host which is characterized by $y = \tilde{y}_h$ and $z = \tilde{z}_h$. Upon infection, defense increased and either returned to its initial level, when the infection is cleared, or stabilized to an intermediate level due to the remaining controlled pathogen population (*Figure 3B*, $\gamma = 1.8$).

External challenges are expected to disrupt host homeostatic state prior to infection, which can lead to contrasted consequences. A host may be, for example, infected through a wound. This wound may cause damage which in turn could either trigger the immune system (*Asri et al., 2019*; *Kenmoku et al., 2017*), and potentially help the host to fight the infection, or hampers the expression of immune defense, and facilitates the infection (*Stoecklein et al., 2012*). We reproduced the later situation by setting $z_0$ above the homeostatic level of damage $\tilde{z}_h$. In our model, this means that defenses are initially hampered by the effects of damage, causing the immune response to occur with a delay (*Figure 3B*, gray curve with $\gamma = 5$ and $z_0 = \tilde{z}_h + 10\%z_d$). This finding has been illustrated in *Chambers et al., 2014*, where the addition of sterile injury to *Drosophila melanogaster* prior to infection increased mortality to *Providencia rettgeri* compared to infections without wounding. This increase in mortality is suggested to be due to a decrease in resistance. In *Figure 3B*, the wound allowed pathogens to proliferate during early infection, but the host nevertheless managed to clear the infection. We demonstrated (see Appendix 1) that this is because the strong constitutive production of defense ($\gamma = 5$) in this simulation guaranties a high homeostatic level of defense. More generally, we proved that clearance is possible if and only if $\gamma \geq \gamma_c$, with

$$\gamma_c = \frac{\varphi/\delta}{G(1/\delta, \eta/(\xi\delta))} \,. \tag{4}$$

Infection resolution is thus possible only when a sufficient level of immune effectors is present before the infection starts. In our simulations, this level $\tilde{y}_h$ is set by the constitutive expression of defense $\gamma$, and characterizes the homeostatic state. As *Equation 4* demonstrates, the level of constitutive expression required to cure the infection decreases when defenses are more efficient ($\delta$ increases) or persist longer ($\varphi$ decreases), and increases with down-regulation of defense production.

The wound did not change the outcome of infection in hosts with strong constitutive immunity (gray curves in *Figure 3A*). It did in hosts with $\gamma = 1.8$, which survived infection when not wounded (green curve in *Figure 3A*) but died from it when wounded prior to infection ($z_0 = \tilde{z}_h + 0.3\%z_d$ in the green curves of *Figure 3D*). The initial immune handicap caused by the wound indeed facilitated infection, so that pathogen load reaches extreme values which eventually killed the host. Environmental challenges other than wounds might have a direct negative impact on defense production, which we could reproduce in our model by setting $y_0$ below $\tilde{y}_h$. This type of challenge could also facilitate infections by opportunistic microorganisms, with trajectories similar to the green curve of *Figure 3D–F*. As expected, hosts with weak constitutive defense can die from infection even when not wounded ($\gamma = 0.5$, red curves *Figure 3D–F*). In the two lethal infections presented in *Figure 3D–F*, the defense collapsed as damage rapidly amassed during the final phase of the disease (see *Figure 3E*), because $\psi > 0$ makes damage hinder defense production. In both cases, damage eventually exceeded the maximum level the host can sustain ($z_d$), and the host died from the infection (see *Figure 3F*).

Although identical in their final outcome, the two simulations of *Figure 3D–F* differ in the times the infection took to kill the host. When constitutive immunity is weak (red curve, $\gamma = 0.5$), pathogens reached high loads before defense managed to slow down their proliferation. The hosts, therefore, succumbed rapidly. When constitutive immunity is stronger but the host is wounded (green

curve, $\gamma = 1.8$), the host went through a phase where pathogen proliferation was under temporary control, and infection transiently stabilized at a SPPL. This SPPL differs fundamentally from that of *Figure 3A–C*, because slowly accumulating damage ended up bringing defense below the level which permits efficient control of pathogen proliferation. This SPPL was, therefore, transient, while that of *Figure 3A–C* was stable.

## Bistability can make the SPPL transient

The host depicted by the green curves of *Figure 3A–C* suffered from a chronic but benign infection; that of the green curves of *Figure 3D–F* died from a severe infection. Still these two hosts and their respective pathogens can be considered as genetically identical, as the two simulations were run with the very same parameter values. The only difference between them was that the host in *Figure 3D–F* had been wounded before being infected. This result demonstrates that our model can be bistable: for a fixed set of parameters, the outcome of an infection can depend on initial conditions.

We were not able to determine sufficient conditions which would guarantee bistability, but we demonstrated that bistability will occur if $F$ decreases for high pathogen loads (i.e. if $u < v$ and $x$ is large) or if $G$ decreases with damage (i.e. if $\psi > 0$, see Appendix 1). We further demonstrated that this conclusion would hold if the damage repair mechanisms were regulated (see Appendix 1 for a complete analysis of bistability in modified versions of our model). The first condition for bistability, where $F$ decreases when $x$ gets large, implies that pathogens have a direct negative effect on defense production, as *Ellner et al., 2021* assumed in their model. The second condition, where $\psi$ is non null, means that damage hinders the production of defense. We, therefore, propose that empirical evidence of bistability, such as in *Figure 1*, indicate that the infection lowers the immune defense either directly, as in *Ellner et al., 2021*, or indirectly, through resource diversion, as in *van Leeuwen et al., 2019*, or damage accumulation, as in our model.

The negative impact of infection on defense production is a necessary condition for bistability, but it is not sufficient on its own. Additional, more specific conditions are required, which cannot be

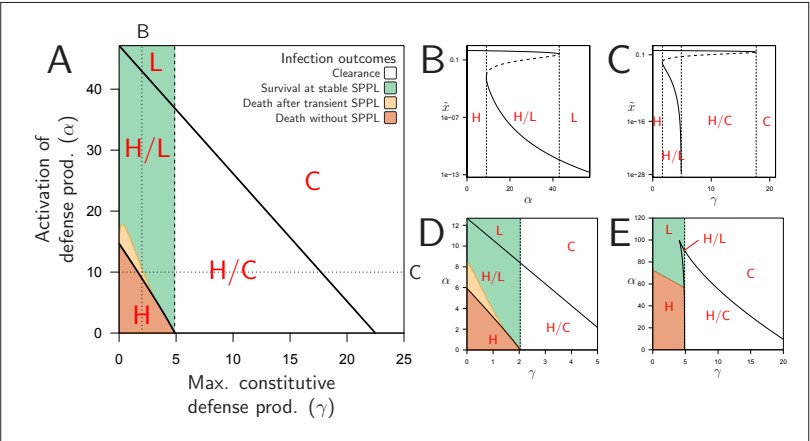

**Figure 4.** Constitutive and inducible defense production determine bistability and infection outcome. (**A**) The types of stable equilibria when maximum rate of constitutive defense production (γ) and defense activation rate (α) vary. Parameters are as in *Figure 2* and both α and γ are varied. The vertical dashed line indicates the value of $\gamma_c$ above which clearance is stable. Red labels indicate which equilibria are stable: H, high load equilibrium, L, low load equilibrium and C, clearance (the distinction between H and L equilibrium is exemplified in figures **B and C**). The two oblique solid lines delimit a parameter region for which the system is bistable (H/L or H/C). The color indicates which equilibrium the infection actually reaches when the initial load is $x_0 = 10^{-6}$ and when $y_0$ and $z_0$ are set to the homeostatic state. (**B**) The equilibrium load $\tilde{x}$ as a function of α when parameters are as in A but γ is fixed to 2 (which corresponds to the vertical dotted line in **A**). Solid lines indicate stable equilibria while dashed lines are unstable ones. The distinction between high load and low-load equilibria is appropriate here because equilibria with intermediate loads are not stable. (**C**) The equilibrium load $\tilde{x}$ as a function of γ with parameters as in A except α fixed to 10 (which corresponds to the horizontal dotted line in A). In the bistable region, the high load equilibrium is always possible; the second possible equilibrium is either low load, when γ is below $\gamma_c$, or clearance otherwise. (**D**) Parameters as in A except $\psi = 0.5$, which reduces the negative impact of damage on defense production. (**E**) Parameters as in A except $u = 0.75$, which lowers defense production at low pathogen load.

mathematically determined. In the following, we focused on two parameters which describes the constitutive and the inducible parts of defense production (namely γ, the maximum constitutive defense production and α, the activation of defense production). In **Figure 4A**, where the two parameters were varied, two lines delimit a region within which the infection dynamics is bistable. When γ is fixed to 2 and only α is varied (which corresponds to the vertical dotted line in **Figure 4A**) the equilibrium load $\tilde{x}$ is very high for low values of α and drops to low values when α is high (**Figure 4B**). For intermediate values of α, the system is bistable: both high load and low load equilibria are possible. The same sort of pattern occurs when α is fixed to 10 and γ only is varied (the horizontal dotted line in **Figure 4A**): the equilibrium load is high when γ is low, the infection is always cleared when $\gamma$ is high, and the system becomes bistable when γ has intermediate values. Inside this bistable region, the high load infection is always possible; the second possible equilibrium is either low load infection, when $\gamma < \gamma_c$, or clearance otherwise (**Figure 4B**).

In bistable situations, the equilibrium load is either high or low, with no possible intermediate situation. This happens because in such systems the two possible stable equilibria are separated by a third, unstable, one that the system cannot reach (the dashed curves in **Figure 4B and C**). Which stable equilibrium will be reached then depends on initial conditions. This is represented by the colored areas of **Figure 4A**, with the inoculum $x_0$ fixed to $10^{-6}$ and $y_0$ and $z_0$ set to the homeostatic state. When only the high load equilibrium is possible, the host always dies from the infection because the equilibrium level of damage $\tilde{z}$ exceeds $z_d$. When only the low load equilibrium is possible, the host survives, and the infection becomes chronic. In the bistable area, the infection also becomes chronic when clearance is not possible ($\gamma < \gamma_c$), with load stabilizing at a SPPL. But two situations must be distinguished here: first, when the immunity is strong enough to control pathogen's proliferation (the green area in 4 A), the infection reaches the low load equilibrium. The host will then survive the infection and load will stabilize at the SPPL for its whole life. Second, when the host immunity is weaker (the yellow area in **Figure 4A**) defense controls proliferation but for a limited period of time, as in the green curve of **Figure 3D**. The SPPL is then transient: the infection will eventually break loose, as accumulating damage causes the collapse of defense, ultimately killing the host.

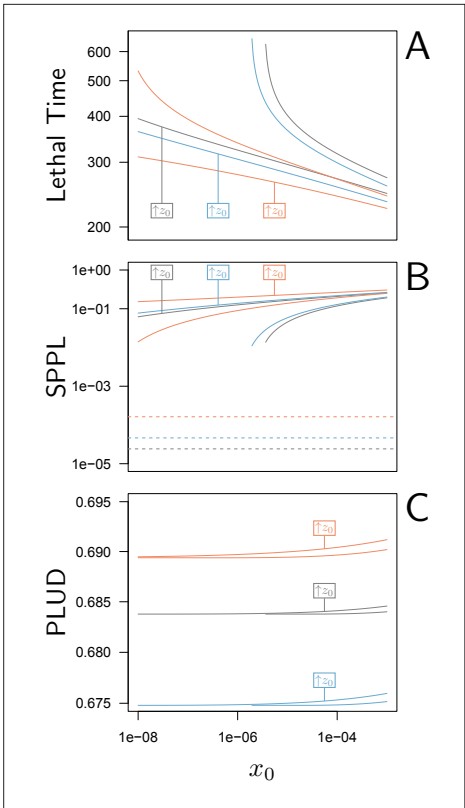

**Figure 5.** Lethal Time, Set-Point Pathogen Load (SPPL), and Pathogen Load Upon Death (PLUD) are three measurable quantities which reflect both resistance and tolerance. The three quantities are represented here as functions of the inoculated dose. The black curves have been computed with parameters as in **Figure 2**, while the red curves correspond to a decrease in resistance (5% decrease in defense efficiency $\delta$) and the blue to a decrease in tolerance (5% decrease in damage repair efficiency $\xi$). For each parameter set, $y_0$ was set at its homeostatic state and $z_0$ was either at the homeostatic state or increased by 1% of $z_d$ (indicated by boxes in the figures). (**A**) The Lethal Time (LT) decreases with inoculated dose and with $z_0$. Decrease in resistance or in tolerance both reduces the time it takes for the pathogen to kill the host. Note that a decrease in tolerance (blue curves compared to black curves) only shifts the relationship between LT and dose, without altering its slope. (**B**) The stable SPPL (dashed lines) is by definition independent from initial conditions. The transient SPPL (solid lines) conversely increases with both $x_0$ and $z_0$. Decreased resistance and decreased tolerance both increase the SPPL. Variations in transient SPPL mirror that of LT, which indicates that the SPPL is a good predictor of the host lifespan. (**C**) The PLUD is almost independent from the initial dose and slightly increases with $z_0$. It increases with lowered resistance (low $\delta$, red curves) but decreases when tolerance is lowered (low $\xi$, blue curves).

Demonstrating experimentally that a SPPL is transient might prove difficult, first because it is not always feasible to monitor pathogen load until the host dies and, second, because even if it was, it may still be challenging to demonstrate that the death of an individual host was actually caused by the infection. Our model offers a solution to this problem. Mathematically, a transient SPPL indeed occurs when the three variables progress slowly, because they have approached an equilibrium point, where all derivatives $dx/dt$, $dy/dt$ and $dz/dt$ are close to zero. However, this equilibrium must be unstable, as the trajectories eventually moves away from it and head towards a high load infection. This indicates, first, that the SPPL cannot be transient unless an unstable equilibrium point exists with $\tilde{x} > 0$, which in turn requires that the system is bistable (this unstable equilibrium corresponds to the dashed curves in *Figure 4B and C*). Second, it also proves that a transient SPPL is not a true equilibrium situation. It is in fact close to the notion of a quasi-static state in thermodynamics. Therefore, the transient SPPL should vary with initial conditions. We have shown that the SPPL does indeed increase with the initial dose $x_0$ when it is transient, while it is independent from initial conditions when stable (see *Figure 5*). These results are in line with empirical results, as the SPPL depends on initial inoculum when *Drosophila* are infected by pathogenic bacteria (*Chambers et al., 2019*; *Duneau et al., 2017a*; *Acuña Hidalgo et al., 2021*), but not when infected by non-pathogenic bacteria such as *E. coli* (*Ramirez-Corona et al., 2021*). We, therefore, propose that testing the link between the SPPL and the initial dose is a way to experimentally demonstrate that the control over infection is transient.

Bistability, as we have already discussed, is possible in our model when accumulating damage negatively impacts defense production. As expected, reducing this impact by lowering the value of $\psi$ reduces the range of parameters which permits bistability. With $\psi = 2$ in *Figure 4A*, 94% of the parameter area where the high load equilibrium is possible is bistable; in *Figure 4D*, where $\psi$ is reduced to 0.5, only 82% of this same area is bistable. Finally, the distinction between high and low load equilibria, although convenient, is not always possible. In *Figure 4E*, with $u = 0.75$, the bistable area forms a closed triangular shape. In this case, when low constitutive defense production (i.e. low γ) does not allow for bistability, a continuum exists between high-load equilibria (low values of α) and low-load equilibria (high values of $\alpha$), without a bistable region in between.

## Load and mortality measurements should reflect both tolerance and resistance to infection

The SPPL and the PLUD can both be estimated from experimental infections, as earlier work by *Duneau et al., 2017a* has shown. They have been proposed to reflect different aspects of the host immune response: the SPPL, stable or transient, being a pathogen load stabilized by the host immunity, may be taken as a proxy for resistance; the PLUD, being the maximum pathogen load the host can endure, may rather quantify the host's tolerance to infection. Measurements of host survival to infection are also commonly used as proxies of resistance or tolerance (e.g. *Gupta and Vale, 2017*; *Louie et al., 2016*). Here, we have used our model to question the way these quantities are used as surrogate measurements for tolerance or resistance. For this purpose, we computed them from simulated infections and surveyed how they relate to immunity parameters (see Appendix 3,4 and 5 for a complete analysis).

As expected, the time it takes for a pathogen to kill its host decreases if inoculum size is increased (*Figure 5A*) or if the host is wounded prior to infection (i.e. when $z_0$ is increased by 1% of $z_d$ above the homeostatic state). Lethal Time (LT) thus strongly depends on the conditions in which the infection has been initiated. Our simulations further show that any genetic change in the host or the pathogen that would reduce the host resistance (red curves in *Figure 5A*) or tolerance (blue curves in *Figure 5A*) would accelerate death. We also showed that contrary to what some authors have proposed (e.g. *Gupta and Vale, 2017*; *Louie et al., 2016*), a reduction in tolerance does not necessarily modify how LT relates to dose; a reduction in resistance, conversely, makes LT less dependent on dose (compare red and blue curves in *Figure 5A*, see Appendix 5 for a complete analysis on this point). Hence, the comparison of lethal time in response to different doses is more likely to characterize differences in resistance than in disease tolerance.

The SPPL, when it is stable, does not depend on initial conditions (dashed lines in *Figure 5B*). This is because a stable SPPL is an equilibrium towards which the infection will converge, no matter where it starts from, as long as the host survives the infection. The transient SPPL, conversely, is not an equilibrium. We showed that it does increase with dose and is higher when the host is wounded

prior to infection. Our simulations finally indicate that both transient and stable SPPL increase when either tolerance or resistance are impeded (see Appendix 1 and 3 for a complete analysis). In fact, if by definition stable SPPL does not correlate with the time the host will die, variations in the transient SPPL almost exactly mirrors that of the LT. This is because the control of proliferation does not last long when the transient SPPL is high: infections which maintain high pathogen loads rapidly progress towards host death. A first practical consequence of this observation is that measurements of the SPPL and of the LT bear the same information on the host immunity. Therefore, it is useless to measure both, unless one wants to predict transmission rate, which should quantitatively depend on both bacterial load and infection duration. A second practical consequence is that the SPPL, when transient, can be used as a predictor of the infected host's lifespan, as already demonstrated in HIV infections (*Fraser et al., 2007*; *Mellors et al., 1996*).

The two hosts of *Figure 3D–F* die at very different times but at almost indistinguishable PLUD. This is confirmed by *Figure 5A*, where the PLUD does not vary with changes in inoculum size that do otherwise yield threefold variations in LT (compare *Figure 5C* to *Figure 5A*). More generally, the PLUD is only weakly influenced by the conditions in which the infection has been initiated, albeit we predict that it should slightly increase when the host is wounded. Our model, therefore, reproduces an important property of the PLUD which *Duneau et al., 2017a* have documented (see *Figure 1*): for a given pair of host and bacterial pathogens, the PLUD in *D. melanogaster* is almost constant and does not correlate with the time to succumb from the infection. In our model, this comes from the fact that the pathogen load $x$ evolves much faster than other variables at the start of the infection (as detailed in Appendix 4). Pathogens thus rapidly reach the highest possible load allowed by the current amount of defense ($x_t \lesssim 1 - \delta y_t$). The dynamics of $x$ is then 'enslaved' by that of $y$ and $z$ (*Van Kampen, 1985*) and, as the disease enters its final stage, the load increases slowly while $y$ gradually diminishes under the effect of accumulating damage. Because of this very particular dynamics, the way $x_t$ relates to $z_t$ will be very similar for all infections that enter their final phase, which in the end renders the PLUD almost independent from the conditions that prevailed at the onset of the infection.

Another particularity of the PLUD, is that it increases when resistance is diminished (red curves in *Figure 5C*) but decreases when tolerance is lowered (blue curves in *Figure 5C*). The PLUD is, therefore, a composite measurement which reacts to both resistance and tolerance traits, just like the SPPL and the LT, but has the unique feature that it responds differently to these two types of variation (see Appendix 4 and the following section for a more complete elasticity analysis).

## Combining PLUD and hazard ratio (HR) to characterize the host's ability to handle infections

We have seen before that variations in resistance and variations in disease tolerance have distinctive impacts on the PLUD. Still, a host lineage with higher than average PLUD could either be less resistant or more tolerant than other lineages. The PLUD is, therefore, difficult to interpret by itself, but we shall demonstrate that its relationship with mortality measurements are informative. To investigate this, we performed elasticity analyses of the PLUD and the Hazard Ratio, a measure of death risk relative to a chosen reference host (HR, see Appendix 6 for definitions and analyses). As our model allows to determine the time to death of a given infection, the HR can be obtained from a Cox proportional hazard model (*Cox, 1972*) adjusted on simulated survival curves.

As expected, we found that reducing the risk to die from the infection can be achieved through a tighter control of pathogen proliferation or, alternatively, by mitigating the damage the infection causes (*Figure 6A and B*). For example, HR decreases both when the defense is made more efficient ($\delta$ increases) and when damage repairing is accelerated ($\xi$ increases).

Variations in PLUD are more complex, but *Figure 6A* suggests that reducing damage (e.g. by increasing damage repair, $\xi$) or making the host more damage-tolerant (by increasing the maximum damage level the host can sustain, $z_d$) should increase the PLUD. As a result, variations in parameters which have a direct impact on the dynamics of damage produce a negative correlation between PLUD and HR. *Figure 6B* further suggests that anything that reduces pathogen proliferation, like increasing the production or efficacy of defense ($\alpha$ or $\delta$), comes with a decreased PLUD. Variations in parameters that control the defense, therefore, result in a positive correlation between the PLUD and HR. In summary, *Figure 6* indicates that faster death (i.e. increased HR) would indicate lower tolerance when associated with lower PLUD, when it would rather suggest lower resistance in case of higher PLUD.

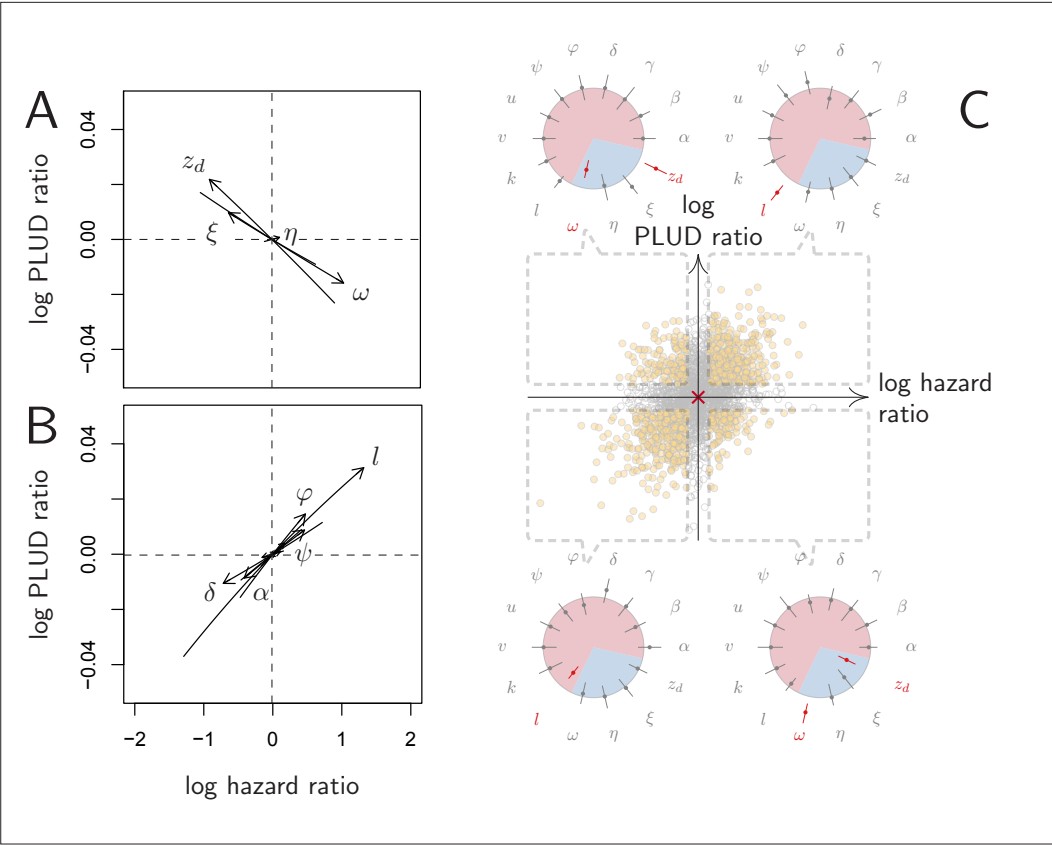

**Figure 6.** Effect of parameter variations on Pathogen Load Upon Death (PLUD) and hazard ratio (HR). In (**A and B**), parameters are initially fixed as in *Figure 2* except for $\gamma = 0.5$, as in red curve if *Figure 3*. Parameters are then varied one at a time, from −5 to +5%. For each modified parameter value, the PLUD was computed and 100 survival simulations were run with $x_0$ randomly drawn from a log-normal distribution with average $10^{-6}$ and variance 0.5. A Cox proportional hazard model was then fitted on simulated data so that log Hazard Ratio (log HR) could be related to parameter variation. The horizontal dashed lines would correspond to parameters which have no influence on PLUD; the vertical dashed line would correspond to those which do not impact HR. (**A**) The effect of tolerance parameters (defined as in *Figure 2A*). Arrows indicate that, as expected, any increase in the damage induced by the pathogen ($\omega$) decrease the PLUD, while increasing damage repair ($\xi$) or tolerance to damage ($z_d$) has the opposite effects. For the parameter set we have used here, increasing $\eta$ has almost no effect on PLUD and only slightly increases HR. (**B**) The effects of resistance parameter (defined as in *Figure 2A*). Variation in any of these parameters produces a positive correlation between PLUD and HR. (**C**) The variation in PLUD and HR when all parameters are all randomly drawn from independent Gaussian laws. The red cross corresponds to a strain with parameters as in *Figure 2* and each dot is a 'mutant strain' which parameters have 95% chances to deviate by less than 0.5% from that of the 'wild-type strain.' Open circles are mutants which either log PLUD ratio or log HR ratio do not significantly deviate from zero (see text for details). We sorted mutants in four groups, according to whether their PLUD and HR has significantly increased or decreased compared to the wild strain. The distributions of traits are indicated for each group on colored circles, with segments spanning 75% of the deviation range and the central dot corresponding to the median deviation. Segments in red indicate that 75% of the mutants have either lower trait values than the wild-type strain (when the segment is inside the circle) or higher trait values (when outside).

The case of $\eta$, which quantifies the importance of defense side effects, demonstrates that although appealing, this method has its limits. This parameter clearly relates to damage production and we demonstrate that the lower bound of the PLUD expectedly decreases when $\eta$ increases (see *Equation (S4-1)* in Appendix 4). But when immunity is weak, this effect can reverse, with increasing $\eta$ impairing control and resulting in higher PLUD (as we demonstrate in our elasticity analysis, see *Appendix 4— figure 2* when both α and γ are low). In hosts with very weak immunity, therefore, variations in the detrimental side-effects of defense ($\eta$) can create a positive correlation between PLUD and HR (as in *Figure 6A*). In addition, *Figure 6A* shows that different forms of variation in tolerance (e.g. variations

in pathogenicity $\omega$, in damage repair $\xi$ or in $z_d$ the maximum level of damage the host can sustain) will produce the same signal. *Figure 6B* demonstrates a similar situation for variations in resistance. Therefore, experimental measurements of PLUD and HR can be used to distinguish variations in tolerance from variations in resistance, broadly speaking, but will likely fail to identify the precise mechanisms involved.

The practical use of the above proposed method will largely depend on which specific parameters differ the most among the hosts and pathogens to compare. We cannot predict much on this matter, as everything will obviously depend on the genetic variation of these traits. A crude way to test the utility of the method, though, is to let all parameters vary and see which effect predominates. In *Figure 6C*, we randomly drew all 14 parameters from independent Gaussian laws with average set at the parameter values given in *Figure 2*, and a variance chosen so that 95% of the random values differ by less than 0.5% from this average. If we consider that the parameter set given in *Figure 2* represents a wild-type host, each random parameter set can be considered as being a mutant strain. We randomly sampled 2000 such mutants. We then simulated 100 survival curves for each of them, which we compared to that of the wild type by using a Cox proportional hazard model to estimate HR. We used the same simulations to compare the PLUD of the mutants to that of the wild-type, as if load was estimated by plating. For that purpose, we randomly drew numbers of Colony Forming Units (CFU) from Poisson distributions with average given by the predicted WHD, which we then compared between the mutant and the wild-type hosts using a Poisson glm. We have drawn one number of CFU per host from a Poisson distribution with average 1000 times the simulated PLUD, so that we have similar statistical powers when comparing PLUDs and HRs. In *Figure 6C*, the red cross lying on the origin is the wild-type; mutants which significantly deviate from it (closed circle) are separated in four groups, according to the deviation signs. For each group, we compared the distribution of mutant traits to that of the wild-type. We found that mutants with significantly higher HR and PLUD have almost systematically a stronger down-regulation of defense production (with clearly higher value of $l$) and a slightly lower efficiency of defense ($\delta$). Hence, such mutants die faster than the wild-type because they are less resistant. Mutants with higher HR but lower PLUD have most often a lower tolerance to damage (lower $z_d$) and are more susceptible to the pathogen virulence (greater $\omega$). These mutants, therefore, die faster than the wild-type strain because they are less disease tolerant.

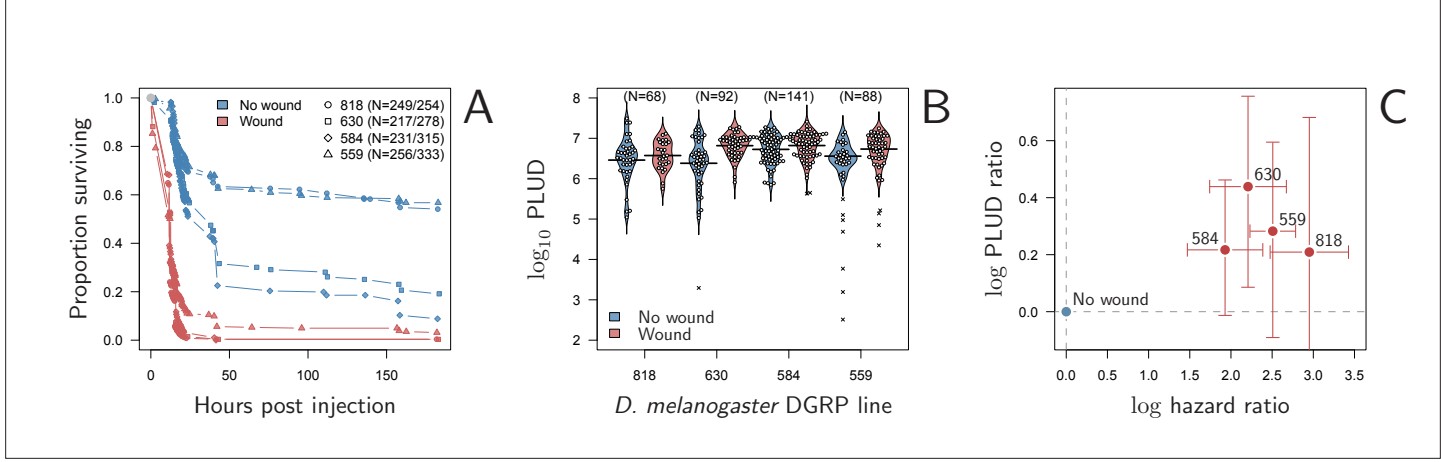

**Figure 7.** Hazard ratio (HR) and Pathogen Load Upon Death (PLUD) estimations on *D. melanogaster* infected with *Providencia rettgeri* with or without thorax wound prior to injection. We performed the experiment on four genotypes sampled from the *Drosophila* Genetic Reference Panel (lines RAL-818, RAL-630, RAL-584 and RAL-559, which have no known difference in immunity effectors). (**A**) Proportion of surviving flies as a function of hours post injection. Numbers in legend indicate sample sizes for no wound/wound treatments, respectively. In all lines, the wound significantly and sharply reduces survival. (**B**) PLUD for each *D. melanogaster* line. Each point represents an individual fly and the bars represent the means. Crosses are PLUD estimates which have been categorized as outliers by a Rosner test. (**C**) Log PLUD ratio as a function of log Hazard Ratio. For each line, we computed the log-ratio of PLUD in wounded flies to that in non-wounded (with outliers excluded). The value of zero, therefore, corresponds to a non-wounded reference, as for log-HR. Bars represents the 95% CI obtained by bootstrap. The wound consistently increases both the HR and the PLUD, as our model predicts.

## Wounding *Drosophila melanogaster* increases the PLUD

Our model predicts that when damage hinders defense production, hosts that are wounded prior to infection should have higher PLUD (see *Figure 5C*). We tested this prediction by measuring the PLUD of the bacterial pathogen *Providencia rettgeri* injected in *D. melanogaster* that were either injured in the thorax before being injected in the abdomen or not injured (see *Figure 7* and Appendix 7 for methodological details). Wound could in principle increase defense, e.g. as hemocyte innate immune training triggered by DAMPs (*Chakrabarti and Visweswariah, 2020*), but hemocytes have no detectable effects on infections caused *by Providencia rettgeri* (*Duneau et al., 2017a*). Therefore, we do not anticipate any positive impacts of the wound. Instead, we expect that the wound decreases resistance, as indicated by *Chambers et al., 2014*. We found that indeed, the wound significantly reduced survival (Cox proportional hazard model with random experimental blocks: $X^2$ = 1397.44, degrees of freedom (df) = 1, p - value < $2.2e^{-16}$). PLUD estimates were found to be highly variable, with some individuals dying at load less than $10^4$. These individuals may have died from reasons other than the *P. rettgeri* infection; we, therefore, conducted two analyses, with these individuals either included or removed (outliers being identified by a Rosner test, see Appendix 7 for details). We found in both analyses that wounded hosts die at a PLUD significantly higher than non-wounded (linear mixed model with random experimental blocks: $X^2$ = 11.29, df = 1, p - value = 0.008 with outliers removed; $X^2$ = 11.04, df = 1, p - value = 0.0009 with outliers retained) with no significant difference among genotypes (p - value = 0.73 and p - value = 0.59) with outliers removed or not removed, respectively although the effect is much clearer on line RAL-630 than in the three other lines (see *Figure 7*). Overall the wound does increase both hazard ratio and PLUD, as our model predicts. This result can be taken as an evidence that inflicting damage to *D. melanogaster* hinders the production of defense, and thus indirectly reduces resistance.

## Suppressing *Drosophila melanogaster*'s active effectors increases the PLUD

Our model also predicts that mutations that decrease resistance should in most cases increase the PLUD. This seems in contradiction with *Duneau et al., 2017a* who did not find any effect of immunosuppression on the PLUD in *P. rettgeri* infections. The original analysis in this paper used a linear mixed model where the effects of Imd, Melanization and Toll pathways were assessed together. We reanalized the dataset and examined each pathway separately. We observed that the PLUD exhibited an increase when the Imd pathway was suppressed (Kruskal-Wallis: $X^2$ = 7.87, df = 1, p-value = 0.005) or when melanization was inhibited (Kruskal-Wallis: $X^2$ = 15.56, df = 1, p-value = $7.9e^{-5}$). However, there was no significant change in PLUD when the Toll pathway was suppressed (Kruskal-Wallis: $X^2$ = 0.03, df = 1, p-value = 0.86).

A limitation of the previous study is that the wildtypes used were not ideal genetic controls, given that the mutants did not undergo backcrossing. In order to strengthen the test of our theoretical prediction, we quantified the PLUD in *D. melanogaster* mutants with some immune effectors (AMPs) either deleted or silenced, and compared it to PLUD measured in appropriate controls (see Appendix 7).

In a first experiment, we used two isogenic lines sharing the same genetic background but from which genetic deletions have induced a loss-of-function either in Defensin (line *A*) or in Drosocin, Attacins, and both Diptericins (line *B*). We also used a third mutant line (line *AB*) which combines the loss-of-function of both lines *A* and *B*. Line *A* lacks Defensin, an antimicrobial peptide strongly upregulated upon *P. rettgeri* infection (*Duneau et al., 2017b*; *Troha et al., 2018*) but with moderate effects on survival (*Hanson et al., 2019*). Lines *B* and *AB*, which both lack Diptericins, are conversely expected to be highly susceptible to *P. rettgeri* (*Hanson et al., 2019*; *Unckless et al., 2016*). We have used as a control a fourth mutant line which lacks Bomanins (line $Bom^{\Delta 55C}$). $Bom^{\Delta 55C}$ is a good reference as it is mutated in the same background than the other genotypes, but the AMPs it lacks are mostly involved into fighting Gram-positive bacterial infections (*Clemmons et al., 2015*). In addition, previous experiments have shown that $Bom^{\Delta 55C}$ mutants have a PLUD comparable to controls when infected with the gram positive bacteria *Enterococcus faecalis* (*Lin et al., 2020*).

We found that, compared to line $Bom^{\Delta 55C}$, lines *A*, *B*, and *AB* have higher death rates when infected by *P. rettgeri* (*Figure 8A*; Cox proportional hazard model with random experimental blocks: $X^2$ = 138.12, df = 3, p-value = $9.6e^{-30}$) and higher PLUD (*Figure 8B*; linear mixed model with random experimental

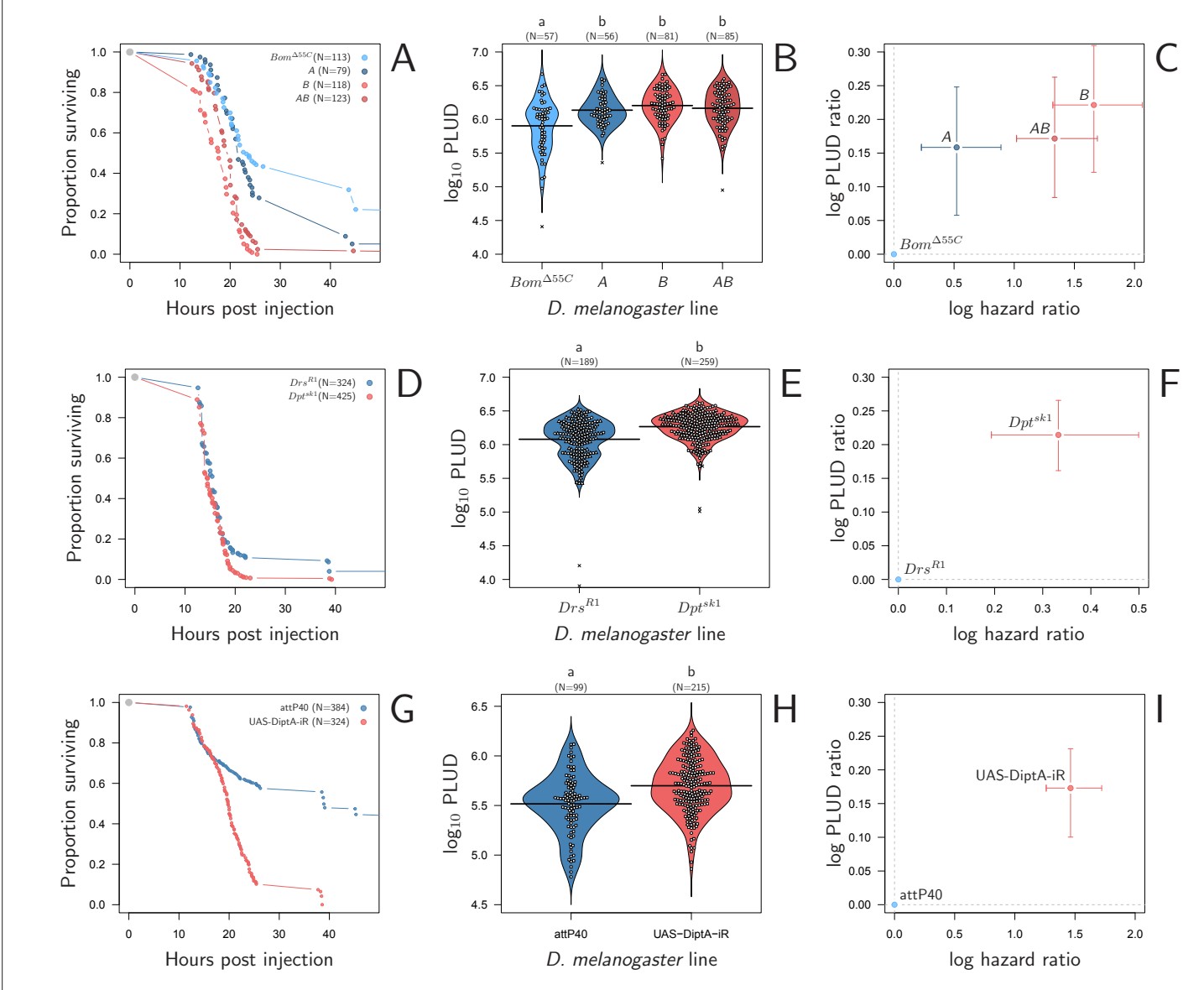

**Figure 8.** Hazard ratio (HR) and Pathogen Load Upon Death (PLUD) estimations in immuno-deficient lines of *D. melanogaster* infected with *Providencia rettgeri*. (**A**, **D**, and **G**) give the proportion of surviving flies as a function of hours post-injection. (**B**, **E**, and **H**) present per fly estimations of PLUD. Black lines represent the means, different symbols correspond to distinct replicate experiments, and small letters above graph indicate significant differences, as tested by a pairwise Wilcoxon test. (**C**, **F**, and **I**) present per line estimations of log PLUD ratios, as a function of estimations of log HR. Ratios are here computed relative to the control line and bars represents the 95% CI as obtained by bootstrap. In (**A-C**), three lines which lack effectors active against *P. rettgeri* (A Defensin deleted, B all Diptericins deleted and AB both Defensin and Diptericins deleted) are compared to $Bom^{\Delta 55C}$. $Bom^{\Delta 55C}$ lacks Bomanins, immune effectors which are inactive against *P. rettgeri*, and is thus used here as a control. All three lines die significantly faster than $Bom^{\Delta 55C}$ and have a significantly higher PLUD (**B and C**). In panels (**D-F**), following a similar logic, we compared $DptA^{SK1}$, which has Diptericin A deleted, to $Drs^{R1}$, which has Drosomycin deleted. Diptericin A has been shown to be the most active AMP against *P. rettgeri* while Drosomycin is active mostly against gram positive bacteria and fungi. (**D**) demonstrates that deleting Diptericin A is sufficient to make the mutant more susceptible to the infection than *Drs* mutant. (**E** and **F**) demonstrate that this increase in susceptibility goes together with an increase in PLUD. In (**G-I**) we used RNAi to silence Diptericin A. The control is the match genetic background recommended for the TRiP RNAi panel. Silencing Diptericin A accelerates death (**G**) and significantly increases the PLUD (**H-I**).

blocks: $X^2$ = 17.01, df = 3 , p-value = 0.0007). We challenged this first result by comparing another mutant that, contrary to line *B*, lacks only Diptericins ($Dpt^{SK1}$) (*Hanson et al., 2019*). We compared this mutant to $Drs^{R1}$, a mutant which shares the same genetic background (*Hanson et al., 2019*) but has Drosomycin deleted, Drosomycin being an antifungi AMP that is inactive against bacteria (*Fehlbaum*

*et al., 1994*). We found again that death is faster and PLUD higher when Diptericins are deleted (*Figure 8D–F*; linear mixed model with random experimental blocks: $X^2$ = 10.11, df = 1, p-value = 0.001). But if *Bom*$^{\Delta 55C}$ and Drosomycin are clearly less important for resistance against *P. rettgeri* than Diptericin, studies start to suggest that they may play a role in tolerance to fungi toxin or are involved in other functions which could influence tolerance (*Araki et al., 2019*; *Xu et al., 2023*). Therefore, we also challenged our hypothesis by injecting *P. rettgeri* in flies that constitutively express, thanks to an ubiquitous driver (Actin5C-Gal4), a RNAi which silences specifically Diptericin A. This time we used as control line the advised match genetic background for the attP40 genetic background of the TRiP genetic RNAi panel. We found that flies with silenced Diptericin A die faster and at a higher PLUD than controls (*Figure 8G–I*; linear mixed model with random experimental blocks: $X^2$ = 8.42, df = 1, p-value = 0.0037). Overall, based on previous results and on those three different approaches, we concluded that a reduction in resistance caused by a lack of effectors provokes an increase in both HR and PLUD. Finally, it should be noticed that, as our model predicts, the increase in PLUD is remarkably constant throughout experiments (approx. +0.2 in *Figure 8C, F and I*) even though the HR varies from approx. +0.3 to +1.5.

## Suppressing catalase expression in *Drosophila melanogaster* decreases the PLUD

No gene is unambiguously identified as involved in disease tolerance in *D. melanogaster*, probably in part because damage are uneasy to quantify experimentally. *CrebA* and *Bombardier* are candidate genes (*Troha et al., 2018*; *Lin et al., 2020*), but they have been determined as such using the PLUD as a tolerance proxy, which would make our reasoning circular. Other genes have been proposed, but their roles were mostly identified using approaches that our model suggests are limited for establishing a proxy of disease tolerance (*Figure 5A*, see Appendix 5). We thus decided to test our method on a gene which function is well understood and should contribute to disease tolerance.

Reactive oxygen species (ROS) have been shown to be critical agents in oxygen toxicity, disrupting the structural and functional integrity of cells. Catalase is among several enzymes involved in scavenging oxygen free radicals and protecting those cells. The activity of Catalases into reducing reactive oxygen species has been shown in *Drosophila* infections (*Ha et al., 2005*) and mosquito systemic infections (*DeJong et al., 2007*), making it a strong candidate for a tolerance gene.

We found that silencing Catalase ubiquitously with Actin-Gal4 driving an RNAi does increase flies susceptibility to systemic bacterial infection when comparing to a match genetic background as control (i.e. attP2 control line, the advised RNAi control for the attP2 genetic background of the TRiP genetic RNAi panel, see *Figure 9A*). The log(HR) is comparable in strength to that of silencing Diptericin A. We further observed that silencing Catalase does decrease the PLUD (see *Figure 9B*; linear mixed model with random experimental blocks and a correction for heteroscedasticity: $X^2$ = 23.33, df = 1,

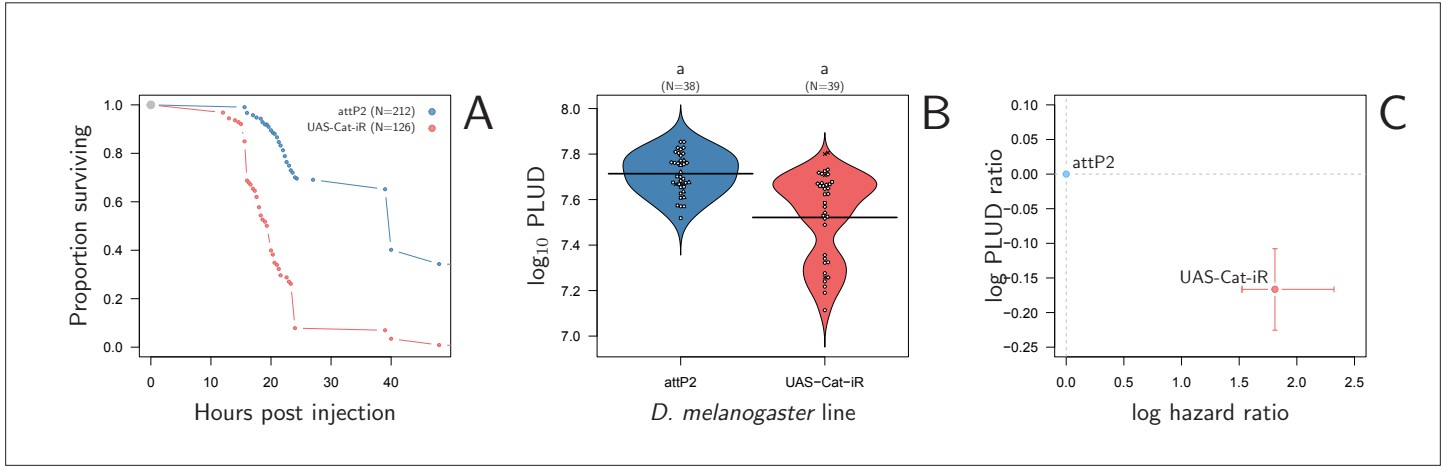

**Figure 9.** Hazard ratio (HR) and Pathogen Load Upon Death (PLUD) estimations in *D. melanogaster* with RNAi-silenced Catalase infected with *Providencia rettgeri*. As in *Figure 8G–I*, the control is the match genetic background recommended for the TRiP RNAi panel. Silencing Catalase accelerates death (panel **A**) and significantly decreases the PLUD (panels **B-C**).

p-value = $1.37e^{-6}$). Again, although the level of PLUD was different, likely due to the difference in genetic background, the relative difference (i.e. log(PLUD ratio)) was comparable in strength to that of silencing Diptericin A, but of opposite sign. Therefore, based on its known functions and the fact that our experimental results follow the predictions of our theoretical model, we propose that Catalase is part of the arsenal that enable a fly to tolerate bacterial infections.

## Discussion

Signs and symptoms of infectious diseases vary widely: severe diseases can cause rapid and certain death if untreated, while benign diseases may go almost unnoticed. Pathogen load is the proximal cause distinguishing severe infections from benign ones. However, the factors determining pathogen load itself are complex, as pathogen proliferation is influenced not only by environmental conditions but also by the interacting genetically fixed characteristics of both the host and the pathogen. This is often simplified by categorizing host and pathogen traits into two types: those that determine resistance to infection (how effectively the immune system controls pathogen proliferation) and those that allow tolerance to the damage inflicted by the infection. To explore these concepts of resistance and tolerance, we have developed a process-based model of Within Host Dynamics (WHD) of pathogens. Our approach involves studying how the proxies used in the literature to quantify resistance and tolerance relate to the processes encompassed by our model.

To ensure this approach is possible, our model behavior needs to match experimental evidence. We first found that it recreates the most commonly observed types of WHD documented to date in *D. melanogaster* and in other insect host systems. The model thus provides a general theoretical framework which can be useful to investigate most situations of experimental infection. Most importantly, it finely reproduces the properties of two experimental proxies which have been used to evaluate resistance or tolerance to disease: the SPPL increases with dose in chronic lethal infections but not in chronic benign infections; the PLUD increases if the host is wounded before infection but is insensitive to inoculum size. We take these results as evidences that the biological basis on which we grounded the model are sound.

We shall in the following section summarize how the SPPL and the PLUD relate to model parameters and how they can yield insights into resistance and tolerance. A general conclusion is that they reflect both mechanisms. We were nevertheless able to predict what information can be gained from each of them and propose experimental methods to tease apart resistance and tolerance effects.

### Exploring chronicity: Stable and unstable set-point pathogen loads

When a host survives the infection but does not successfully clear the pathogen, the disease enters a phase of chronicity (*Virgin et al., 2009*) and load stabilizes at a SPPL. Such chronic phases characterize a large range of infectious diseases, and we have shown that this is determined by the host immune defense. Notably, strong constitutive defense prevents chronicity because it permits the host to clear pathogens; weak constitutive defense combined to slow activation upon pathogen detection also prevents chronicity because hosts then die from the infection.

A more surprising prediction of our model is that chronicity can be transient, the SPPL being then unstable. This happens when the damage which accumulates during the infection eventually obstructs the production of immune defense. Pathogen proliferation is then unleashed which leads to host death. Illustrations of such infections could be, in humans, *Mycobacterium tuberculosis* causing tuberculosis infections or HIV causing AIDS; in *Drosophila*, this could be *Providencia rettgeri* infections. When the SPPL is stable, by contrast, the host never clears pathogens but it will not die either from the infection. In human, this could be *Porphyromonas gingivilis* causing gum infections; in *Drosophila*, this could be *Escherichia coli* infections. Our model clearly highlighted those two types of chronic infections and suggests that their SPPL cannot be interpreted the same way.

In particular, we found that unstable SPPLs depend on how the infection was initiated. For example, we predict that they should be higher when the host is injured prior to contamination and should correlate positively with the inoculum size (see *Figure 5*). These predictions agree with previous study of bacterial infection of *D. melanogaster* (*Acuña Hidalgo et al., 2021*; *Chambers et al., 2014*; *Chambers et al., 2019*; *Duneau et al., 2017a*). Our model predicts that stable SPPLs are, conversely, independent from how the infection is initiated. In particular, they are insensitive to inoculum size, in

accordance with benign *Escherichia coli* infections of *D. melanogaster* (*Ramirez-Corona et al., 2021*). The stable SPPL, therefore, reflects stable genetically determined characteristics of both the host and the pathogen, but it is affected by traits related to both tolerance and resistance. For example, low stable SPPL could indicate either that the host has efficient immune defense, or that it tolerates well infections.

This is not to say that no information can be gained from the study of SPPL, be it stable or unstable. In probably all chronic infections, everything being equal, the pathogen load during the chronic phase is an appropriate tool to compare the capacity to be transmitted to another host: the higher the SPPL, the most the host sheds pathogens (*Fraser et al., 2007*; *Matthews et al., 2006*). The unstable SPPL is also an important concept because it is a good predictor of the duration of chronicity: we found indeed that the higher the SPPL, the earlier the host dies from the infection. Monitoring survival to infection and transient SPPL is, therefore, providing redundant information. This is particularly well established in the specific case of HIV infections, where high SPPL is associated to short asymptomatic phases and rapid progression to the final stage of the disease (*Fraser et al., 2007*). The SPPL in HIV infections has also raised considerable attention because of two striking characteristics which are not fully understood. First, it is highly variable among patients; second, it is heritable, which means that the SPPL of a newly infected patient positively correlates to that of the person it has been infected by. The most common explanation of this correlation is that part of the variation in SPPL is caused by genetic variation in viruses. But if variance in SPPL has a genetic origin, the SPPL should evolve during the course of the infection and load should, therefore, increase over time.

The heritable nature of SPPL, therefore, contradicts the observation that load is constant throughout the asymptomatic phase of the disease (although some possible solutions to this paradox have been proposed *Bonhoeffer et al., 2015*; *Hool et al., 2014*). Our model suggests that part of the large variance in SPPL could be non-genetic yet heritable. If transmission events occur during chronicity, the inoculum size should indeed increase with the SPPL of the donor host (as suggested by *Fraser et al., 2007*). As we have demonstrated that the transient SPPL does increase with inoculum size, the SPPL of a newly contaminated host should reflect that of the donor host. The fact that the transient SPPL increases with inoculum size could, therefore, suffice to create a form of non-genetic heritability in SPPL.

Finally, we propose that quantifying the SPPL for various doses in a chronic disease would allow, if the SPPL increases with dose, to demonstrate that it is unstable. This could be used as an experimental method to prove that chronicity is transient. In experimental systems which our model reproduces, it would in addition demonstrate that the system is bistable (a necessary condition for the existence of unstable SPPL) which in turn requires that the infection impairs immunity by reducing the production of defense, either directly as in *Ellner et al., 2021* or indirectly through the accumulation of damage (*Stoecklein et al., 2012*; *van Leeuwen et al., 2019*; *DeJong et al., 2007*), or by making defense less efficient (e.g. *Zhang, 2016*). Hence, showing that the SPPL is transient informs on the fact that the pathogens, or the damage they cause, obstruct the immune response. The positive link between dose and SPPL has been found in some experiments (*Shultz et al., 2016*; *Chambers et al., 2019*); additional work is required to demonstrate that the explanation our model provides applies to these experiments.

We have so far assumed that pathogens do not evolve during the course of the infection. We did not consider this possibility, first because it is beyond the scope of our study, and second because *Duneau et al., 2017a* demonstrated that within-host evolution did not explain bifurcation in their experiments. But in situations where, for example, frequent mutations arise that make pathogens resistant to host immunity, a phenomenon resembling bistability could occur: hosts where resistance has evolved would be killed by the infection while others would control it and survive. If this happens, bistability would have nothing to do with damage hindering defense. But this does not compromise our predictions because the SPPL would then probably not vary with dose (as demonstrated by *Ellner et al., 2021* in the modified version of their model where bacteria are protected from AMPs).

## Characterizing lethal diseases: The pathogen load upon death

The Pathogen Load Upon Death is the maximum pathogen load a host can sustain before it dies from the infection. It has been logically used as a measure of disease tolerance (*Duneau et al., 2017a*; *Huang et al., 2020*; *Lin et al., 2020*; *Troha et al., 2018*) and of pathogenicity (*Faucher et al., 2021*).

As expected, our model confirms that the PLUD strongly depends on tolerance to damage ($z_d$, see Appendix 4). However, if the PLUD was an unambiguous quantification of tolerance, as proposed by *Duneau et al., 2017a*, it should not depend on any parameter other than $z_d$, or at least not on parameters that determine the control of pathogen proliferation. We conversely predict that hosts with an impeded immune response are expected to have a higher PLUD, which we confirmed with an experimental design more powerful than that of *Duneau et al., 2017a* (see *Figure 8*).

In our model, the PLUD increases when less defense are produced (e.g. when α or γ decrease, see *Appendix 4—figure 2*) or when defense are made less efficient (i.e. when $\delta$ is decreased, see *Figure 6*). This is because the PLUD is a load which, as such, reflects the ability of pathogens to proliferate inside the host. Hence, like the transient SPPL and probably like any quantity derived from load estimations, the PLUD is subject to mixed influences of immune control of proliferation and of damage mitigation.

The PLUD, unlike the transient SPPL this time, is independent from inoculum size: flies injected with large inoculum die faster but at the same load than flies infected with small inoculum (*Duneau et al., 2017a*). Our model reproduces this observation, and we have demonstrated mathematically that this happens because pathogen load is a 'fast variable' (*Van Kampen, 1985*, see Appendix 4). In practice, once the accumulation of damage due to the infection is starting to affect the immune system in a way that the defense are impeded, the system is entering a run-away process: pathogens proliferate faster, inflicting new damage to the host, which decrease the defense further. Infections that enter this dynamics tend to all follow the same typical trajectory, and therefore kill the host at the same pathogen load. We found that this occurs in most parameter combinations (see Appendix 4), except when hosts die at very low damage (i.e. when $z_d$ is low). In that particular case, death occurs so rapidly that infections have no time to converge to their typical final trajectory and the PLUD depends on inoculum size. This could represents situations where the host is already extremely weak or about to die at the time of the infection. However, in most relevant cases, the PLUD being independent of how the infection has started, it reflects the genetically fixed traits of both hosts and pathogens that influence disease tolerance and resistance.

## The PLUD as an aid to compare disease tolerance and resistance among hosts or diseases

It has been proposed that disease tolerance could be quantified as the slope of a linear relation of time to death to a load measured at a fixed point during the infection (e.g. *Louie et al., 2016*; *McCarville and Ayres, 2018*) or with the linear or logistic relationship between initial inoculum size and time to death (e.g. *Gupta and Vale, 2017*). However, our analysis strongly suggests that these approaches mostly measure resistance (see *Figure 5*). This is probably due to the fact that any pathogen load reflects resistance mechanisms. Instead, we propose to use the relation between the HR and the PLUD ratio to determine whether differences in susceptibilities are mainly due to deficiencies in tolerance or resistance mechanisms. Our method is similar but differs in two key aspects. First, we propose using a load that reflects a specific moment of the infection rather than a load measured at an arbitrary, fixed time. This approach ensures that loads can be compared among hosts and among pathogens, as they will always have the same biological interpretation. Second, we used measures of Hazard Ratio obtained for a fixed inoculum size, as changing the inoculum size has little impact on the relation between the PLUD and the Hazard Ratio, unless very high inoculum are used, which can lead to artifacts by artificially bypassing the immune response.

Observing that a particular genotype of host has a lower than average PLUD cannot indicate whether this genotype is less tolerant or more resistant than others to the infection. However, our model shows that if this low PLUD genotype also has a higher than average risk to die from the infection (or a shorter time to death) its susceptibility is probably due to a default in tolerance to infection. If conversely a genotype has a higher than average risk to die with a higher PLUD, we would conclude that its susceptibility is mostly due to a default in resistance (see *Figure 6*).

We confirmed this prediction in the case of *P. rettgeri* infecting *D. melanogaster*: host genotypes with impeded immune effectors have both a higher Hazard Ratio and a higher PLUD than host genotypes with a more efficient immune response. Using Catalase, an enzyme involved in protecting cells from damage by ROS, as a candidate gene for tolerance, we also confirmed that faster death, which we assumed to indicate lower tolerance, was associated with a lower PLUD. Although the link between

a lower pathogen load sustained before death and a default in tolerance seems logical, this probably needs further confirmation with other candidate genes for which we will start to understand the mechanisms by which they contribute to tolerance. As we already mentioned, potential tolerance genes in *D. melanogaster* have been identified using the PLUD as a tolerance proxy (*CrebA* and *Bombardier Lin et al., 2020*; *Troha et al., 2018*), or methods of which we discussed the limits, (*grainyhead, debris buster,* and *G9a Howick and Lazzaro, 2017*; *Merkling et al., 2015*). In fact, the role of *G9a* in controlling bacterial load is still debated (*Gupta and Vale, 2017*). We, therefore, here provide a clear indication that our method allows to detect defaults in tolerance, but complete validation will require that other candidate genes are tested.

One of the great benefit of theoretical studies is that they allow to explore notions which arose from empirical work. Concepts which seem well and clearly defined may often be underlain by complex and intermingled mechanisms. Our approach has helped to clarify the notions of disease tolerance and resistance, in particular by showing how experimental measurements can be misleading when interpreted as unambiguous measurements of one or the other mechanism. Many other mechanisms which determine important aspects of infection pose the same sort of difficulty. For example, the co-regulation of damage repair and immune response is now well accepted (*Martins et al., 2019*; *Pucillo and Vitale, 2020*) but remains difficult to study. Similarly, we do not know how the presence of different types of damage could affect disease outcome. We advocate that pursuing this combination of theoretical and empirical approaches is needed, first to predict how such intermingled mechanisms impact the infection outcomes, and second to design appropriate proxies and experimental methods to quantify their impact.

## Acknowledgements

We thank Jennifer Regan and Helen Alexander, and three anonymous reviewers for their comments on the manuscript. The project was supported by the French Laboratory of Excellence project 'TULIP' (ANR- 10-LABX-41 and ANR-11-IDEX-0002–02), and by the LIA BEEG-B (Laboratoire International Associé-Bioinformatics, Ecology, Evolution, Genomics and Behavior) (CNRS). PL was partly supported by the Darwin Trust of Edinburgh PhD studentship. XJ and NB were supported by NIH 5R01AI148541-05 and NSF IOS 1398682.

## Additional information

### Funding

| Funder | Grant reference number | Author |
|---|---|---|
| Agence Nationale de la Recherche | ANR- 10-LABX-41 | David Duneau |
| Agence Nationale de la Recherche | ANR-11-IDEX-0002-02 | Jean-Baptiste Ferdy |
| Agence Nationale de la Recherche | LIA BEEG-B | David Duneau |
| National Institutes of Health | 5R01AI148541-05 | Nicolas Buchon |
| National Science Foundation | IOS 1398682 | Nicolas Buchon |

The funders had no role in study design, data collection and interpretation, or the decision to submit the work for publication.

### Author contributions

David Duneau, Jean-Baptiste Ferdy, Conceptualization, Resources, Data curation, Formal analysis, Supervision, Validation, Investigation, Visualization, Methodology, Writing - original draft, Project administration, Writing – review and editing; Pierre DM Lafont, Data curation, Formal analysis, Methodology, Writing – review and editing; Christine Lauzeral, Formal analysis, Writing – review and editing;

Nathalie Parthuisot, Data curation, Validation, Methodology, Project administration, Writing – review and editing; Christian Faucher, Data curation, Validation, Methodology, Writing – review and editing; Xuerong Jin, Nicolas Buchon, Data curation, Validation, Writing – review and editing

**Author ORCIDs**
David Duneau https://orcid.org/0000-0002-8323-1511
Xuerong Jin https://orcid.org/0009-0000-2862-4075
Nicolas Buchon https://orcid.org/0000-0003-3636-8387
Jean-Baptiste Ferdy http://orcid.org/0000-0002-4020-1208

**Decision letter and Author response**
Decision letter https://doi.org/10.7554/eLife.104052.sa1
Author response https://doi.org/10.7554/eLife.104052.sa2

## Additional files

### Supplementary files
MDAR checklist

Source code 1. Rmarkdown file including codes and analyses associated with the experimental data.

### Data availability
Details of the shiny application can be found on Zenodo at *Lafont, 2024*; https://doi.org/10.5281/zenodo.13309654. Scripts and analyses are available on Zenodo at https://doi.org/10.5281/zenodo.14451399.

The following dataset was generated:

| Author(s) | Year | Dataset title | Dataset URL | Database and Identifier |
|---|---|---|---|---|
| Ferdy J-B, Duneau D | 2024 | A within-host infection model to explore tolerance and resistance -- experimental data and analysis | https://doi.org/10.5281/zenodo.14451399 | Zenodo, 10.5281/zenodo.14451399 |

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

## Appendix 1

### The equilibrium states of the infection

#### High constitutive immunity is required to cure infections, but may kill the host

Clearance is the equilibrium characterized by $x = 0$. Let $\tilde{y}_h$ and $\tilde{z}_h$ be the equilibrium levels of defense and damage for this equilibrium (with $\tilde{y}_h$ the homeostatic level of defense). From *Equation 1*, we know that $\tilde{z}_h = \tilde{y}_h \eta/\xi$ so that $\tilde{y}_h$ is a solution of $\gamma\, G(\tilde{y}_h, \eta/\xi\, \tilde{y}_h) = \varphi \tilde{y}_h$, which yields

$$-\frac{\gamma}{\varphi} + \tilde{y}_h + \tilde{y}_h^{k+1} + \psi\, \tilde{y}_h \left(\frac{\eta}{\xi}\tilde{y}_h\right)^l = 0. \tag{S1-1}$$

The left term of *Equation (S1-1)* is a strictly increasing function of $\tilde{y}_h$, which has exactly one positive root because $-\gamma/\varphi \leq 0$. From *Equation (S1-1)*, it is also easy to see that this root increases with $\gamma$.

A necessary condition for clearance to be stable is that $dx/dt(x, \tilde{y}_h, \tilde{z}_h)$ is negative when $x$ tends towards zero, which in turn requires that $\tilde{y}_h > 1/\delta$. From this and the previous result, we conclude that clearance is stable if and only if $\gamma > \gamma_c$ with

$$\begin{aligned}
\gamma_c &= \frac{\varphi}{\delta}\, \frac{1}{G\left(1/\delta, \eta/(\delta\xi)\right)}, \\
&= \frac{\varphi}{\delta}\left(1 + \left(\frac{1}{\delta}\right)^k + \psi\left(\frac{\eta}{\xi\delta}\right)^l\right).
\end{aligned} \tag{S1-2}$$

Increasing $\gamma$ raises the level of defense $y$, even in the absence of infection (as *Figure 3B* illustrates); but it also add to damage $z$, because immune defense have negative side effects when $\eta > 0$ (see *Figure 3C*). If constitutive defense are over-expressed, these negative effects could even kill the host in the absence of any infection. This condition is met when $\tilde{z}_h > z_d \Leftrightarrow \gamma > \gamma_s$ with

$$\gamma_s = \varphi \frac{\xi z_d}{\eta}\left(1 + \left(\frac{\xi z_d}{\eta}\right)^k + \psi z_d^l\right). \tag{S1-3}$$

### Bistability requires that damage hinders defense production

Let $\tilde{x}$, $\tilde{y}$, and $\tilde{z}$ be the state variables at equilibrium. Let $\bar{y}(x)$ and $\bar{z}(x)$ be the $y$ and $z$ values such that $\frac{dx}{dt}(x, \bar{y}(x), \bar{z}(x)) = \frac{dz}{dt}(x, \bar{y}(x), \bar{z}(x)) = 0$. From *Equation 1*, it is easily shown that

$$\begin{cases} \bar{y}(x) = & (1 - x)/\delta, \\ \bar{z}(x) = & \left(x(\omega - \eta/\delta) + \eta/\delta\right)/\xi, \end{cases} \tag{S1-4}$$

with $\tilde{y} = \bar{y}(\tilde{x})$ and $\tilde{z} = \bar{z}(\tilde{x})$. The equilibrium load $\tilde{x}$ can be found by solving $dy/dt(x, \bar{y}(x), \bar{z}(x)) = 0$, which can be written as

$$H(\tilde{x}) = F(\tilde{x})G\left(\bar{y}(\tilde{x}), \bar{z}(\tilde{x})\right) - \varphi \bar{y}(\tilde{x}) = 0. \tag{S1-5}$$

The system is, therefore, bistable only if $H$ has at least two roots in $[0, 1]$, which implies that $H'$ has at least one root in the same interval. We obtain

$$\begin{aligned}
H'(x) = {} & F'(x)G(\bar{y}(x), \bar{z}(x)) \\
& + F(x)\left(\dot{\bar{y}}(x)\frac{\partial G}{\partial y}(\bar{y}(x), \bar{z}(x)) + \dot{\bar{z}}(x)\frac{\partial G}{\partial z}(\bar{y}(x), \bar{z}(x))\right) - \varphi \dot{\bar{y}}(x)
\end{aligned} \tag{S1-6}$$

with $\dot{\bar{y}} = d\bar{y}/dx = -1/\delta$ and $\dot{\bar{z}} = d\bar{z}/dx = (\omega - \eta/\delta)/\xi = \theta/\xi$ is positive under the assumption that $u \geq v$, and all derivatives of $G$ are negative. If $\psi = 0$, $\partial G/\partial z = 0$ and all remaining terms in *Equation S1-6* are positive. $H'$, therefore, never cancels and $H$ must have a single root: the system cannot be bistable. If, conversely, $\psi > 0$ and if $\theta = \omega - \eta/\delta > 0$, then $\dot{\bar{z}}\partial G/\partial z < 0$ and $H$ can have more than one root. $\psi > 0$ and $\theta > 0$ are, therefore, necessary conditions for the system to be bistable.

We shall now investigate the robustness of this conclusion when three assumptions of our model are relaxed. We have assumed so far that

(i) Damage repair occurs at a constant rate $\xi$. Relaxing this assumption would make $\xi$ (or even $\omega$ and $\eta$) depend on $x$. The derivative $\dot{\bar{z}}(x)$ would then be modified,

(ii) Damage accumulation can only decrease defense production. But it is known that host cells attacked by pathogens can emit molecules which trigger immune defense. When this occurs, defense production may increase with damage, at least while damage stays moderate. This situation can be reproduced in our model by letting $F$ increase with $z$,

(iii) Defense are not consumed when they kill pathogens. Many other models (e.g. *Ellner et al., 2021*; *Souto-Maior et al., 2018*) rather include a term which reduces the amount of defense in proportion to their anti-pathogen activity (i.e. they include a negative term proportional to $\delta xy$).

If only the first assumption is relaxed, the derivative $\dot{\bar{z}}(x)$ is changed, as we said, but it will still cancel out in *Equation S1-6* when $\psi = 0$. Modulation of damage repair alone would, therefore, not invalidate our conclusion that bistability is impossible when $\psi = 0$. If $\psi > 0$ and $\theta > 0$, bistability will further require that that damage increases with load ($\dot{\bar{z}}(x) > 0$) which may be untrue early in the infection if $\xi$ increases very fast with load.

Relaxing the three assumptions altogether yields the following.

$$H'(x) = \left( \frac{\partial F}{\partial x}(x, \bar{z}(x)) + \dot{\bar{z}}(x)\frac{\partial F}{\partial z}(x, \bar{z}(x)) \right) G(\bar{y}(x), \bar{z}(x))$$
$$+ F(x, \bar{z}(x)) \left( \dot{\bar{y}}(x)\frac{\partial G}{\partial y}(\bar{y}(x), \bar{z}(x)) + \dot{\bar{z}}(x)\frac{\partial G}{\partial z}(\bar{y}(x), \bar{z}(x)) \right)$$
$$- \varphi\dot{\bar{y}}(x) - \delta_y\bar{y} - \delta_y x\dot{\bar{y}}. \tag{S1-7}$$

where $\delta_y$ is the parameter which determines the rate of defense consumption. We will assume in the following that $F$ increases with both $x$ and $z$. As in our first analysis, bistability will be possible if some of the terms in *Equation S1-7* are negative.

Let us first consider the case where damage increases with load ($\dot{\bar{z}} > 0$). The only negative terms in *Equation S1-7* are then $\dot{\bar{z}}(x)\frac{\partial G}{\partial z}(\bar{y}(x), \bar{z}(x))$ (as in *Equation S1-6*) which quantifies the negative impact of damage accumulation on defense production, and a term corresponding to the rate of defense consumption, $-\delta_y x$. Defense consumption is, therefore, clearly a mechanism which facilitates bistability. The activation of defense production upon damage detection will rather make it more difficult, because $\dot{\bar{z}}(x)\frac{\partial F}{\partial z}(x, \bar{z}(x))$ is positive.

In case damage decreases with load ($\dot{\bar{z}} < 0$) because repairing mechanisms are made more efficient when load increases, the negative terms in *Equation S1-7* are $\dot{\bar{z}}(x)\frac{\partial F}{\partial z}(x, \bar{z}(x))$ and, again, $-\delta_y x$. This time, the activation of defense production upon damage detection will facilitate bistability.

In summary, as long as damage increases with pathogen load, bistability will be made possible by any mechanism that reduces defense when the infection progresses, be it defense consumption or lowered defense production caused by the accumulation of damage.

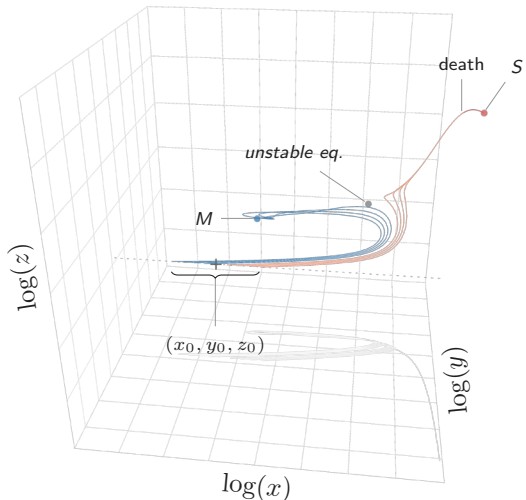

**Appendix 1—figure 1.** The dynamics of the system in a bistable configuration.

## Appendix 2

### Load and damage at equilibrium

#### Co-variations between load and damage at equilibrium

We took advantage of the implicit function theorem to gain information on how $\tilde{x}$ and $\tilde{z}$ covary when parameters are changed. Let us define $H_x$, $H_y$ and $H_z$ as

$$
\begin{cases}
H_x(x, y, z) = & dx/dt, \\
H_y(x, y, z) = & dy/dt, \\
H_z(x, y, z) = & dz/dt,
\end{cases}
\tag{S2-8}
$$

so that $H_x(\tilde{x}, \tilde{y}, \tilde{z}) = H_y(\tilde{x}, \tilde{y}, \tilde{z}) = H_z(\tilde{x}, \tilde{y}, \tilde{z}) = 0$. Assuming $\tilde{x} > 0$ and $\tilde{z} > 0$, $H_x$, $H_y$, and $H_z$ are all smooth. The Jacobian matrix J of our system is defined by the partial derivatives of functions $H_x$, $H_y$, and $H_z$ which, taken at the equilibrium point, yields

$$
J(\tilde{x}, \tilde{y}, \tilde{z}) =
\begin{pmatrix}
-\tilde{x} & -\delta\tilde{x} & 0 \\
F'(\tilde{x})G(\tilde{y}, \tilde{z}) & F(\tilde{x})\dfrac{\partial G}{\partial y}(\tilde{y}, \tilde{z}) - \varphi & F(\tilde{x})\dfrac{\partial G}{\partial z}(\tilde{y}, \tilde{z}) \\
\omega & \eta & -\xi
\end{pmatrix}
\tag{S2-9}
$$

of which determinant is

$$
|J(\tilde{x}, \tilde{y}, \tilde{z})| = -\xi\delta\tilde{x}H'(\tilde{x}).
\tag{S2-10}
$$

Assuming that $\tilde{x} > 0$, $|J(\tilde{x}, \tilde{y}, \tilde{z})|$ will be zero when $H'(\tilde{x}) = 0$. This condition is met only in the limit case which corresponds to the boundary of the bistable parameter region (see *Figure 4*). In all other situations, therefore, $|J(\tilde{x}, \tilde{y}, \tilde{z})| \neq 0$ and the implicit function theorem applies in the neighbourhood of $(\tilde{x}, \tilde{y}, \tilde{z})$. Each equilibrium variable $\tilde{x}$, $\tilde{y}$, and $\tilde{z}$ can then be written as a unique function of parameters which partial derivatives might be obtained by implicit differentiation.

#### The effect of parameters involved in response modulation

Let us first consider the effects of variation in $\gamma$, the reasoning applying in fact to any parameter which is not directly involved in $H_x$ and $H_z$. We have

$$
\begin{cases}
\dfrac{\partial H_x}{\partial \gamma} = 0 = & (1 - \tilde{x} - \delta\tilde{y})\dfrac{\partial \tilde{x}}{\partial \gamma} - \tilde{x}(\dfrac{\partial \tilde{x}}{\partial \gamma} + \delta\dfrac{\partial \tilde{y}}{\partial \gamma}), \\
\dfrac{\partial H_z}{\partial \gamma} = 0 = & \omega\dfrac{\partial \tilde{x}}{\partial \gamma} + \eta\dfrac{\partial \tilde{y}}{\partial \gamma} - \xi\dfrac{\partial \tilde{z}}{\partial \gamma}.
\end{cases}
\tag{S2-11}
$$

Let us now assume that $\tilde{x} > 0$, i.e., the infection has not been cleared at equilibrium. We obtain

$$
\begin{cases}
\dfrac{\partial \tilde{x}}{\partial \gamma} + \delta\dfrac{\partial \tilde{y}}{\partial \gamma} = 0, \\
\omega\dfrac{\partial \tilde{x}}{\partial \gamma} + \eta\dfrac{\partial \tilde{y}}{\partial \gamma} - \xi\dfrac{\partial \tilde{z}}{\partial \gamma} = 0,
\end{cases}
\tag{S2-12}
$$

which, by eliminating $\dfrac{\partial \tilde{y}}{\partial \gamma}$, yields

$$
\frac{\partial \tilde{z}}{\partial \gamma} = \frac{\left(\omega - \dfrac{\eta}{\delta}\right)}{\xi}\frac{\partial \tilde{x}}{\partial \gamma}.
\tag{S2-13}
$$

As long as $\theta = \omega - \dfrac{\eta}{\delta} > 0$, damage therefore increases with load. If ever $\theta < 0$, increasing defense (e.g. by increasing γ) will decrease load but increase damage. In these circumstances, the secondary effects associated to greater defense production exceeds the benefit of better control over pathogen proliferation.

## The effect of defense efficiency

The same type of calculation has been applied to other parameters, including $\delta$. This time, we obtain

$$\frac{\partial \tilde{x}}{\partial \delta} + \tilde{y} + \delta \frac{\partial \tilde{y}}{\partial \delta} = 0 \Rightarrow \frac{\partial \tilde{y}}{\partial \delta} = -\frac{1}{\delta}\frac{\partial \tilde{x}}{\partial \delta} - \frac{\tilde{y}}{\delta}. \tag{S2-14}$$

Compared to the effect of changes in γ (**Equation S2-12**, **Equation S2-14**) shows that changes in $\delta$ will not necessarily produced variations in $\tilde{y}$ which are proportional to that in $\tilde{x}$. Writing $\partial H_z/\partial \delta$ and eliminating $\frac{\partial \tilde{y}}{\partial \delta}$, we now obtain

$$\frac{\partial \tilde{z}}{\partial \delta} = \frac{\left(\omega - \frac{\eta}{\delta}\right)}{\xi}\frac{\partial \tilde{x}}{\partial \delta} - \frac{\tilde{y}}{\xi\delta}. \tag{S2-15}$$

As in **equation S2-13**, $\partial \tilde{x}/\partial \delta$ and $\partial \tilde{z}/\partial \delta$ might have opposite signs. But this time, the sign difference cannot be predicted from the sole sign of $\theta$. If $\theta > 0$, any decrease in $\tilde{x}$ following an increase in $\delta$ will indeed be accompanied by a decrease in $\tilde{z}$; but the same types of variation may occur when $\theta < 0$, provided that $\tilde{y}/\delta$ is high enough.

## The effect of parameters involved in damage dynamics

Following the same line of reasoning, we can describe how changes in $\omega$, $\eta$, and $\xi$ impact load and damage at equilibrium:

$$\frac{\partial \tilde{z}}{\partial \omega} = \frac{\left(\omega - \frac{\eta}{\delta}\right)}{\xi}\frac{\partial \tilde{x}}{\partial \omega} + \frac{\tilde{x}}{\xi}, \tag{S2-16}$$

$$\frac{\partial \tilde{z}}{\partial \eta} = \frac{\left(\omega - \frac{\eta}{\delta}\right)}{\xi}\frac{\partial \tilde{x}}{\partial \eta} + \frac{\tilde{y}}{\xi}, \tag{S2-17}$$

$$\frac{\partial \tilde{z}}{\partial \xi} = \frac{\left(\omega - \frac{\eta}{\delta}\right)}{\xi}\frac{\partial \tilde{x}}{\partial \xi} - \frac{\tilde{z}}{\xi}. \tag{S2-18}$$

In these three expressions, as in **Equation S2-15**, the change in $\tilde{z}$ cannot be determined solely from that of $\tilde{x}$ and from $\theta = \omega - \frac{\eta}{\delta}$. In fact, because we are here varying parameters which directly control damage dynamics, right hand terms in **Equations S2-16**, **Equations S2-17** and **Equations S2-18** all include a term which quantifies this direct effect ($\tilde{x}/\xi$, $\tilde{y}/\xi$, and $-\tilde{z}/\xi$, respectively). The other term rather measures damage variations which are indirect consequences of load variations.

## Variations in equilibrium load when parameters vary

We shall now write the partial derivatives of $H(\tilde{x})$ (viewed as a function of the parameters) to obtain derivatives of $\tilde{x}$ relative to each parameter. Starting with $\gamma$, we obtain

$$\begin{aligned}\frac{\partial H(\tilde{x})}{\partial \gamma} = &\left(1 + \frac{\partial \tilde{x}}{\partial \gamma}F'(\tilde{x})\right)G(\tilde{y}, \tilde{z})\\ &+ F(\tilde{x})\left(\frac{\partial \tilde{y}}{\partial \gamma}\frac{\partial G}{\partial y}(\tilde{y}, \tilde{z}) + \frac{\partial \tilde{z}}{\partial \gamma}\frac{\partial G}{\partial z}(\tilde{y}, \tilde{z})\right) - \varphi\frac{\partial \tilde{y}}{\partial \gamma} = 0\end{aligned}. \tag{S2-19}$$

The right hand term of **Equation S2-19** simplifies to $G(\tilde{y}, \tilde{z}) + \frac{\partial \tilde{x}}{\partial \gamma}H'(\tilde{x})$ which, therefore, yields

$$\frac{\partial \tilde{x}}{\partial \gamma} = -\frac{G(\tilde{y}, \tilde{z})}{H'(\tilde{x})}, \tag{S2-20}$$

with $H'$ the derivative of $H$ given by **Equation S1-6**. Assuming that $(\tilde{x}, \tilde{y}, \tilde{z})$ is a stable equilibrium, we know that $|J(\tilde{x}, \tilde{y}, \tilde{z})| < 0$. Using **Equation S2-10**, we can conclude that $H'(\tilde{x})$ is positive, and **Equation S2-20** therefore predicts that any increase in γ will induce a decrease in $\tilde{x}$.

Let us now consider the case of the unstable equilibria with $\tilde{x} > 0$, which exists only when the system is bistable. Using the Routh-Hurwitz criteria for 3×3 matrices, we can state that $(\tilde{x}, \tilde{y}, \tilde{z})$ is stable if the three conditions $\mathrm{tr}J(\tilde{x}, \tilde{y}, \tilde{z}) < 0$, $|J(\tilde{x}, \tilde{y}, \tilde{z})| < 0$ and $\mathrm{tr}J(\tilde{x}, \tilde{y}, \tilde{z})a < |J(\tilde{x}, \tilde{y}, \tilde{z})|$ are all satisfied, with $\mathrm{tr}J(\tilde{x}, \tilde{y}, \tilde{z})$ the trace and $a$ the sum of all principal 2×2 principal minors of $J(\tilde{x}, \tilde{y}, \tilde{z})$. From *Equation S2-9*, it is clear that the first condition is always met. It can also easily be demonstrated that, the system being bistable which requires that $\theta > 0$, $a > \tilde{x}\delta H'(\tilde{x})$. From this, it can be shown that the third condition of Routh-Hurwitz criteria is also always satisfied. We can, therefore, conclude that if $(\tilde{x}, \tilde{y}, \tilde{z})$ is unstable, $|J(\tilde{x}, \tilde{y}, \tilde{z})| > 0$ and, therefore, $H'(\tilde{x}) < 0$. This result yields the simple conclusion that, at the unstable equilibria of a bistable system, any increase in $\gamma$ will make $\tilde{x}$ increase.

The same sort of calculation can be performed for each parameter. *Appendix 2—table 1* summarizes these results, with the sign of derivatives being given for both stable and unstable equilibria. *Appendix 2—figure 1* gives a numerical illustration of these computations.

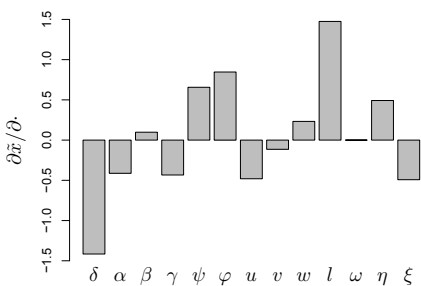

**Appendix 2—figure 1.** Elasticity analysis of equilibrium load. Parameters are here as in *Figure 2* and the derivatives of log equilibrium load $\tilde{x}$ relative to log parameters, obtained from , are computed for the low load equilibrium. For this equilibrium, $\tilde{x}$ is the load at which the chronic infection stabilizes and can, therefore, be considered as a stable Set-Point Pathogen Load (SPPL). The parameters with greatest influence on SPPL are first the efficiency of defense ($\delta$) and second the parameter which shapes the impact of damage on defense production ($l$).

**Appendix 2—table 1.** Derivatives of equilibrium load with respect to model parameters. For each parameter, the derivative is given together with its sign for stable and unstable equilibria.

| | $\partial\tilde{x}/\partial\cdot$ | st. eq. | unst. eq. |
|---|---|---|---|
| $\delta$ | $-\frac{\varphi\tilde{y}}{\delta}\left(k\tilde{y}^k + \frac{\psi\eta}{\xi}l\tilde{y}\tilde{z}^{l-1}\right) \times \frac{G(\tilde{y},\tilde{z})}{H'(\tilde{x})}$ $-\frac{\varphi}{\delta}\tilde{y} \times \frac{1}{H'(\tilde{x})}$ | $< 0$ | $> 0$ |
| $\alpha$ | $-\frac{\tilde{x}^u}{1+\beta\tilde{x}^v} \times \frac{G(\tilde{y},\tilde{z})}{H'(\tilde{x})}$ | $< 0$ | $> 0$ |
| $\beta$ | $\alpha\frac{\tilde{x}^u}{1+\beta\tilde{x}^v} \times \frac{G(\tilde{y},\tilde{z})}{H'(\tilde{x})}$ | $> 0$ | $< 0$ |
| $\gamma$ | $-\frac{G(\tilde{y},\tilde{z})}{H'(\tilde{x})}$ | $< 0$ | $> 0$ |
| $\psi$ | $\varphi\tilde{y}\tilde{z}^l \times \frac{G(\tilde{y},\tilde{z})}{H'(\tilde{x})}$ | $> 0$ | $< 0$ |
| $\varphi$ | $\tilde{y} \times \frac{1}{H'(\tilde{x})}$ | $> 0$ | $< 0$ |
| $u$ | $-\log(\tilde{x})(F(\tilde{x}-\gamma)) \times \frac{G(\tilde{y},\tilde{z})}{H'(\tilde{x})}$ | $> 0$ | $< 0$ |
| $v$ | $\log(\tilde{x})\frac{\alpha\beta\tilde{x}^{u+v}}{(1+\beta\tilde{x})^2} \times \frac{G(\tilde{y},\tilde{z})}{H'(\tilde{x})}$ | $< 0$ | $> 0$ |

*Appendix 2—table 1 Continued on next page*

*Appendix 2—table 1 Continued*

| | $\partial \tilde{x}/\partial \cdot$ | st. eq. | unst. eq. |
|---|---|---|---|
| $k$ | | $> 0$ if $\tilde{y} > 1$ | $< 0$ if $\tilde{y} > 1$ |
| | $\log(\tilde{y})\psi\varphi\tilde{y}^{k+1} \times \frac{G\tilde{y},\tilde{z}}{H'(\tilde{x})}$ | $> 0$ if $\tilde{z} > 1$ | $< 0$ if $\tilde{z} > 1$ |
| | $\frac{\varphi\psi l}{\xi}\tilde{x}\tilde{y}\tilde{z}^{l-1} \times \frac{G\tilde{y},\tilde{z}}{H'(\tilde{x})}$ | $> 0$ | $< 0$ |
| $\eta$ | $\frac{\varphi\psi l}{\xi}\tilde{y}^2\tilde{z}^{l-1} \times \frac{G\tilde{y},\tilde{z}}{H'(\tilde{x})}$ | $> 0$ | $< 0$ |
| $\xi$ | $-\frac{\varphi\psi l}{\xi}\tilde{y}\tilde{z}^l \times \frac{G\tilde{y},\tilde{z}}{H'(\tilde{x})}$ | $< 0$ | $> 0$ |

## Appendix 3

### The SPPL

Definition: Stable vs. transient SPPL

The SPPL can be defined as a sub-lethal load which stabilizes for a significant time during the infection. This corresponds well to what has been historically measured on HIV infections, but it remains vague as long as we do not specify what a 'significant time' is. In a mathematical model of infection, the SPPL could correspond to a stable equilibrium load, at which the infection stabilizes permanently. Its relation to parameters is then described by the analysis presented in .

But we shall see in the next section that the SPPL can also corresponds to non-equilibrium situations, where the progression of the infection slows down temporarily only. We must, therefore, distinguish stable from transient SPPL, which both characterize chronic infections but have, as we shall see, very different properties.

### Bistability can make the SPPL transient

A transient SPPL is a load which stabilizes for a long but finite period of time. This situation somehow resembles that of a chronic infection, but control is only transient and the infection will eventually enter a final phase where it reaches high loads and finally kills the host.

The progression of the infection slows down when a transient SPPL is approached because the trajectory evolves in the neighborhood of an equilibrium point; the fact that the infection eventually moves away from this point demonstrates that this equilibrium is unstable. Therefore, the SPPL cannot be transient unless an unstable equilibrium exists with $\tilde{x} > 0$, which in turns requires that the system is bistable.

In such conditions, the pathogen load typically first peaks, then decreases until it reaches a plateau (a local minimum) and finally increases again while the disease proceeds towards its final phase. We computed the SPPL is such situations as the load at the local minimum. Computing the duration of chronicity is not as straightforward. We considered that control starts when load peaks, because it corresponds to the time at which immunity manages to curb bacterial proliferation. We further considered that control ends when load, after having decreased to the SPPL, reaches again the peak load. We estimated the duration of chronicity as the time that elapses between these two moments.

In most conditions, increasing $x_0$, decreasing $y_0$ or increasing $z_0$ increases the SPPL and shortens chronicity (*Appendix 3—figure 1*). In summary anything that strengthens immunity does lower the SPPL and, most of the time, lengthen chronicity. Exceptions to this rule exist, though: *Appendix 3—figure 1* indeed illustrates cases (e.g. when $\psi = 0.5$) where the duration of chronicity decreases when $y_0$ is increased. It should be noted, however, that this unexpected result may originate from the way we have defined the duration of control. We chose to start the period of transient chronicity at the peak that precedes the SPPL but other methods could be imagined, and it may be that some of the results we have obtained on this quantity depend on how it is precisely defined.

When the SPPL is transient, both its value and the duration of control vary with initial conditions, because they both depend on how close the trajectory will get from the unstable equilibrium. When stable, the SPPL corresponds to a stable equilibrium. It, therefore, lasts forever (which in more realistic terms means that the host will die from something else than the infection) and does not depend on initial conditions. This provides a method to experimentally distinguish the two situations: if the SPPL increases with the dose of pathogen a host is infected by, it must be transient. This could be used as an indirect way to demonstrate first that the system constituted by the interacting host and pathogens is bistable, and second that damage caused by the infection hinders the production of immune defense (see Bistability requires that damage hinders defense production).

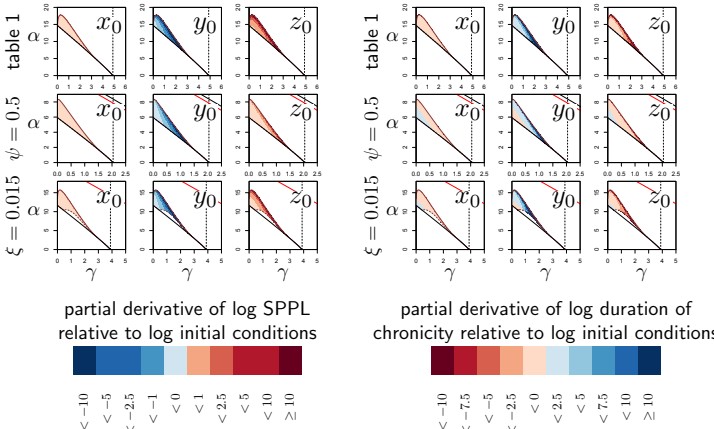

**Appendix 3—figure 1.** The influence of initial conditions on transient Set-Point Pathogen Load (SPPL). Left panel: each graph represents the partial derivative of log-SPPL relative to log-transformed inoculum size ($\log(x_0)$, first column), pre-infection level of defense ($\log(y_0)$, second column) and pre-infection level of damage ($\log(z_0)$, third column) with $x_0 = 10^{-6}$ and $y_0$ and $z_0$ set to the homeostatic state. In the first row, parameters are set as in *Figure 2*, while $\psi = 0.5$ in the second row, and $\xi = 0.015$ in the third. The vertical dashed line is the value $\gamma_c$ of γ, above which clearance is stable. Two black curves (with only one visible except when $\psi = 0.5$) delimits the parameter region in which the system is bistable. Within this region, the solid and dashed dark-red curves delimit a sub-region where the SPPL is transient in the conditions of the simulations. Below the red curve (which is visible only when $\psi = 0.5$ or $\xi = 0.015$) a high load infection always kills the host. The case $u = 0.75$, which we analyzed in *Figure 4*, is not presented here as the SPPL cannot be transient in this condition. The blue color indicates better control of the infection, i.e., decreased SPPL, while red means increased SPPL. In each case, anything that weakens the host (higher $x_0$, lower $y_0$ or higher $z_0$) lowers the SPPL. The analysis also indicates that sensitivity to variations in $y_0$ and $z_0$ increase with both α and γ. Right panel: This analysis follows the same logic but the partial derivatives of the log of the duration of chronicity are computed. The blue color indicates again better control, i.e., longer chronicity. In most cases, anything that weakens the host shortens control. But conditions exist where this is not true (e.g. in second and third rows when γ and α are low). Contrary to what is expected, chronicity may, therefore, last longer with increased SPPL when immunity is weak.

## A sensibility analysis of transient SPPL

We propose that anything that strengthens the host immunity decreases the SPPL and, in most situations, lengthens the duration of control. We explored this rule in more details by running a elasticity analysis of the SPPL and the duration of chronicity with $x_0 = 10^{-6}$ and $y_0$ and $z_0$ set at their homeostatic state (see *Appendix 3—figure 2* and *Appendix 3—figure 3*). For each parameter, we numerically computed the derivative of the SPPL using a five point approximation. In the case of parameter $u$, we computed right derivatives, so that the condition $u > v$ holds in our computation. Similarly, we did not compute derivatives with respect to parameter $v$ alone, but rather varied both $u$ and $v$ at the same time, keeping $u = v$, so that $F$ always increases with $x$. Overall, we found that increasing the production of defense (by increasing γ or α) and making those defense more efficient (by increasing δ) lower the SPPL. Increasing the rate at which damage is repaired (by increasing $\xi$) does the same (*Appendix 3—figure 2*).

What determines variation in the duration control is more complex. For example, when $\psi = 0.5$ parameter combinations exists for which the duration of control increases when immunity causes more damage (i.e. when $\eta$ increases, see *Appendix 3—figure 3*). This is reminiscent to the situation we discussed in the previous section, and again we must warn that different ways of calculating duration of control are possible and that this paradoxical result might come from the method we have chosen. It should be noted however that the parameter range over which the duration of control is not negatively related to the value of SPPL is extremely limited.

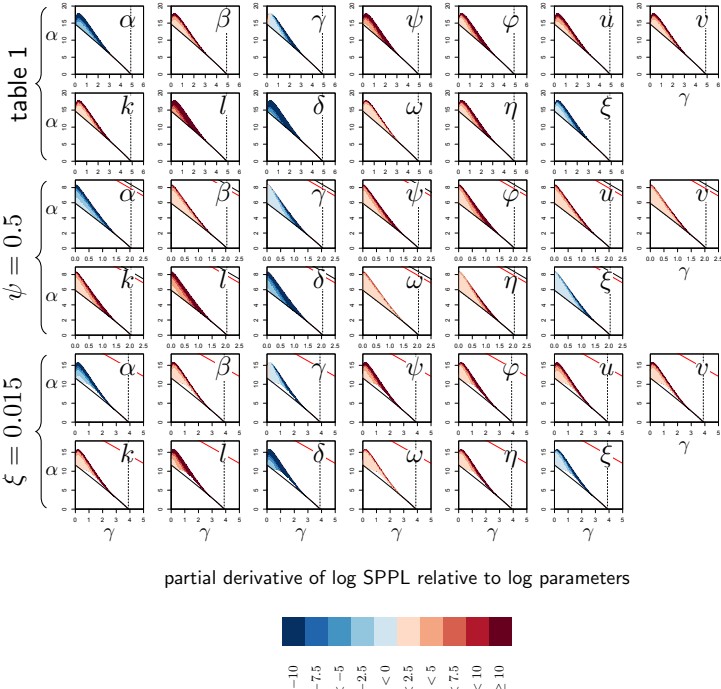

partial derivative of log SPPL relative to log parameters

**Appendix 3—figure 2.** Elasticity analysis of Set-Point Pathogen Load (SPPL) in transient chronicity according to parameters. Each graph represents the partial derivative of log SPPL with respect to one (log-transformed) parameter, as a function of α and γ. Blue color corresponds to negative derivatives (i.e. increasing the parameter lowers the SPPL) while red indicates positive derivatives (i.e. increasing the parameter increases the SPPL). The pairs of rows correspond to the three parameter sets studied in *Appendix 3—figure 1*. Derivatives with respect to $z_d$ have been omitted in this analysis as, by definition, $z_d$ has no influence on SPPL. The vertical dashed line is the value $\gamma_c$ of γ, above which clearance is stable. Two black curves (with only one visible except when $\psi = 0.5$) delimits the parameter region in which the system is bistable. Within this region, the solid and dashed dark-red curves delimit a sub-region where the SPPL is transient in the conditions of the simulations. Below the red curve (which is visible only when $\psi = 0.5$ or $\xi = 0.015$) a high load infection always kills the host.

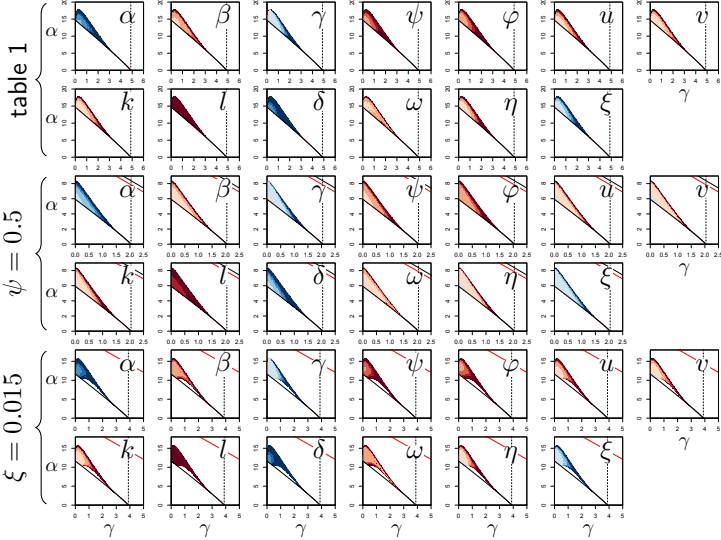

partial derivative of log duration of chronicity relative to log parameters

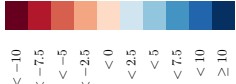

**Appendix 3—figure 3.** Elasticity analysis of the duration of transient chronicity according to parameters. Each graph represents the partial derivative of the log duration of chronicity with respect to one (log-transformed) parameter, as a function of α and γ. Blue color corresponds to positive derivatives (i.e. increasing the parameter lengthen chronicity) while red indicates negative derivatives (i.e. increasing the parameter shortens chronicity). The three pairs of rows correspond to the three parameter sets studied in *Appendix 3—figure 1*. Derivatives with respect to $z_d$ have been omitted in this analysis as, by definition, $z_d$ has no influence on Set-Point Pathogen Load (SPPL). The vertical dashed line is the value $\gamma_c$ of γ, above which clearance is stable. Two black curves (with only one visible except when $\psi = 0.5$) delimits the parameter region in which the system is bistable. Within this region, the solid and dashed dark-red curves delimit a sub-region where the SPPL is transient in the conditions of the simulations. Below the red curve (which is visible only when $\psi = 0.5$ or $\xi = 0.015$) a high load infection always kills the host.

## Appendix 4

### The PLUD

### Definition

The PLUD is the load $x_d$ at which a host dies. It is, therefore, the $x$ coordinate of the unique point where a trajectory first intersects the plane defined by $z = z_d$, with $z_d$ the amount of damage the host can sustain.

### Why is the PLUD almost independent from initial conditions?

In principle, the exact form of a trajectory depends on its starting point. The PLUD should, therefore, also vary with initial conditions. But trajectories tend to merge when approaching the high load equilibrium, as can be seen in *Appendix 1—figure 1*; if this occurs before $z$ has exceeded $z_d$, different infections will kill their host at similar loads, even though they do so at very different times.

This particular behavior comes from the fact that in infections which enter their final stage, just before the host dies, accumulating damage makes defense decrease. This implies that $\frac{dy}{dt} < 0$ and thus $0 < F(x)G(y,z)/y < \varphi$. Assuming that $\varphi$ is small and $y$ moderate, we can consider that $F(x)G(y,z)$ is also small so that $y$ is a slow variable compared to $x$. Because $x$ depends on $y$ but not on $z$, it should, therefore, rapidly get close to $\frac{dx}{dt} = 0$.

Said differently, if defense decay is a slow process, pathogens should rapidly reach the highest possible load allowed by the current amount of defense ($\frac{dx}{dt} \gtrsim 0 \Rightarrow x_t \lesssim 1 - \delta y_t$). The dynamics of $x$ will then be 'enslaved' by that of $y$ *Van Kampen, 1985* and load will increase slowly while $y$ gradually diminishes under the effect of accumulating damage. Because of this very particular dynamics, the way $x_t$ relates to $z_t$ will almost be the same for all infections that enter their final phase, which in the end renders the PLUD almost independent from the conditions that prevailed at the onset of the infection.

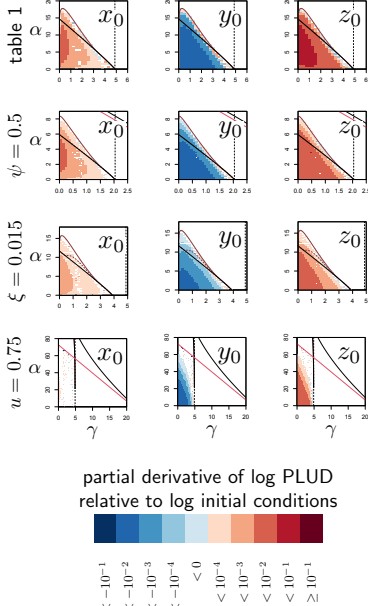

partial derivative of log PLUD
relative to log initial conditions

**Appendix 4—figure 1.** Elasticity analysis of the Pathogen Load Upon Death (PLUD) relative to initial conditions. Each graph represents the partial derivative of log-PLUD relative to log-transformed inoculum size ($\log(x_0)$, first column), pre-infection level of defense ($\log(y_0)$, second column) and pre-infection level of damage ($\log(z_0)$, third column) with $x_0 = 10^{-6}$ and $y_0$ and $z_0$ set to the homeostatic state. In the first row, parameters are set as in *Figure 2*, while $\psi = 0.5$ in the second row, $\xi = 0.015$ in the third and $u = 0.75$ in the fourth. The blue color indicates that the PLUD decreases, while red means that PLUD increases. In each case, anything that weakens the host (higher $x_0$, lower $y_0$ or higher $z_0$) increases the PLUD although the effect of dose ($x_0$) is really weak. In the particular case $u = 0.75$, the dependence on $x_0$ is so low that numerical derivatives of PLUD cannot be

*Appendix 4—figure 1 continued*

distinguished from zero. The vertical dashed line is the value $\gamma_c$ of $\gamma$, above which clearance is stable. Two black curves (with only one visible except when $\psi = 0.5$ or $u = 0.75$) delimits the parameter region in which the system is bistable. Within this region, the solid and dashed dark-red curves delimit a sub-region where the Set-Point Pathogen Load (SPPL) is transient in the conditions of the simulations (except when $u = 0.75$, where this never happens). Below the red curve (which is visible only when $\psi = 0.5$ or $u = 0.75$) a high load infection always kills the host.

*Figure 5C* gives a simple illustration of this result. ***Appendix 4—figure 1*** provides a more complete elasticity analysis, where the impact of variations in $x_0$, $y_0$ and $z_0$ are systematically explored following the same logic as in ***Appendix 3—figure 1***. The comparison with ***Appendix 3—figure 1***, where the SPPL is analyzed, demonstrates that variations in initial conditions have a much weaker impact on the PLUD than on the SPPL (derivatives relative to initial conditions span from −0.1 to 0.1 for the PLUD, see ***Appendix 4—figure 1***, while they span from −10 to 10 for the SPPL, and ***Appendix 3—figure 1***). The analysis also demonstrates that the PLUD should be more sensitive to initial conditions (and therefore more variable if these conditions are not fully controlled) when defenses are weakened (here when γ or α are low).

## A lower bound for the PLUD

In conditions where $z$ also evolves faster than $y$ during the final stage of the disease, trajectories leading to host's death should rapidly approach the intersect of the two isocline planes defined by $dx/dt = 0$ and $dz/dt = 0$. Building on this, we assume that points on the trajectory should then satisfy $y_t = \bar{y}(x_t)$ (i.e. $dx/dt = H_x(x, y) = 0$) and $x_t = \bar{x}(z_t) = (\xi z_t - \eta/\delta)/(\omega - \eta/\delta)$ (i.e. $dz/dt = H_z(x, y, z) = 0$). From there, we obtain that the host should die at a load close to $\bar{x}(z_d) = (\xi z_d - \eta/\delta)/(\omega - \eta/\delta)$. Note that if the high load equilibrium satisfies $\tilde{z} = z_d$, the PLUD is then exactly $x_d = \bar{x}(z_d)$.

We shall now demonstrate that this quantity is, in most situations, a lower bound of the PLUD. For this purpose, let us consider the case where $x$ and $z$ both always increase over time, until the host dies. Points on a trajectory leading to host death then necessarily satisfy $H_x(x, y) > 0$. As $\partial H_x/\partial y < 0$ and $H_x(x, \bar{y}(x)) = 0$, we can conclude that $y < \bar{y}(x)$ for all points of the trajectory. Because $z$ increases over time, points on the trajectory leading to death must also satisfy $H_z(x, y, z) > 0$. We know that $\partial H_z/\partial x > 0$ and $\partial H_z/\partial y > 0$. Recalling that $y < \bar{y}(x)$, we can therefore conclude that $x > \bar{x}(z)$ for all points on the trajectory. From this, we can conclude that the trajectory necessarily reaches $z = z_d$ for a value of $x$ which is greater than $\bar{x}(z_d)$:

$$x_d \geq (\xi z_d - \eta/\delta)/(\omega - \eta/\delta). \tag{S4-1}$$

Under some circumstances, though, $x$ and $z$ might decrease as $z$ approaches $z_d$. This happens for trajectories which spiral when approaching the high load equilibrium point (i.e. when the Jacobian matrix of the system has two complex eigenvalues for this equilibrium). Numerical simulations indicate, however, that $\bar{x}(z_d)$ is still a lower bound of the PLUD in these conditions.

## An elasticity analysis of the PLUD

The PLUD has been proposed as a proxy for tolerance to infection. But the lower bound we obtained in ***equation (S4-1)*** demonstrates that it varies with both tolerance and resistance parameters. $\delta$, for example, has an influence on the PLUD lower bound. The more complete elasticity analysis presented in ***Appendix 4—figure 2*** further demonstrates these mixed influences.

In most cases, the PLUD increases when control of proliferation is loosened (see derivatives according to $\alpha$, $\beta$, $\gamma$, $\psi$, $\varphi$, $u$, $v$, $k$, $l$ and $\delta$ in ***Appendix 4—figure 2***) and when damage production is lowered (see derivatives according to $\omega$, $\eta$ and $\xi$ in ***Appendix 4—figure 2***) or the host's tolerance to damage increased (see derivatives according to $z_d$ in ***Appendix 4—figure 2***). Exception to this rule do exist, though. An increase in $\eta$ can indeed either decrease the PLUD, as expected; but under certain parameter combinations it can also increase the PLUD. This happens when both $\alpha$ and $\gamma$ are low in ***Appendix 4—figure 2*** and probably comes from the fact that increasing $\eta$ strengthens the negative feedback of damage on defense production.

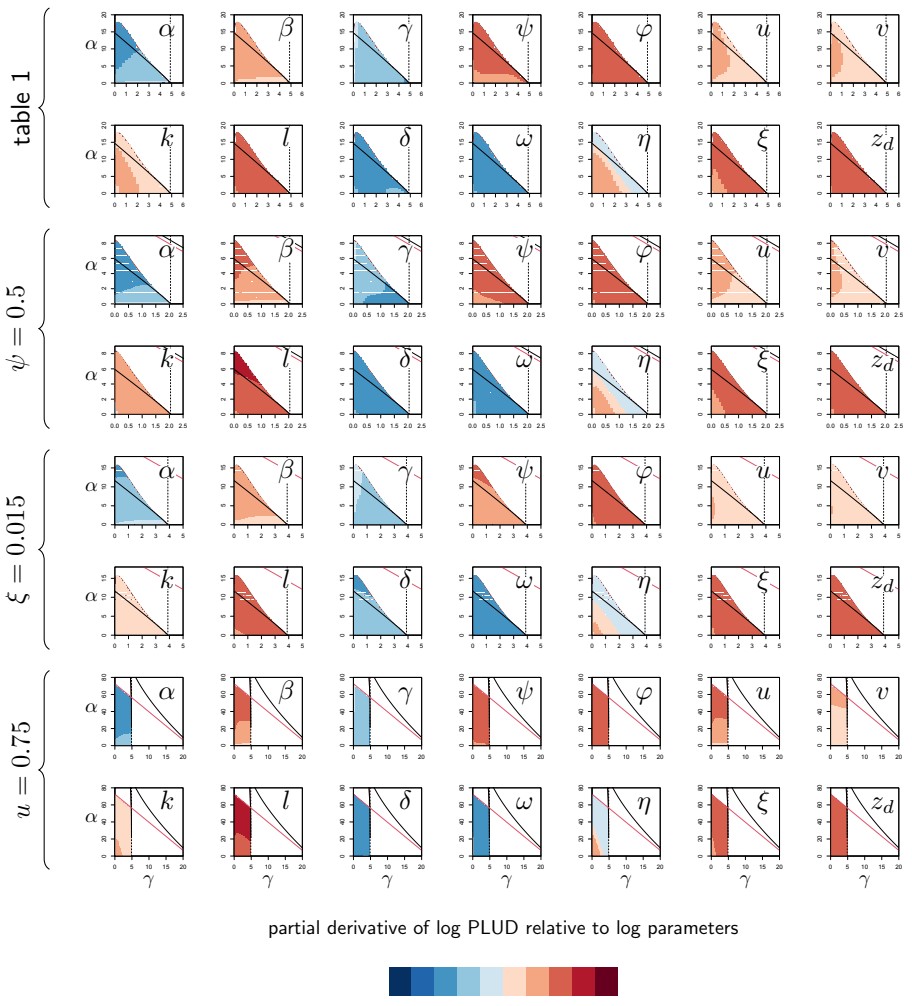

partial derivative of log PLUD relative to log parameters

**Appendix 4—figure 2.** Elasticity analysis of Pathogen Load Upon Death (PLUD). Each graph represents the partial derivative of PLUD with respect to one parameter, as a function of α and γ. Blue color corresponds to negative derivatives (i.e. increasing the parameter decreases the PLUD) while red indicate positive derivatives (i.e. increasing the parameter increases the PLUD). The black curves delimit the region of bistability, as in *Figure 4*; above the red line, the equilibrium $z$ is below $z_d$ even for the most severe infection. The four braces correspond to the four parameter sets studied in *Figure 4*. The vertical dashed line indicates $\gamma_c$. Black and red curves are as in *Appendix 4—figure 1*.

# Appendix 5

## The regression of LT on dose to quantify tolerance

*Louie et al., 2016* proposed that tolerance could be assessed by measuring how fast the time to death decreases when the dose of pathogen is increased. We tested this idea by computing the slope of the relation of LT on dose and then running an elasticity analysis on the slope of the regression line, similar to that we ran on SPPL, PLUD, and HR. More precisely, we estimated slopes by varying the average dose $x_0$ from $10^{-7}$ to $10^{-5}$ (with 50 values sampled) and computing corresponding LT. We then adjusted a regression line to the values obtained. We then computed numerical derivatives of the slope of the regression line respective to each of 14 parameters of the model, using a method similar to that we utilized for other quantities.

As any other parameter measured on WHD or survival curves, the slope of the regression of LT on dose depends on both tolerance and resistance parameters (see *Appendix 5—figure 1*). For example, an increase in tolerance to damage $z_d$ brings the slope closer to zero. This is in line with the expectations of *Louie et al., 2016*: the more tolerant the host, the less LT should vary with dose. But an increase in $\delta$, the efficiency of defense, a parameter which should clearly be categorized as controlling resistance rather than tolerance, yields a steeper regression line.

Finally, increasing $\xi$, the rate at which damage is repaired, also makes the regression line steeper. Being able to repair damage efficiently should probably be considered as part of what makes a host tolerant to the infection. It is, therefore, surprising that $\xi$ and $z_d$ have opposite influences on the slope of the regression of LT on dose. We found that the influence of $\xi$ is lowered when $\psi$ is decreased (see *Appendix 5—figure 1*, $\psi = 0.5$) which indicates that the unexpected effect of $\xi$ could stem from the negative impact of damage on the production of immune defense.

Overall, the slope of the regression of LT on dose cannot be considered as a simple proxy for tolerance. We found that it is sensitive to the complex interplay between damage accumulation and defense production, which makes it complicated to interpret.

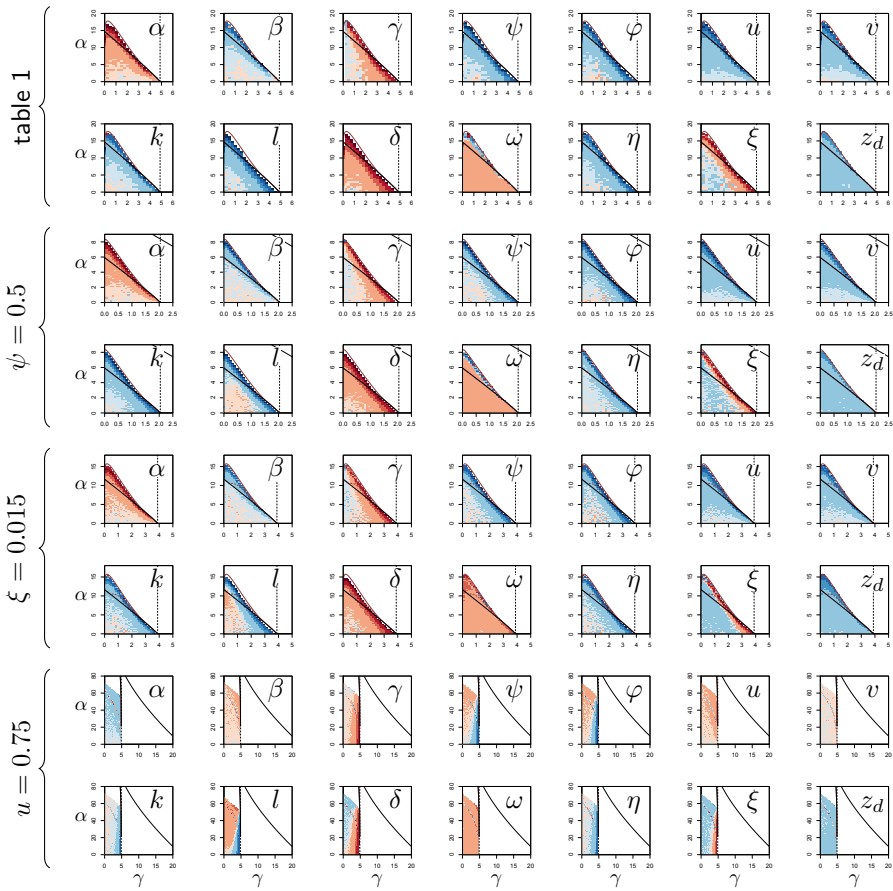

partial derivative of the slope of $\log(LT)$ to $\log(x_0)$, relative to log parameters

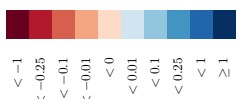

**Appendix 5—figure 1.** Elasticity analysis of the Slope of Lethal Time (LT) to dose. Each graph represents the partial derivative of hazard ratio with respect to one parameter, as a function of $\alpha$ and $\gamma$. Blue color corresponds to positive derivatives (i.e. increasing the variable brings the slope closer to zero) while red indicate negative derivatives. The black curves delimit the region of bistability, as in **Figure 4**; above the red line, the equilibrium $\tilde{z}$ never exceeds $z_d$ so that the infection cannot kill the host. Each of the four rows corresponds to one of the parameter sets studied in **Figure 4**. The vertical dashed line indicates $\gamma_c$. Black and red curves are as in **Appendix 4—figure 1**.

# Appendix 6

## Using HR to compare the lethality of two infections

### Definitions

Mortality measurements are by definition metrics which summarize survival curves. The median and average time to death (or Lethal Time, LT) are among those. One limitation of this type of statistics is that their interpretation might vary depending on the actual distribution of time to death. Another practical limitation is that their computation is problematic for mild pathogens which kill only a small proportion of the infected hosts. An alternative and popular approach is to use HR to compare one group of hosts (or pathogen) to another group used as a reference. The HR at time $t$ is the ratio of the death probabilities in the focal group to that in the reference group, computed for individuals that have survived up to time $t$. No assumptions are made here on the distribution of time to death, but HR are often assumed to be constant over time (as in the classic Cox Proportional Hazard model). Under this assumption, the comparison of two survival curves, therefore, yields one unique HR value.

We chose here to use HR as a convenient way to quantify relative death risk. Simulations proved that our model can produce survival curves which do not necessarily fulfill the proportional hazard hypothesis. We nevertheless found that deviations from this assumption were sufficiently mild to use HR in our analyses.

### An elasticity analysis of the HR

We then ran an elasticity analysis of HR relative to both initial conditions and parameter values, following the same logic than for the SPPL and the PLUD. In this analysis, we computed one survival curve for a reference situation and a second one with either one parameter or one initial condition changed. We then compared the two curves using a Cox Proportional Hazard model, which yielded an estimate of HR. The survival curve for the reference group was obtained by simulating infections for 100 individual hosts, with $y_0$ set at the homeostatic state. $\log(x_0)$ and $\log(z_0)$ were randomly drawn in Gaussian distributions with averages $\log(10^{-6})$ and $\log(z_h)$, respectively, and variance set to 0.25 in both cases (as in the simulations presented in *Figure 6C*).

As expected, increasing the dose (increasing $x_0$), weakening immune defense (lowering $y_0$), or damaging the host (increasing $z_0$) upon infection increase HR. We nevertheless found that alterations of $y_0$ and $z_0$ have much stronger impacts that changes in the dose (see *Appendix 6—figure 1*).

The analysis of how parameter changes influence mortality demonstrates that, as expected, any parameter variation which allows tighter control of proliferation reduces the hazard ratio (see derivatives according to α, $\beta$, $\gamma$, $\psi$, $\varphi$, $u$, $v$, $k$, $l$, and $\delta$ in *Appendix 6—figure 2*). As expected also, parameter variations which reduce the production of damage (see derivatives according to $\omega$, $\eta$, and $\xi$ in *Appendix 6—figure 2*) or increase the host's tolerance to damage (see derivatives according to $z_d$ in *Appendix 6—figure 2*) do reduce the HR. We never observed any exception to this rule.

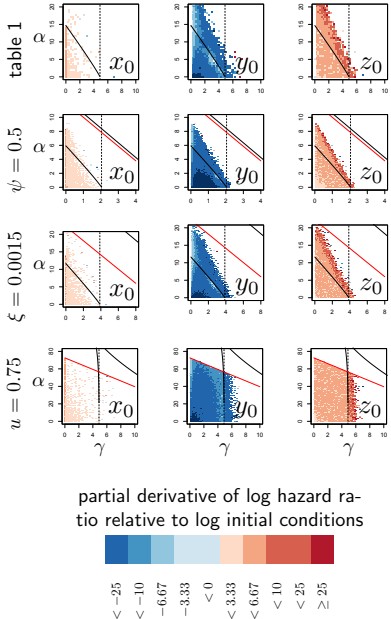

**Appendix 6—figure 1.** Elasticity analysis of Hazard Ratio, relative to initial conditions. Each graph represents the partial derivative of log(HR) with respect to the log-transformed initial state of one variable, as a function of α and γ. Blue color corresponds to negative derivatives (i.e. increasing the variable decreases hazard ratio, HR) while red indicate positive derivatives. The black curves delimit the region of bistability, as in **Figure 4**; above the red line, the equilibrium $\tilde{z}$ never exceeds $z_d$ so that the infection cannot kill the host. Each of the four row corresponds to one of the parameter sets studied in **Figure 4**. The vertical dashed line indicates $\gamma_c$. Black and red curves are as in **Appendix 4—figure 1**.

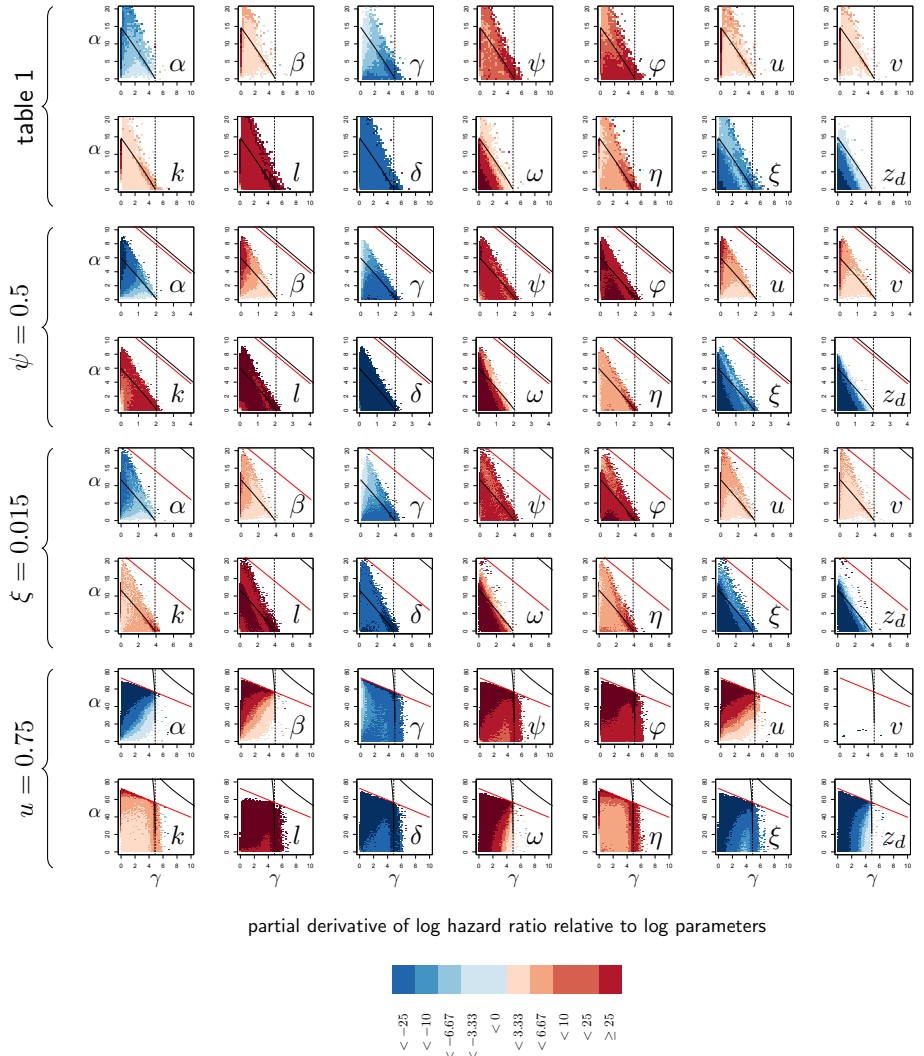

partial derivative of log hazard ratio relative to log parameters

**Appendix 6—figure 2.** Elasticity analysis of Hazard Ratio, relative to parameters. Each graph represents the partial derivative of hazard ratio with respect to one parameter, as a function of $\alpha$ and $\gamma$. Blue color corresponds to negative derivatives (i.e. increasing the variable decreases HR) while red indicate positive derivatives. The black curves delimit the region of bistability, as in *Figure 4*; above the red line, the equilibrium $\tilde{z}$ never exceeds $z_d$ so that the infection cannot kill the host. Each of the four rows corresponds to one of the parameter sets studied in *Figure 4*. The vertical dashed line indicates $\gamma_c$. Black and red curves are as in *Appendix 4—figure 1*.

## Appendix 7

### Measuring the correlation between HR and PLUD: A case test using *Drosophila melanogaster*

Experimental methods

*D. melanogaster* were reared until adulthood on glucose-yeast medium (82 g/L yeast, 82 g/L glucose, 10 g/L agar, propionic acid 3 g/L, Tegosept 3 g/L) in large fly glass bottles. At day 2 after hatching, adults were isolated in groups of approximately 50 males and 50 females per 900 mL plastic box. Husbandry and experiments were conducted at 24 °C (±1 °C) with a 12 hr:12 hr light:dark cycle. We used four *D. melanogaster* lines sampled from the *Drosophila* Genetic Reference Panel (*Mackay et al., 2012*, DGRP: RAL-630, RAL-818, RAL-559 and RAL-584), which all produce a fully functional Diptericin A (*Unckless et al., 2016*), i.e. with a serine residue 69 in the mature peptide, see and five genotypes (*BomΔ55C*, A, B, AB, Drs[R1], and Dpt[sk1], see text for further details) described in *Hanson et al., 2019* which were kindly provided by Bruno Lemaitre's laboratory from the Swiss Federal Institute of Technology in Lausanne. We also used RNAi experiments to study the impact of a defect of resistance (DptA-RNAi) and tolerance (Cat-RNAi BDSC#24621) on the BLUD. Both RNAi were crossed with Actin5C-Gal4 (BDSC#4414) ubiquitous driver. As both are part of the TRiP RNAi panel, we used the recommended control background lines attP40 (BDSC#36304) and attP2 (BDSC#36303) crossed with the same Actin driver.

All experiments were conducted with mated males 5–8 d post-eclosion. Males were infected with a strain of *Providencia rettgeri* isolated from wild-caught *D. melanogaster* strain Dmel *Galac and Lazzaro, 2011*; *Juneja and Lazzaro, 2009* which is moderately virulent (i.e. kills 20–50% of the individuals over 3 d when hosts are infected with approximately 1500 bacteria). Liquid cultures of bacterium were grown to saturation overnight at 37 °C. Saturation cultures were pelleted, then resuspended and diluted in phosphate buffered saline (PBS, pH 7.4) to an optical density (OD) of 0.1 (600 nm wavelength). We injected 23 nL of bacterial suspension into each fly abdomen using a Nanoject II (Drummond). For each experiments, we confirmed that this injection corresponded to an inoculum of approximately 1500 viable bacteria per fly by injecting the suspension in PBS and estimating the number of viable bacteria as described below. Flies were anesthetized with CO2 during the injection procedure, and then were observed shortly after injection to confirm recovery from manipulations. We verified that flies injected with PBS did experience minimal to no mortality during the experiments. There is virtually no intrinsic mortality over the few days the experiments lasted when flies are not infected. After injection, flies were kept together in 900 mL plastic boxes, labelled with information about treatment and genotype, with ad libitum access to food.

To monitor PLUD, dead flies were removed every 30 min from 12 to 24 hr after injection. Flies which just died were individually homogenized in 250 µL of sterile PBS with a TissueLyser homogenizer (Qiagen) with 2 mm glass beads. The homogenate was diluted in PBS (see, for details on the dilution protocol) and a 5 µL drop of each dilution was deposited onto LB agar with Erythromycine added (5 µg/mL final concentration), using a Viaflo multichanel pipette (Integra). Number of replicates for each drops are indicated in and defined to maximize accuracy of the estimation. Plates were incubated overnight at source data and defined to maximize accuracy of the estimation. Plates were incubated overnight at 37° C and scanned so that bacterial colonies could be subsequently counted. The number of viable bacteria was estimated for each fly as the intercept of a Poisson GLM with an offset corresponding to the log-transformed dilution factor. An average estimate was obtained for each line by adjusting a Negative-Binomial GLM with 'Line' as a main effect and 'Replicate' as random effect using the R package 'spaMM' (*Rousset and Ferdy, 2014*). We realized that CFU counts could be under-estimated when density is high and many CFUs overlap. We, therefore, decided to ignore drops with more than 50 counted CFUs in our analyses. Count of CFU were performed without knowing the information (i.e. genotype or treatment) about the individuals. As the experiment for studying the BLUD in Catalase mutant has been performed in the Buchon Lab at Cornell University, CFU per flies were estimated differently. Individual flies were homogenized in 500 µl of sterile PBS with an HT homogenizer (OPS Diagnostics) at each timepoint post-injection. The homogenate was then diluted to 1:100 or 1:1000 in PBS to ensure that plate counts remained within the limits of resolution of the plating system. We plated 70 µl onto LB agar using a WASP II Autoplate spiral plater (Microbiology International).

We monitored survival first over the same period than the PLUD, and subsequently 24 hr and 48 h post-injection. Individuals alive after 48 d were recorded as censored observations. Host survival differences were analyzed using a Cox's proportional hazards mixed model, with 'Line' as main effect and 'Replicate' as random effect using the R package 'coxme' (*Therneau, 2020*). We did not included non-infected flies to correct for intrinsic mortality, as the experiments are lasting only a few days (most infection-related deaths occur within 2 d), and those young flies would not die within this timeframe.

## Analyzing PLUDs when some individuals do not die from the infection

In any experiment of the type we described above, some deaths may occur for reasons other than infection. Measuring the PLUD on individuals that were not killed by the infection has of course no biological relevance. It may even cause spurious results if ever the risk of such death differs between treatments: treatments where individual tend to die before being killed by the infection should have lower PLUD, which may lead to flawed conclusions.

One way to limit this risk would be to remove extremely low PLUD values from experimental data. This can be done by fixing a somewhat arbitrary PLUD threshold; it could also be achieved more rigorously by using statistical tests specifically designed to detect outliers. In the following simulations, we evaluated the efficacy of this latter approach by using the Rosner test, which assumes that samples of size $n$ contain $n - k$ data points which are drawn from a Gaussian distribution while the $k$ remaining points are outliers.

For this purpose, we first used our model to simulate PLUD data. We then assumed that for each individual time to death is either determined by how fast the infection proceeds, as predicted by our model, or rather randomly drawn from a Weibull distribution, when death is caused by events that are independent from the infection. If the model predicts that death should occur later than this randomly drawn time value, we considered that the individual has not died from the infection. In such cases, we used our model to simulate the pathogen load at the randomly drawn death time.

Using a Weibull distribution allows us to consider cases where the background mortality rate does vary over time, during the course of experiment. We considered here a shape parameter of 1 (constant background mortality, yielding exponential distribution), 3 or 6 (meaning that mortality increases over time). Having fixed the shape parameter, we adjusted the scale parameter of the Weibull distribution so that we could control the proportion of individuals dying from causes other than infection. We varied this proportion from 2.5% to 25%.

We adjusted model parameters so that the distribution of PLUD values approximately matches the one we observed in our experiment (see *Appendix 7—figure 2*). We also considered that part of the variance in PLUD originates from uncertainties in load estimations. To reproduce this, we considered that actual load is $10^7$ and explicitly simulated the procedure of plating samples. More specifically, we used the dilution scheme presented above to predict the average number of CFU in a 5 µL droplet, and randomly drawn actual CFU numbers from a Poisson distribution. We then considered that only CFU numbers below 50 are countable (as in our experiments) and estimated load using a right-truncated Poisson model.

We found that the Rosner test adequately detects most of the individuals which died before being killed by the infection (*Appendix 7—figure 2*). Nevertheless, approximately 10% of them were not detected, whatever Weibull distribution we used. We evaluated how this lack of power could impact subsequent PLUD comparisons, by running simulations where a Wilcoxon rank test is used to compare two 50 individuals samples, one where all individuals died from the infection and another where some individuals died from other causes. All infections having the same dynamics in these simulations, the PLUD should not not differ among samples. But *Appendix 7—figure 3* demonstrates that the presence of individuals who died before being killed by the infection creates spurious differences. As expected, increasing the proportion of such individuals worsen the problem. Conversely, increasing the shape parameter of the Weibull distribution tend to reduce the difference between samples because individuals tend to die late, even when not killed by the infection, at a load which, therefore, approaches the PLUD.

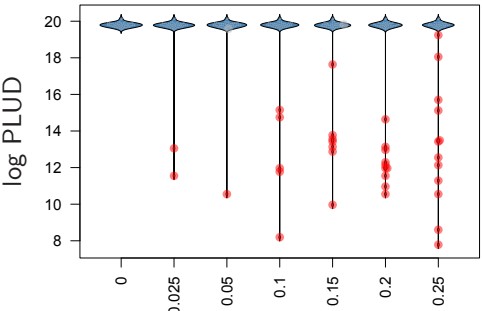

**Appendix 7—figure 1.** The distribution of log Pathogen Load Upon Death (PLUD) as a function of the proportion of individuals which did not die from the infection in a chosen simulated sample of 50 individuals. Background mortality was assumed to be constant so that death time in the absence of infection are exponentially distributed. An individual was considered as killed by the infection if the time to death predicted by our model was lower than the time randomly drawn in an exponential distribution. The average of the exponential distribution was adjusted sa that we could control the proportion of individuals which did not die from the infection (represented by large dots). Red dots indicate outliers as detected by a Rosner test. PLUD was simulated with $\delta = 0.125$, $\varphi = 0.015$, $\gamma = 3$, $\alpha = 10$, $\beta = 1$, $u = 0.1$, $v = 0.1$, $k = 0.5$, $\psi = 2$, $l = 1.5$, $\omega = 10$, $\eta = 0.01$, $\xi = 0.01$, $z_d = 100$ so that the simulated distribution mimics that of experimental results.

The online version of this article includes the following source data for appendix 7—figure 1:

**Appendix 7—figure 1—source data 1.** Protocol of dilution to estimate the Pathogen Load Upon Death (PLUD).

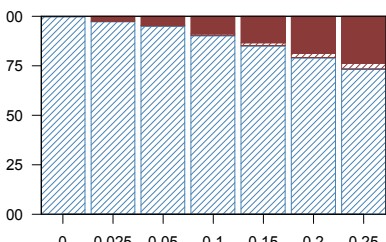

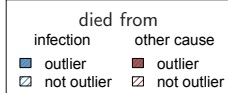

**Appendix 7—figure 2.** Proportion of detected outliers using a Rosner test, with background mortality producing exponentially distributed death times (i.e. Weibull distribution with shape parameter equals one). The proportion of Pathogen Load Upon Death (PLUD) values detected as outliers adequately reflects the proportion of individuals which died from causes other than infection. We simulated here 100 samples of 50 individual PLUD values and ran a Rosner test on each sample, using a five percent threshold p-value.

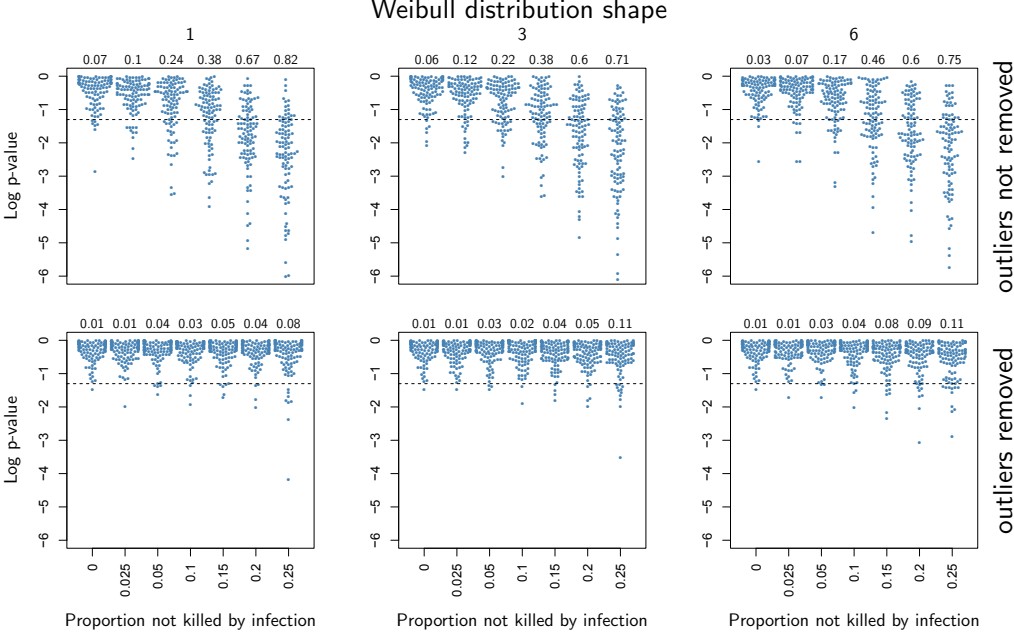

**Appendix 7—figure 3.** p-values of a Wilcoxon rank test where two 50 individuals samples were compared, one where all individuals died from the infection and another where some individuals died from other causes. Each column corresponds to a different shape of Weibull distribution, 1 being the case where natural death rate is constant over time, 3 and 6 being being cases where natural death rate increases with age. Numbers given in the top margin of the graphs indicate the proportion of tests (out of 100) which yielded a significant difference, using a 0.05 significance level. In the first row, the presence of individuals which did not die from the infection creates significant Pathogen Load Upon Death (PLUD) difference. In the second row, a Rosner test is applied to detect outliers before the Wilcoxon test is run. The proportion of test yielding significant difference is markedly reduced in this situation.

The second row of *Appendix 7—figure 3* demonstrates that the problem is almost suppressed when outliers detected by a Rosner test are removed before the Wilcoxon test is run. Spurious significant differences can still occur with a 10% probability, but only when one sample contains more than 10% of the individuals which do not die from the infection. Using adequate controls in the experiment should allow to detect such situations: indeed, only individuals from the second sample should die in the absence of infection.

