## [Editor Report]

Duneau et al. provide an extensive effort to model parameters of infection, an important topic in disease management. The theoretical findings of this study are important and will be of interest to mathematical biologists to model infection. The empirical data support the arguments, but may be incomplete, and more could be done in experiment design to shore up the robustness of these findings. This study helps us to better understand the complex course of infection.

---

## [Decision Letter]

**Decision letter after peer review:**

[Editors’ note: the authors submitted for reconsideration following the decision after peer review. What follows is the decision letter after the first round of review.]

Thank you for submitting the paper "A within-host infection model to explore tolerance and resistance" for consideration at *eLife*. As you can see from the three reviewers and despite their interest on the manuscript, there are substantial issues needed to improve the manuscript both at the theoretical and experimental level. Note that *eLife* is opened to the idea of a new submission if the authors can address the points raised by reviewers. The reviewers did share that this article may be more suitable in its current format for a more specialized journal;

*Reviewer #1 (Recommendations for the authors):*

The authors provide an impressive, detailed, and nuanced discussion of parameters contributing to outcome of infection. This statistical model considers a variety of theoretical contributors to pathogen growth, immune defense, and damage amelioration. The topic at hand is of general interest in modelling infectious disease processes, and to understand what parameters contribute most to readouts of disease progression. The framework the authors set out is logically consistent, and there is a clear effort to be thorough in constructing their statistical framework.

However I am unconvinced that the experimental data support the conclusions of the model (Figure 7-8). As the model is entirely theory, it is essential that the experimental design and data used to validate the model robustly support its predictions. The authors may be able to address these concerns in revisions, and increase confidence in their conclusions.

The authors use different experimental treatments (ex. wounding prior to infection) and three key experimental readouts to inform on the interactions of their model parameters (Survival, SPPL, PLUD). However I am unconvinced by the experimental data presented to validate their model. This unfortunately derails the paper right at the critical stage, and deflates confidence in the conclusions.

Lines 484-497 (Figure 7) – The key question is if mortality increase caused by wounding is loss of resistance or tolerance. The metric used to test this is PLUD. The authors do find a general higher PLUD in wounded flies, though only in 2/4 genotypes. I am concerned this result comes from survivor bias effects.

As SPPL is variable across individuals, any flies surviving the initial phase of infection will suppress pathogen growth to a transient SPPL. However flies dying rapidly from the infection have uninterrupted pathogen growth towards PLUD. As seen in Figure 7A, many No Wound flies survive this initial infection, but have a steady mild mortality rate at later time points. On the other hand, Wounded flies die almost 100%, and survival curves plateau.

Therefore Wounded flies mostly die from unchecked bacterial growth. However No Wound flies die from either pathogen growth (majority) or at later time points, despite having suppressed the initial infection (minority). I suspect this is caused by a factor the authors ignore: Zd of their model is not fixed. Damage from the initial infection and autotoxicity from the immune response may not kill hosts in the first 24h, but can ultimately be the cause of death (ex. organ failure). In beetles, it is known that autotoxic effects of the immune response reduce lifespan associated with malpighian tubule dysfunction (doi:10.1098/rspb.2017.0125). Even a fly that successfully suppresses pathogens can die from failing organ systems without needing recurrent pathogen growth, which may not even be possible when the blood is antimicrobial.

The significance of Figure 7B is driven almost entirely by low-PLUD outliers present mostly in No Wound flies. This suggests PLUD differences are driven by loss of tolerance in No Wound survivors, and not loss of resistance in dying Wounded flies. This is the exact opposite conclusion of the authors. If the authors have time of death associated with their data points, this could confirm or counter my concern regarding low-PLUD outliers.

Lines 498-520 (Figure 8) – The authors use Bom mutants as a control, saying they have little effect on gram-negative bacteria (Hanson et al). But checking Hanson et al., they do not infect Bom mutants with gram-negative bacteria. In fact, Duneau et al. (2017,BMC BIOL) reports that the Toll pathway mediates a sexual dimorphism in response to P. rettgeri infection. Duneau et al. (2017,*eLife*) also showed PLUD does not vary across Imd, Toll, Phagocytosis, or Melanization mutants? These studies from one of the authors are in direct contradiction to the logic of this experiment.

Also Bomanin function is not known, and there is no evidence they are directly antimicrobial based on existing studies. Instead, the authors cite Lin et al. in their discussion (Lines 653-657), which is a study implicating Bomanin effects on tolerance through Bombardier, including a mortality associated with immune activation by heat-killed bacteria. It is likely Bomanins affect tolerance, so use of Bom mutants as a baseline for comparison is wholly inappropriate. Why did the authors not use a wild type control here?

Additionally, A group died more than Bomanin, so their claim that Defensin does not affect survival seems untrue in their conditions. The difference in PLUD in A, B, and AB is less convincing. Defensin is also associated with clearance of aberrant cells (doi:10.7554/*eLife*.45061), so even assuming a mild difference is true, it can again be due to tolerance effects. Why did the authors not compare Diptericins specifically in their question, given previous studies on the Diptericin and P. rettgeri? This is not a strict request for additional experiments, but I would be more convinced by use of AMP genotypes that might have specific activity against P. rettgeri, and genotypes with related but less important activity. A wild type control is necessary.

– It is not clear why the authors do not modulate JAK-STAT stress responses as a test of resistance vs tolerance (Lines 149-155 are not convincing).

– What is the status of the Diptericin locus in RAL-818, RAL-630, etc… Given Unckless et al. (2016)?

– In the discussion the authors comment on SPPL nicely. I am curious if they have considered the scenario where bacteria in SPPL stage are in a hibernation state. This is common in uropathogenic *E. coli*, which reside in host epithelial cells and cause recurrent outbreaks. But those *E. coli* are not constantly causing damage while in hibernation. Does the model assume a constant rate of damage for bacteria in SPPL phase?

– The authors should include a README file explaining the columns in their supplemental data files

*Reviewer #2 (Recommendations for the authors):*

Lafont et al. developed a novel theoretical model that describes how host resistance and tolerance affect within-host pathogen dynamics. Their model focuses on recently documented with-host bacterial dynamics in *Drosophila*. Specifically, it was previously shown that in some cases the same inoculation dose can lead to two distinct outcomes: (1) the host successfully controls pathogen growth, which leads to an apparently stable set-point pathogen load (SPPL), and (2) the pathogen growth out of control reaching very high levels termed pathogen load upon death (PLUD), which causes rapid host death. The developed model can successfully reproduce this type of branching process. In addition to other existing models, the authors are able to reproduce the empirically observed pattern that the SPPL can increase with an increasing inoculum dose. Two findings of the model analysis are particularly relevant for empiricists studying resistance and tolerance. First, the results contradict the previous belief that only tolerance affects the PLUD. Instead, the authors now demonstrate that also resistance can affect the PLUD. Second, the authors raise considerable doubt on the validity of a commonly applied method for quantifying tolerance, which is based on measuring the reaction norm of host fitness in relation to pathogen load (measured either by the inoculum dose or by pathogen load at one point during the infection). Specifically, the authors show that this reaction norm can be more strongly influenced by resistance than by tolerance. To overcome these problems, the authors propose a novel method to infer variation in resistance and tolerance, which is based on integrating measures of the PLUD and host survival. Finally, the authors validated parts of their model with experimental infection studies on *Drosophila melanogaster*. These studies demonstrate the predicted effect that the PLUD increases and survival decrease due to wounding, and due to the knockout of important resistance genes. The agreement between the predicted and observed effects is interpreted by the authors as a confirmation of the general validity of their model and the proposed method.

Taken together, the authors present very interesting theoretical and empirical results with potentially far-reaching consequences for understanding and measuring how hosts respond to pathogen infections. Nevertheless, there are some limitations of this study, which I think were not sufficiently considered when interpreting the results.

1. The theoretical and empirical work is biased towards resistance. The pronounced differences in the way both host strategies were modelled and empirically investigated could strongly limit the validity and generality of the model and the proposed method. As the authors explain, due to the lack of known tolerance genes in *Drosophila*, they were not able to empirically test tolerance specific predictions of their model. The authors acknowledge this limitation, but do not see it as a major problem. However, it appears doubtful that the empirically measured resistance effects are sufficient for concluding that also the tolerance effects are correctly predicted by the model. In addition to these limitations on the empirical side, the theoretical side contains the limitation that tolerance was modelled in a much more simplistic way compared to resistance. In the model, host resistance is characterized by three major features: it depends on host condition (i.e. the level of damage), it is costly (because it generates damage), and it is modulated by pathogen load. In contrast, the implementation of tolerance lacks all of these features: it is cost-free, it is independent of host condition and it is independent of pathogen load or amount of damage. The authors acknowledge that tolerance mechanisms are known to be modulated during an infection, but because related details are still unknown they chose to model tolerance in a very simplistic way. This choice is certainly a reasonable first step. Nevertheless, it leaves the possibility that more complex tolerance effects could strongly affect the dynamics predicted by the model. Especially in combination with the lack of empirical data on tolerance, this makes it challenging to fully assess the validity and generality of the model and the proposed method.

2. Effects on reaction norms are not very well explored. The authors have shown that the relationships between inoculation dose and host survival contradicts assumptions made in empirical studies (Figure 5). However, the relationship between SPPL and host survival is not explicitly shown and it is, therefore, hard to assess whether the problem also applies to this relationship. A more detailed analysis would be particularly informative for empirical studies that measure pathogen load during the infection. Furthermore, conducting empirical tests of the predicted effects is an important task that remains to be done before concluding that the reaction norm approach is generally flawed.

3. Host background mortality is not considered. The authors developed a deterministic model in which host death can only occur due to an infection. Host background mortality due to other causes is not considered. This is not necessarily a problem for the theoretical analyses. However, to avoid biased results, empirical applications of the proposed method should control for potential variation in background mortality among different host lines.

4. Very wide and skewed empirical PLUD distributions. In some cases, the empirically measured PLUD distributions are very skewed and very wide with a difference of up to five orders of magnitude between the smallest and the largest values (Figure 7B). It is not clear how this enormous variation arose and whether this might indicate a mismatch to the dynamics predicted by the model.

5. Differences between observed and predicted branching dynamics. There seems to be a mismatch between the empirically observed temporal pattern of branching (Figure 1) and the corresponding model dynamics (Figure 3D). In the model, branching starts immediately after inoculation with a rather slow separation of both branches, whereas in the empirical data branching appears to occur much later with a quite sudden, strong separation of both branches. However, the example shown in Figure 3D might not be a general representation of branching dynamics occurring in the model. Furthermore, it is hard to assess whether a mismatch would necessarily indicate a problem that is relevant to the main findings of this study.

L117-119: It would be appropriate to acknowledge that Ellner et al. (2021) proposed an extension of their basic model that includes a protected state, which allows for higher SPPLs.

Figure 3: I was wondering whether the colours are suitable for colour-blind people.

L 247: It would be nice if there would be a corresponding illustration of this result.

L 348-349: At first glance, this reads as a generalization beyond the model, which would not be appropriate at this point. It might be useful to add some clarification, e.g. "In the model …"

L 605-612: All this makes sense if the model correctly captures the dynamics of the investigated host-pathogen system. However, whether this is indeed the case for other host-pathogen systems still needs to be demonstrated. It seems to me that at this point it would be appropriate to remind the reader of this limitation.

*Reviewer #3 (Recommendations for the authors):*

In this manuscript, the authors study a model of within-host dynamics of pathogens that aims to capture the fact that some hosts may survive infections while others die from them, even if these infections are identical. The model recovers previous experimental observations, and a prediction from it is tested through new experiments.

Specifically, the authors propose a deterministic model based on coupled partial differential equations, where various parameters describe host resistance (via immune response) and tolerance to the disease. The equations include a specific nonlinear immune regulation, written as the product of an activation of immune defense production by pathogen load, and of a negative feedback that depends both on immune defense level itself and on damage caused. The model can be bistable, allowing for different outcomes (death or survival) under the same parameters, with different initial conditions. A difference in the host initial state (preexisting damage level) suffices to cause a different outcome for the same infection.

The authors make a thorough analysis of the equilibrium states of the model, and of the impact of each parameter. They demonstrate that in this model, infections can evolve either to clearance, to death (once damage exceeds a certain threshold), or become chronic. The authors show that the latter case can in fact be transient, and that the set-point pathogen load, which depends on initial conditions, is then a predictor of life span. True chronicity is also possible. In deadly cases, the authors find that the pathogen load upon death is almost independent from inoculum size, and they demonstrate that the value of this load, together with a hazard ratio, could be employed to distinguish the effects of tolerance and resistance.

Finally, the authors report experiments where *Drosophila melanogaster* is infected by Providencia rettgeri. Their experimental study follows up on a previous one where some of the same authors observed that some hosts survived while others died, under the same infection conditions [Duneau, D. et al. (2017) *eLife*, 6, e28298] – an observation that is recovered by the present model. The new experiments demonstrate that wounding the hosts or suppressing some of their immune effectors both increase the pathogen load upon death. This provides a test of the model prediction that damage should increase the pathogen load upon death as it hinders defense production.

Strengths:

The manuscript provides a comprehensive analysis of the model proposed. An important strength of this work is that a prediction of the model is directly tested by a novel experiment. A Shiny App performing a numerical resolution of the model is provided and allows the reader to directly experiment with its outcomes.

Weaknesses:

The model builds on previous ones which are cited. Some of them already featured bistability [Pujol, J. M. et al. (2009) PLoS Computational Biology, 5 (6), e1000399; Souto-Maior, C. et al. (2018) PLoS Neglected Tropical Diseases, 12 (3), e0006339; Ellner, S. P. et al. (2021) Proceedings of the Royal Society B, 288 (1951), 20210786]. The main formal difference is the specific nonlinear immune regulation form that is chosen. The main consequence is that a rather high set-point pathogen load (SPPL) is possible, and that it can be transient (but note that Ellner et al. proposed another mechanism to obtain a high SPPL). However, no full theoretical insight is provided on the key feature that allows this behavior, as the nonlinear immune regulation chosen is quite complex and includes multiple parameters. The default parameter values are not explicitly related to realistic ones.

While this manuscript is very interesting, I have some concerns that prevent me from recommending publication at least in the present form.

1. The authors should highlight more clearly the impact of the differences between the model that is proposed and previous ones, especially those that already aimed to describe the results of [Duneau, D. et al. (2017) *eLife*, 6, e28298].

– Can a link be made to the model which was proposed in [Duneau, D. et al. (2017) *eLife*, 6, e28298], in particular to the tipping point which played an important role there?

– What novel insights does the present model bring compared to [Ellner, S. P. et al. (2021) Proceedings of the Royal Society B, 288 (1951), 20210786], which is directly motivated by [Duneau, D. et al. (2017) *eLife*, 6, e28298]? The authors mention that a difference is that a nonlethal infection gets almost cleared in that paper (no high SPPL), but this is only true of the "conceptual model" proposed there, and Ellner et al. then propose that some pathogens may be protected from the host immune response, which can result in a high SPPL. This point should be discussed.

2. The model that is chosen is quite complex, which raises several questions:

– The authors refer to the previous study [Mayer, H. et al. (1995) Chaos 5 (1), 155-161] to justify the specific nonlinear immune regulation form they chose, but in that paper, the functions F and G are added and not multiplied in the equation regarding the immune defense level. The authors should motivate their choice.

– Multiple parameters are introduced, and their default values are listed in Figure 2. It would be important how these values are chosen, and to assess how realistic these choices are, and how robust the conclusions are to parameter variations in the realistic range. For instance, is the transient SPPL expected to last for a duration substantially shorter than host lifetime or not?

– Is this the simplest model that allows to have a high SPPL in addition to the bistability already present in other models? What key ingredient allows this?

3. Experimentally, what is the impact of the wound alone (without infection)?

4. The manuscript is quite long and conciseness would make it better. I recommend focusing on the key new insights and minimizing repetitions between the text and the figure legends, as well as between the Results and the Discussion.

5. The use of inappropriate theoretical terms should be avoided.

– In the legend of Figure 3, please avoid the term "stochastic simulations" as the model is entirely deterministic. What is done here is varying the initial conditions used to numerically solve the deterministic equations.

– The model is called "Lotka-Volterra" in reference to the prey-predator model but the similarities are not very strong. For instance, there are no oscillations in the dynamics here, while they are a hallmark of the Lotka-Volterra prey-predator model. Thus, unless there is a specific reason for calling the model "Lotka-Volterra" I would recommend refraining from using this name.

Detailed points:

1. Providing a Shiny App is great as it allows the reader to try the model out, but I strongly recommend to also post the code on GitHub and archive it to Zenodo, as it is durable and identifiable by a DOI.

2. Line 143: "For the sake of simplicity, the first equation of system (1) is written dimensionless": in fact, all three equations are in dimensionless form.

3. Line 166: Is α assumed to be positive? If yes, it would be good to mention it here.

4. Some letters are quite small in figures. In Figure 3 it would be helpful to use different colors and to put explicit legends for each curve.

5. Legend of Figure 3: x_0 is set to 10^-6, not log(x_0). Is the initial level of damage increased by 3% of z_d (legend) or 0.3% (figure)? Please clarify.

6. Figure 4 A,D,E: Can a qualitative explanation be provided for the way the white region size varies between these cases?

7. Line 483: Please spell out GLMM and explain notations (df etc.).

8. In Eq. S1-1 I believe that eta/xi should be to the power l and not l+1.

9. Line 893: "A necessary condition for clearance to be stable is that dx/dt(0,yh,zh)<0": this is problematic because equilibrium implies that dx/dt(0,yh,zh)=0. Do the authors mean dx/dt(epsilon,yh,zh)<0? Please clarify this.

10. Line 912: I believe that the last > should actually read <.

11. Figure S2-2: I believe that this corresponds to a stable equilibrium. It would be good to specify it.

12. Figure S3-1: Please explain what the various curves and lines are.

13. Line 1062: f and g should read F and G.

14. Figure S4-1: I believe that blue and red indicate increasing and decreasing PLUD, not higher and lower.

[Editors’ note: further revisions were suggested prior to acceptance, as described below.]

Thank you for resubmitting your work entitled "A within-host infection model to explore tolerance and resistance" for further consideration by *eLife*. Your revised article has been evaluated by Wendy Garrett (Senior Editor) and a Reviewing Editor.

The manuscript has been improved, but there are some remaining issues that need to be addressed, as outlined below:

*Reviewer #1 (Recommendations for the authors):*

1) Experimental data set

I fully disagree with response 3. My concerns of Bomanins affecting tolerance have now been validated by further studies since this manuscript was first submitted (Xu et al. 2023 EMBO Rep). This supports signals already seen in Lin et al. (the authors should review Lin et al. Fig5bbd-early vs late and Fig6). In the new experiments, the authors use of Drs is inappropriate. Drs is involved in hemocyte recruitment to cancerous tissue and Drs OE suppresses *JNK* activation (Krautz et al. 2020 *eLife*). Drs is further implicated in other anti-cancer or traumatic brain injury responses (papers by Inoue lab 2019 and multiple recently by Wassarman lab). Both Bom and Drs can reasonably affect tolerance, and any mutation might have unintended consequences. A wild-type control is essential, and there is no justifiable reason not to include one.

The RNAi experiments are appreciated, and useful. However they raise some concerns as somehow the PLUD of Cat-IR is 10^2 higher than of Dpt-IR. PLUD is not a metric that should be so sensitive to inter-experiment variation, so these data are difficult to reconcile, and their meaning is difficult to trust. Also, what is the control? No detail is given for "mock-RNAi", and in general RNAi is best used as supporting evidence due to the need to mix genetic backgrounds that could affect results in cryptic ways.

2) Regarding my previous point, apologies if this was not clear, but reflecting 2 years later perhaps I can frame this concern better: lower resistance increasing PLUD is misleading phrasing. The way Figure 5 is presented reflects theoretical space. But PLUD is something defined by a biological limit of the host carrying capacity with a physical volume restriction as a theoretical maximum. As shown previously by Duneau et al. 2017, max PLUD of Dmel individuals in Figure 2 of Duneau was log2(25) to log2(26) across all *D. melanogaster* studied. What is different across strains here (and in Duneau et al. 2017 to some extent) is variance of PLUD. Thus why it's odd to frame it as "PLUD increases by loss of resistance," because in fact the maximum PLUD in Fig7 is pretty consistent across all genotypes and treatments, and no increase is really possible (physical/biological limits). Instead, loss of resistance leads to more consistent microbial growth, faster, and reflected by more consistent mortality outcomes. Thus you get more consistent PLUDS near the maximum physical PLUD. In a wild-type host, resistance creates more complex dynamics, and opens the door for tolerance to impact the outcome and the PLUD. The response supplementary data support this concern exactly. Here, seemingly in 3of4 cases, the PLUD data have an intrinsic survivor bias: blue data points with bimodal survival outcomes have a right skew, while red data points lacking diverse survival outcomes skew to the left, and may even show less diverse ranges (certainly true of 630 and 559).

3) The theory of the model, to this reviewer, seems exceptionally detailed, consistent, and logical. This study has merits and contributes to a body of literature that is seeking to formalize host-pathogen interactions in a mathematical biology framework. I remain concerned with the application of this model to empirical data, which appears to be complex to interpret.

*Reviewer #2 (Recommendations for the authors):*

General appreciation: The authors put a lot of effort in responding to all the reviewer comments, which I think greatly improved the manuscript.

1) There is from my point of view only one issue remaining, which has not been sufficiently addressed. I apologize in case my previous comment on that matter was not clear enough. I had remarked that potential variation in host background mortality should be controlled for in empirical analyses. The authors addressed this issue in the context of their PLUD analyses by removing outliers. However, variation in host background mortality could also be a serious issue for the survival analyses. It seems that currently the implicit assumption in the conducted survival analyses is that there is no variation in background mortality among the compared strains or treatments. Thus, any inferred survival difference is attributed to different infection dynamics. If the possibility of different background mortalities is considered, then a correct interpretation of the survival analyses seems to require the analysis of appropriate non-infection controls. In the simplest case, it might be sufficient to show that there are no apparent survival differences among strains or treatments in non-infection controls. If there are any differences, they would need to be somehow controlled for in the survival analyses of the infected individuals. In case the authors disagree with my argumentation, it would be useful to provide a corresponding explanation in the manuscript why it is not necessary to include non-infection controls in the conducted survival analyses.

*Reviewer #3 (Recommendations for the authors):*

General appreciation: The authors have addressed my comments thoroughly, and I thank them for this. The manuscript is improved as a result. I still have two points about the model.

1) I still find the use of the term "stochastic simulations" misleading in the legend of Figure 3. I recommend that the authors explicitly specify "Dots corresponds to results of stochastic simulations where the initial pathogen load is randomly drawn (…) and then the deterministic equations of the model are solved numerically."

2) I got a bit worried by the authors' response regarding parameter values and robustness to varying them. Indeed they state: "As often with models, duration can be changed almost at will by adjusting parameters or initial conditions!" One would hope that if the parameters are varied in a physiological range, the conclusions do not vary "at will"… This said, I understand the difficulty of precisely determining each parameter.

Overall, I find that the theory-experiment comparison is an important strength of this manuscript.

---

## [Author Response]

[Editors’ note: the authors resubmitted a revised version of the paper for consideration. What follows is the authors’ response to the first round of review.]

Reviewer #1 (Recommendations for the authors):The authors provide an impressive, detailed, and nuanced discussion of parameters contributing to outcome of infection. This statistical model considers a variety of theoretical contributors to pathogen growth, immune defense, and damage amelioration. The topic at hand is of general interest in modelling infectious disease processes, and to understand what parameters contribute most to readouts of disease progression. The framework the authors set out is logically consistent, and there is a clear effort to be thorough in constructing their statistical framework.However I am unconvinced that the experimental data support the conclusions of the model (Figure 7-8). As the model is entirely theory, it is essential that the experimental design and data used to validate the model robustly support its predictions. The authors may be able to address these concerns in revisions, and increase confidence in their conclusions.The authors use different experimental treatments (ex. wounding prior to infection) and three key experimental readouts to inform on the interactions of their model parameters (Survival, SPPL, PLUD). However I am unconvinced by the experimental data presented to validate their model. This unfortunately derails the paper right at the critical stage, and deflates confidence in the conclusions.Lines 484-497 (Figure 7) – The key question is if mortality increase caused by wounding is loss of resistance or tolerance. The metric used to test this is PLUD. The authors do find a general higher PLUD in wounded flies, though only in 2/4 genotypes. I am concerned this result comes from survivor bias effects.As SPPL is variable across individuals, any flies surviving the initial phase of infection will suppress pathogen growth to a transient SPPL. However flies dying rapidly from the infection have uninterrupted pathogen growth towards PLUD. As seen in Figure 7A, many No Wound flies survive this initial infection, but have a steady mild mortality rate at later time points. On the other hand, Wounded flies die almost 100%, and survival curves plateau.Therefore Wounded flies mostly die from unchecked bacterial growth. However No Wound flies die from either pathogen growth (majority) or at later time points, despite having suppressed the initial infection (minority). I suspect this is caused by a factor the authors ignore: Zd of their model is not fixed. Damage from the initial infection and autotoxicity from the immune response may not kill hosts in the first 24h, but can ultimately be the cause of death (ex. organ failure). In beetles, it is known that autotoxic effects of the immune response reduce lifespan associated with malpighian tubule dysfunction (doi:10.1098/rspb.2017.0125). Even a fly that successfully suppresses pathogens can die from failing organ systems without needing recurrent pathogen growth, which may not even be possible when the blood is antimicrobial.The significance of Figure 7B is driven almost entirely by low-PLUD outliers present mostly in No Wound flies. This suggests PLUD differences are driven by loss of tolerance in No Wound survivors, and not loss of resistance in dying Wounded flies. This is the exact opposite conclusion of the authors. If the authors have time of death associated with their data points, this could confirm or counter my concern regarding low-PLUD outliers.

Reviewer #1 suggests that some flies in our experiments could have died from something else than the infection. These individuals should thus both die late and at low pathogen load, which could produce a flawed PLUD difference among treatments if ever the proportion of such flies differ among treatments. We agree that this is a potential problem, although figure 1.2 demonstrates that the PLUD in our experiment does not decrease when time to death increases, as the Referee suggests. It is also worth noting that, even though there was overall more death in the wounded treatment, the period over which the PLUD was collected was similar in each treatment (i.e. between 12 and 24 hr). Furthermore, previous data (Duneau et al. *eLife* 2017) and unpublished data repeatedly show that the PLUD does not depend on the time to death. To fully investigate the concern of the reviewer, we propose that this can also be solved by removing outliers from PLUD data. To explore this solution, we ran simulations where individuals either die from the infection or from some other unrelated cause. In the latter case, we considered that death occurs at a time that follows a Weibull distribution with fixed parameters. More precisely, if a random time value drawn from this distribution was found to be smaller than the time to death predicted by our WHD model, we considered that the host was not killed by the infection. In this case, we used our model to compute the pathogen load at the randomly drawn time of death. We then tested the efficacy of a Rosner test to detect individuals that did not die from the infection in these simulated data. We found that most individuals which did not die from the infection, and have hence a lower PLUD, are efficiently detected by the Rosner test (see Supplementary materials S7.1). Our simulations demonstrate that individuals which did not die from the infection make the PLUD being underestimated; removing them from the dataset is sufficient to suppress this bias. We therefore used this technique to re-analyze our experimental data. We found that keeping or removing outliers does not change our general conclusion: wounding flies before injection does significantly increase the PLUD, as our model predicts. We now present two analyses, with outliers either removed or included.

Lines 498-520 (Figure 8) – The authors use Bom mutants as a control, saying they have little effect on gram-negative bacteria (Hanson et al). But checking Hanson et al., they do not infect Bom mutants with gram-negative bacteria. In fact, Duneau et al. (2017,BMC BIOL) reports that the Toll pathway mediates a sexual dimorphism in response to P. rettgeri infection. Duneau et al. (2017,eLife) also showed PLUD does not vary across Imd, Toll, Phagocytosis, or Melanization mutants? These studies from one of the authors are in direct contradiction to the logic of this experiment.Also Bomanin function is not known, and there is no evidence they are directly antimicrobial based on existing studies. Instead, the authors cite Lin et al. in their discussion (Lines 653-657), which is a study implicating Bomanin effects on tolerance through Bombardier, including a mortality associated with immune activation by heat-killed bacteria. It is likely Bomanins affect tolerance, so use of Bom mutants as a baseline for comparison is wholly inappropriate. Why did the authors not use a wild type control here?Additionally, A group died more than Bomanin, so their claim that Defensin does not affect survival seems untrue in their conditions. The difference in PLUD in A, B, and AB is less convincing. Defensin is also associated with clearance of aberrant cells (doi:10.7554/eLife.45061), so even assuming a mild difference is true, it can again be due to tolerance effects. Why did the authors not compare Diptericins specifically in their question, given previous studies on the Diptericin and P. rettgeri? This is not a strict request for additional experiments, but I would be more convinced by use of AMP genotypes that might have specific activity against P. rettgeri, and genotypes with related but less important activity. A wild type control is necessary.

The most obvious choice for a control may seem to be a line with the same genetic background than the mutant we have tested and no AMP gene deleted, typically a White lineage here. But we know that even though this seems appropriate, it is not always the best solution. We advocate that using a mutant from the same background but with expected lower specificity against the pathogen is an elegant way to control for ”being a loss-of-function mutant” and ”having a given genetic background”. In our specific case, we do not understand the conclusion of the reviewer regarding ”Bomanin effects on tolerance through Bombardier”. In fact, *Bom*^∆55*C*^ mutants have been demonstrated to have a PLUD comparable to controls when infected with the gram positive bacteria *Enterococcus faecalis* (Lin et al., 2020). This, together with the fact that none of the available White mutants have a genetic background completely comparable to the AMP mutants we have studied, made us decide to keep *Bom*^∆55*C*^ as a control in this experiment. We nevertheless decided to strengthen our result with two other independent experiments. In a first one which follows the same logic, we compared *Dpt^sk1^*, which lacks Diptericins (Hanson et al., 2019) to *Drs^R1^*, which lacks Drosomycin. Drosomycin being an antifungi AMP that is inactive against bacteria, we used it as a control. We found that the mutant with Diptericins deleted has a higher PLUD than the control, as in the first experiment. In a third experiment, we compared a mutant with Diptericin A silenced by a RNAi to a control that expresses a mock RNAi. We again found the same result. We insist on the fact that not only do the three experiments yield qualitatively similar outcome, but also the increases in PLUD they each reveal are quantitatively close (approx +0*.*2).

– It is not clear why the authors do not modulate JAK-STAT stress responses as a test of resistance vs tolerance (Lines 149-155 are not convincing).

The difficulty in testing how a loss of tolerance can impact the PLUD lies in the fact that most genes proposed so far as ”tolerance” genes have been identified using the PLUD itself or using methods that our study suggests may not be appropriate. Testing these genes again would not only be redundant but would also make our reasoning circular or contradictory. Another approach is to use candidate genes whose functions are both well understood and compatible with what we believe tolerance mechanisms could be. Genes that control the JAKSTAT stress response could be candidates, but we reasoned that silencing a master regulatory gene, which has numerous downstream targets, would be difficult to interpret. Instead, we opted for the Catalase gene, whose role in detoxifying ROS is well documented and can be considered part of the tools involved in mitigating damage caused by an infection. We first found that silencing this gene with RNAi significantly increased susceptibility to *P. rettgeri* infections. We then demonstrated that, as the model predicts, the PLUD is lower in Catalase-deficient flies than in control flies.

– What is the status of the Diptericin locus in RAL-818, RAL-630, etc… Given Unckless et al. (2016)?

RAL-818, RAL-630, RAL-584 and RAL-559 have all the same Diptericin locus given Unckless et al. 2016, there are dptS69. We now mention this in the legend of figure 7.

– In the discussion the authors comment on SPPL nicely. I am curious if they have considered the scenario where bacteria in SPPL stage are in a hibernation state. This is common in uropathogenic E. coli, which reside in host epithelial cells and cause recurrent outbreaks. But those *E. coli* are not constantly causing damage while in hibernation. Does the model assume a constant rate of damage for bacteria in SPPL phase?

Reviewer #1 suggests that bacteria could be in a dormant state, probably hidden from immune response, when load stabilizes at the SPPL. This is indeed clearly documented in some infections (although not in *Drosophila*), and Ellner et al. (2021) proposed that this could be a way to maintain high SPPL, from which infection could reemerge. We did not consider this possibility, in part because our model does not need this additional mechanism to predict high SPPLs. Referee #1 also asks whether the rate at which pathogens produce damage is constant in our model, or could rather vary depending on whether the pathogen population is at SPPL or not. In our model, we assumed that this rate (*ω*) is constant. This is clearly a simplifying assumption, as many (if not all) pathogenic bacteria do regulate the expression of their virulence genes over the course of the infection (e.g. *Xenorhabdus nematophila* do not express virulence genes upon infection, but only after exponential phase (Faucher et al., 2021)).

– The authors should include a README file explaining the columns in their supplemental data files

We now provide a RMarkdown file which contains all the analyses described in this paper. We believe that this makes README files unnecessary, as graphs and the analyses are described in this file.

Reviewer #2 (Recommendations for the authors):Lafont et al. developed a novel theoretical model that describes how host resistance and tolerance affect within-host pathogen dynamics. Their model focuses on recently documented with-host bacterial dynamics in *Drosophila*. Specifically, it was previously shown that in some cases the same inoculation dose can lead to two distinct outcomes: (1) the host successfully controls pathogen growth, which leads to an apparently stable set-point pathogen load (SPPL), and (2) the pathogen growth out of control reaching very high levels termed pathogen load upon death (PLUD), which causes rapid host death. The developed model can successfully reproduce this type of branching process. In addition to other existing models, the authors are able to reproduce the empirically observed pattern that the SPPL can increase with an increasing inoculum dose. Two findings of the model analysis are particularly relevant for empiricists studying resistance and tolerance. First, the results contradict the previous belief that only tolerance affects the PLUD. Instead, the authors now demonstrate that also resistance can affect the PLUD. Second, the authors raise considerable doubt on the validity of a commonly applied method for quantifying tolerance, which is based on measuring the reaction norm of host fitness in relation to pathogen load (measured either by the inoculum dose or by pathogen load at one point during the infection). Specifically, the authors show that this reaction norm can be more strongly influenced by resistance than by tolerance. To overcome these problems, the authors propose a novel method to infer variation in resistance and tolerance, which is based on integrating measures of the PLUD and host survival. Finally, the authors validated parts of their model with experimental infection studies on *Drosophila melanogaster*. These studies demonstrate the predicted effect that the PLUD increases and survival decrease due to wounding, and due to the knockout of important resistance genes. The agreement between the predicted and observed effects is interpreted by the authors as a confirmation of the general validity of their model and the proposed method.Taken together, the authors present very interesting theoretical and empirical results with potentially far-reaching consequences for understanding and measuring how hosts respond to pathogen infections. Nevertheless, there are some limitations of this study, which I think were not sufficiently considered when interpreting the results.1. The theoretical and empirical work is biased towards resistance. The pronounced differences in the way both host strategies were modelled and empirically investigated could strongly limit the validity and generality of the model and the proposed method. As the authors explain, due to the lack of known tolerance genes in *Drosophila*, they were not able to empirically test tolerance specific predictions of their model. The authors acknowledge this limitation, but do not see it as a major problem. However, it appears doubtful that the empirically measured resistance effects are sufficient for concluding that also the tolerance effects are correctly predicted by the model. In addition to these limitations on the empirical side, the theoretical side contains the limitation that tolerance was modelled in a much more simplistic way compared to resistance. In the model, host resistance is characterized by three major features: it depends on host condition (i.e. the level of damage), it is costly (because it generates damage), and it is modulated by pathogen load. In contrast, the implementation of tolerance lacks all of these features: it is cost-free, it is independent of host condition and it is independent of pathogen load or amount of damage. The authors acknowledge that tolerance mechanisms are known to be modulated during an infection, but because related details are still unknown they chose to model tolerance in a very simplistic way. This choice is certainly a reasonable first step. Nevertheless, it leaves the possibility that more complex tolerance effects could strongly affect the dynamics predicted by the model. Especially in combination with the lack of empirical data on tolerance, this makes it challenging to fully assess the validity and generality of the model and the proposed method.

Reviewer 2 expressed concerns about our oversimplified description of damage dynamics, particularly in comparison to defense modulation, and suggested that this may render our predictions questionable. Indeed, we did simplify the dynamics of damage, much like we simplified the dynamics of pathogens (using a simple logistic growth model and ignoring potential lag phases) and defense (describing it with a single variable, despite the possibility of different regulatory mechanisms for distinct lines of defense). In this respect, Reviewer 2 is correct in stating that our model represents a ”reasonable first step”.

The crucial question is whether this model is useful to better understand Within Host Dynamics. Our first response to this question is that some of our findings remain valid even when damage dynamics are more complex. We have demonstrated that including modulation of damage repair does not change the condition for bistability (as mentioned in lines 283-285, with detailed analysis in Supplementary Material S1.2). We are confident that most of our general qualitative conclusions would hold in a model with more complex damage dynamics. Quantitative predictions, such as the relative impact of parameters on PLUD and HR, could indeed be affected by the modulation of damage repair. However, this would not affect our general conclusions.

Our second response to the model’s usefulness is that, despite its simplifications, it adequately reproduces previous experimental observations (e.g., the SPPL varies with dose, while the PLUD does not) and accurately predicts our own findings (e.g., the PLUD increases when the host is wounded or its immune defense is impaired). In that regard, and because it was a comments shared with other reviewers, we investigated experimentally the interpretation of a lower PLUD. We confirmed that it decreases when a gene likely involved in tolerance is silenced. We think that going any further, by including a more realistic description of damage, could be interesting but is beyond the scope of this paper and would not change qualitatively our conclusions.

2. Effects on reaction norms are not very well explored. The authors have shown that the relationships between inoculation dose and host survival contradicts assumptions made in empirical studies (Figure 5). However, the relationship between SPPL and host survival is not explicitly shown and it is, therefore, hard to assess whether the problem also applies to this relationship. A more detailed analysis would be particularly informative for empirical studies that measure pathogen load during the infection. Furthermore, conducting empirical tests of the predicted effects is an important task that remains to be done before concluding that the reaction norm approach is generally flawed.

The SPPL has no clear relation to LT as hosts which stabilize the infection (at the SPPL) should survive the infection. We nevertheless agree that our analysis of the reaction norm was rather elementary. We have now performed a full fledged elasticity analysis, which we provided as Supplementary Material S6. This was done by varying the inoculated dose from 10^−6^ to 10^−5^, computing the corresponding Lethal Time (LT) and adjusting a regression line of LT on inoculation dose, using log-log scale. We then computed the derivative of the slope of this regression line with respect to each of the 14 parameters of the model (figure S6.1). We found that, as all the other proxies we tested, the slope depends on both resistance and tolerance parameters. Tolerance parameters did not prove to have stronger effects than resistance parameters. We even demonstrate that two tolerance parameters may have opposite effects: increasing the tolerance to damage (*z_d_*) indeed makes LT less dependent on dose, while increasing the rate of damage repair (*ξ*) conversely makes the reaction norm steeper. This later effect probably comes from defense production being connected to damage. We indeed found that the counter-intuitive effect of *ξ* decreases when the parameter which controls this connection (*ψ*) is lowered. The main conclusion of our analysis is that examining the relationship between PLUD and Hazard Ratio is more effective than analyzing the steepness of the reaction norm for distinguishing tolerance from resistance.

We also want to highlight that the approach we propose in our work is somehow similar to the reaction norm approach. Simply, we relate LT (or the hazard ratio which is yet another way to quantify mortality differences) to the PLUD, instead of relating it to dose or to any load measured at an arbitrary time. The advantage of using the PLUD, we advocate, is that it reflects a well defined physiological state of the host. Our work demonstrates, nevertheless, that the approach has its limits see for example our analysis.

3. Host background mortality is not considered. The authors developed a deterministic model in which host death can only occur due to an infection. Host background mortality due to other causes is not considered. This is not necessarily a problem for the theoretical analyses. However, to avoid biased results, empirical applications of the proposed method should control for potential variation in background mortality among different host lines.

Reviewer 2 is correct: we did assume in our model that the sole cause of death is infection, and we therefore did not consider potential variations in background mortality. In reality this comes from the fact that we did neglect ageing in our model, because we aimed at predicting the short term consequences of infections. This could be seen as a limitation of our work, but 1- including ageing would require to complement our model by more equations describing senescence, 2- the systemic infections that we are studying here are over a time frame where the impact of ageing is very limited. However, indeed, ageing could play a role by changing the ability of the host to control or tolerate over the course of the infection when chronicity is long. We are convinced that investigating the interactions between senescence and immune response is of prime importance to better understand the functioning and the evolution of immunity. We nevertheless consider that this is far beyond the scope of this paper.

Reviewer 2 is also concerned that neglecting background mortality could compromise our interpretation of experimental results. This somehow connects to the point Reviewer 1 made on potential bias in PLUD analysis when individuals can die from some causes other than the infection. As mentioned above, we now propose an detailed analysis of this point in Supplementary Materials S7.1.

4. Very wide and skewed empirical PLUD distributions. In some cases, the empirically measured PLUD distributions are very skewed and very wide with a difference of up to five orders of magnitude between the smallest and the largest values (Figure 7B). It is not clear how this enormous variation arose and whether this might indicate a mismatch to the dynamics predicted by the model.

Within our modelling framework, the variations in PLUD can have three distinct origins. First, as Reviewer 1’s suggested, some of the flies which in this experiment probably controlled the infection but died from the wound. These flies therefore die at a load which is much lower than a ”proper” PLUD. We propose a method to detect these outliers and assess their impact on the analysis of the PLUD (Supplementary Materials S7.1).

Second, flies used in the experiments are not completely identical: a small amount of genetic diversity may remain in our stocks, and even though they are raised in the same vial, small developmental differences may exist among flies. This would be analogous, in our model, to having some random fluctuations in parameters.

Third, the condition in which the infection is started may fluctuate: the dose cannot be perfectly controlled, the injection produces a wound which effect may vary from one individual to another, etc. This can be taken into account in our model by considering the initial values of the three variable (*x*_0_, *y*_0_ and *z*_0_) as random variables, as we did when simulating mortality (see sections 5 and 6). On top of these three sources of variation, the PLUD in our experiments have been estimated by plating diluted samples (see Supplementary Materials S7 for a complete description of the procedure). This procedure, as any experimental method to estimate bacterial loads, generates random errors which may inflate the natural variation in PLUD.

Which source of variation prevails in our experiments is a difficult question. We think that Reviewer 2 probably refers to flies with PLUD that are orders of magnitude lower than majority; these flies probably died from something else than the infection. We now control this in our new analysis (section 7).

Finally, the PLUD is the result of an exponential bacterial growth. The difference between a PLUD of 10e6 and another of 10e7, which is the majority, may seem large but in fact it takes only 3 bacterial divisions to go from the first to the latter.

5. Differences between observed and predicted branching dynamics. There seems to be a mismatch between the empirically observed temporal pattern of branching (Figure 1) and the corresponding model dynamics (Figure 3D). In the model, branching starts immediately after inoculation with a rather slow separation of both branches, whereas in the empirical data branching appears to occur much later with a quite sudden, strong separation of both branches. However, the example shown in Figure 3D might not be a general representation of branching dynamics occurring in the model. Furthermore, it is hard to assess whether a mismatch would necessarily indicate a problem that is relevant to the main findings of this study.

The two curves of figure 3D do not represent a situation similar to that of figure 1; they indeed correspond to two simulations performed with different parameter values, which in terms of experiment would be equivalent to having two genotypes with different immune responses. Figure 1 rather represents an experiment where a single fly genotype is analyzed. This would be analogous to having the green curves of 3A and 3D in the same graph.

L117-119: It would be appropriate to acknowledge that Ellner et al. (2021) proposed an extension of their basic model that includes a protected state, which allows for higher SPPLs.

We now mention this modified model of Ellner et al. (2021).

Figure 3: I was wondering whether the colours are suitable for colour-blind people.

We did check this (using https://davidmathlogic.com/colorblind/), and can confirm that the colors we have chosen can be distinguished by colorblind people, including those of figure 3 and 4.

L 247: It would be nice if there would be a corresponding illustration of this result.

We are afraid we do not understand this request: L247 of the previous version of the work was saying that wounding the host did not change the outcome of the infection when *γ* is high. This corresponds to the two gray curves of Figure 3A. The illustration of this result is therefore already here, unless we have misunderstood referee’s point.

L 348-349: At first glance, this reads as a generalization beyond the model, which would not be appropriate at this point. It might be useful to add some clarification, e.g. "In the model …"

Lines 348-349 corresponds to the title of section 5 which was in the previous version ”Experimental measures of loads or of mortality always mix tolerance and resistance”. We understand that using the word ”experimental” in the title may be a problem, as it may suggest we have experimentally tested this prediction. We have changed the title to ”Load and mortality measurements should reflect both tolerance and resistance to infection”. We think that avoiding the word ”experimental” addresses the referees’ concern.

L 605-612: All this makes sense if the model correctly captures the dynamics of the investigated host-pathogen system. However, whether this is indeed the case for other host-pathogen systems still needs to be demonstrated. It seems to me that at this point it would be appropriate to remind the reader of this limitation.

We propose in the discussion of our paper that demonstrating a positive correlation between SPPL and dose would be a way to evidence that the SPPL is unstable. Reviewer 2 suggests that this holds in our model but may be untrue in other host-pathogen systems. We think on the contrary that this specific point is one instance where our conclusions are general: from a mathematical point of view, an unstable SPPL is no more than a particular point along the pathogen dynamics, and as such it must vary with dose. If conversely the SPPL is stable it corresponds to an equilibrium point which by definition cannot vary with initial conditions (and hence dose). From a biological point of view, the problem becomes more quantitative: systems might exist where variations in dose have little impact on the SPPL, even when unstable. In conclusion, evidences that the SPPL increases with dose would prove that the SPPL is unstable; lack of evidence would prove… nothing!

We further propose in the same paragraph that the SPPL being unstable would suggest that ”the pathogens, or the damage they cause, obstruct the immune response”. It could be argued that this holds only in our model, where this is the connection between damage and defense production that makes bistability possible. This is not completely true because Ellner et al. (2021), Yu et al. (2021) and van Leeuwen et al. (2019) also predict that bistability originates from the infection hindering the production of defense. Similarly Zhang (2016) found that quorum sensing molecules which reduce the efficiency of immune defense can yield bistability. The mechanism is quite different from what we considered in our model, but in some sense it follows the same logic: again, the system is bistable because defense is hindered during late infection.

A completely different line of explanation would be that pathogens evolve within their host. Imagine for example that mutations conferring resistance to immunity appear in pathogens with a low probability. Hosts which carry resistant pathogens should rapidly loose control and die from the infection; other hosts should rather stabilize pathogens at a sustainable load and survive. Duneau et al. (2017) demonstrated in his experiment that the bifurcation he observed could not be explained by within host evolution. We therefore ignored this possibility in our analysis, but this is clearly something which may happen in some host-pathogen systems. Considering this situation does not, however, compromise our prediction as the SPPL should not vary with dose in such circumstances. This is in fact what Ellner et al. (2021) have demonstrated in the version of their model where bacteria can shift to a state where they multiply slowly but are protected against AMP.

We have now added a paragraph in the discussion which presents the possibility of within host evolution and discuss the robustness of our predictions concerning the SPPL.

Reviewer #3 (Recommendations for the authors):[…]Weaknesses:The model builds on previous ones which are cited. Some of them already featured bistability [Pujol, J. M. et al. (2009) PLoS Computational Biology, 5 (6), e1000399; Souto-Maior, C. et al. (2018) PLoS Neglected Tropical Diseases, 12 (3), e0006339; Ellner, S. P. et al. (2021) Proceedings of the Royal Society B, 288 (1951), 20210786]. The main formal difference is the specific nonlinear immune regulation form that is chosen. The main consequence is that a rather high set-point pathogen load (SPPL) is possible, and that it can be transient (but note that Ellner et al. proposed another mechanism to obtain a high SPPL). However, no full theoretical insight is provided on the key feature that allows this behavior, as the nonlinear immune regulation chosen is quite complex and includes multiple parameters. The default parameter values are not explicitly related to realistic ones.While this manuscript is very interesting, I have some concerns that prevent me from recommending publication at least in the present form.1. The authors should highlight more clearly the impact of the differences between the model that is proposed and previous ones, especially those that already aimed to describe the results of [Duneau, D. et al. (2017) eLife, 6, e28298].– Can a link be made to the model which was proposed in [Duneau, D. et al. (2017) eLife, 6, e28298], in particular to the tipping point which played an important role there?

In Duneau et al. (2017), the tipping point was defined as a threshold pathogen load above which the host looses control over pathogen proliferation. There is no such threshold in our new model. Still, a set of points (i.e. a region in the phase space) can be loosely defined that corresponds to points where trajectories leading to rapid death separate from those that stabilize (transiently or permanently) at the SPPL.

– What novel insights does the present model bring compared to [Ellner, S. P. et al. (2021) Proceedings of the Royal Society B, 288 (1951), 20210786], which is directly motivated by [Duneau, D. et al. (2017) eLife, 6, e28298]? The authors mention that a difference is that a nonlethal infection gets almost cleared in that paper (no high SPPL), but this is only true of the "conceptual model" proposed there, and Ellner et al. then propose that some pathogens may be protected from the host immune response, which can result in a high SPPL. This point should be discussed.

Ellner et al. (2021) proposed a model based on the same experiment which motivated ours. Their model considers that bacteria produce protease which degrade defense. The model also includes a term by which each defense unit is consumed or inactivated when it contributes to kill pathogens. This mechanism alone makes bistability possible (see for example Gilchrist and Coombs, 2006), but the action of protease on defense facilitates it. We have now extended our analysis of bistability in Supplementary Material S1.1, to include a discussion of what biological mechanisms can create bistability. The conclusion is that bistability requires a mechanism that reduces defense when the infection progresses. This can be defense consumption, the action of protease or any other virulence factor produced by the pathogen which targets immune defense, or even a negative impact of accumulating damage on the production of new defense (van Leeuwen et al., 2019; Yu et al., 2021; Zhang, 2016, e.g.). Of course these different mechanism are not mutually exclusive.

In case of bistability, Ellner et al. (2021) conclude that controlled infections are maintained at a low load, which is incompatible with the SPPL measured in Duneau et al. (2017). They then propose a more complex model where pathogens can change phenotype and ”hide” from the immune defense. These hidden pathogens would contribute to maintain load at a high level, even though the infection is controlled.

We do not include such a mechanism in our model, but still predict SPPL which are compatible with experimental observations. We think that this has not much to do with any fundamental difference between Ellner’s model and ours. It rather comes from the fact that we considered unstable SPPL while Ellner *et al.* analyzed only stable SPPL. In our model too, stable SPPL are low; in addition they do not increase with dose, which contradicts experimental observations. Only unstable SPPL can be high and increase with dose. We have ran some simulations of Ellner’s model, and have found that their model too can predict unstable SPPL, while other models which have only two variables (similar to Souto-Maior et al., 2018) cannot. The possibility of an unstable SPPL is therefore not a specificity of our model, even though Ellner et al. (2021) have not realized it. We believe that considering non-equilbrium situations in the dynamics of an infection is one of the main contribution of our work.

Finally, the main difference between Ellner’s model and ours is that we considered the dynamics of damage accumulation, and we used it to predict death. This is what allowed us to predict the PLUD and study its properties. We believe that this is also an important contribution of our work.

2. The model that is chosen is quite complex, which raises several questions:– The authors refer to the previous study [Mayer, H. et al. (1995) Chaos 5 (1), 155-161] to justify the specific nonlinear immune regulation form they chose, but in that paper, the functions F and G are added and not multiplied in the equation regarding the immune defense level. The authors should motivate their choice.

We acknowledge the difference between the model in Mayer et al. (1995) and ours regarding how function *F*(*x*) and *G*(*y,z*) relate to each other in the differential equation for level of immune defenses (*dy/dt*). Firstly, we want remind the reviewer that the functional form chosen for *G*(*y,z*) differs from Mayer et al. (1995) because in our model we also track damage independently (*dz/dt*). With this in mind we argue that *G*(*y,z*) represents the down regulation of defense production, not the degradation of already produced defense. If we had kept the additive form, *G*(*y,z*) would actually down regulate defense production even when there is no defense produced.

– Multiple parameters are introduced, and their default values are listed in Figure 2. It would be important how these values are chosen, and to assess how realistic these choices are, and how robust the conclusions are to parameter variations in the realistic range. For instance, is the transient SPPL expected to last for a duration substantially shorter than host lifetime or not?

We understand the concern of Reviewer 3 about the duration of control in case of transient SPPL. As often with models, duration can be changed almost at will by adjusting parameters or initial conditions! We believe that the more general concern about parameter values raises two remarks:

(a)the model, as it is now, aims more at producing qualitative scenarios that quantitative predictions;(b)obtaining quantitative predictions that could be directly compared to experimental measurements would require that the model is statistically adjusted on data, so that parameters can be estimated. This is way beyond the scope of this paper, although this is of course something we will try in a next future.(c)parameter values are somewhat arbitrary, and this is precisely the reason why we have run sensitivity analyses of each single quantity that we have studied.

- Is this the simplest model that allows to have a high SPPL in addition to the bistability already present in other models? What key ingredient allows this?

This is a very good point, and the simple answer is no: simpler models can predict bistability. As we have explained above (and as we now detail in Supplementary Material S1.1) simple two variables models that include only a term of defense consumption can be bistable. But it takes an extra variable, as damage in our model or protease in Ellner’s, to predict transient SPPL.

3. Experimentally, what is the impact of the wound alone (without infection)?

The wound alone does reduce survival, with only 59% of the flies injected with sterile PBS surviving at four days when wounded before injection, while 89% of the non wounded flies have survived the PBS injection. When injected with *P. rettgeri* survival at four days drops to 42% for non wounded flies and to 1% for wounded ones. The difference in survival is therefore much higher when bacteria are injected than when sterile PBS is injected (which we have tested using a Cox model, now provided in the RMarkdown file) which means that the wound has a direct effect on fly survival but also increases its susceptibility to infection.

4. The manuscript is quite long and conciseness would make it better. I recommend focusing on the key new insights and minimizing repetitions between the text and the figure legends, as well as between the Results and the Discussion.

We agree that the paper is longer than the average manuscript. This comes from the fact that we want to make our model and theoretical analyzes understandable to empiricists. We therefore, for example, opted for full names rather than symbols and chose to repeat results explanations in figure legends, to make our explanations the easiest possible to understand.

5. The use of inappropriate theoretical terms should be avoided.– In the legend of Figure 3, please avoid the term "stochastic simulations" as the model is entirely deterministic. What is done here is varying the initial conditions used to numerically solve the deterministic equations.

The model is indeed deterministic, although we do introduce uncertainty in our model by randomly sampling initial conditions. The model itself cannot be said to be stochastic but the simulations are, even though the only stochastic process is in choosing the initial conditions. We argue then that the use of stochastic to speak of simulations is appropriate here.

– The model is called "Lotka-Volterra" in reference to the prey-predator model but the similarities are not very strong. For instance, there are no oscillations in the dynamics here, while they are a hallmark of the Lotka-Volterra prey-predator model. Thus, unless there is a specific reason for calling the model "Lotka-Volterra" I would recommend refraining from using this name.

We could argue that the equation which describes the dynamics of pathogens follows the exact same logic than the classic LV predator-prey model, with a logistic growth and a predation term that follows the mass action principle. We could also argue that our model does produce oscillations, in particular when a stable SPPL is approached. However, we agree that the second equation differs from that of the classic LV, and that of course predator-prey models have nothing similar to damage accumulation. We therefore decided not to refer to Lotka-Volterra in the title of the section. However, we kept it in the text as it would be very informative for most theoreticians.

Detailed points:1. Providing a Shiny App is great as it allows the reader to try the model out, but I strongly recommend to also post the code on GitHub and archive it to Zenodo, as it is durable and identifiable by a DOI.

Per reviewer’s recommendation the code for the Shiny web application has been upload on a public GitHub repository and archived to Zenodo with the following DOI: 10.5281/zenodo.13309653

2. Line 143: "For the sake of simplicity, the first equation of system (1) is written dimensionless": in fact, all three equations are in dimensionless form.

We acknowledge that our wording was not adequate. We meant to say that the system was scaled by expressing *x* in terms of proportion of carrying capacity and *t* as a unit of bacterial generation. In turn, all equations are impacted by these changes.

3. Line 166: Is α assumed to be positive? If yes, it would be good to mention it here.

Yes *α* is positive. We now mention this in the text.

4. Some letters are quite small in figures. In Figure 3 it would be helpful to use different colors and to put explicit legends for each curve.

We have increased text size in Figure 3, so that it is easier to read. However we have maintained our initial color choices, as each color represents a specific parameter set. Differences in initial conditions are indicated by labels, which we hope are now clearer with the larger text.

5. Legend of Figure 3: x0 is set to 10−6, not (logx0). Is the initial level of damage increased by 3% of z_d (legend) or 0.3% (figure)? Please clarify.

It should be 0.3 % indeed, legend and text were both corrected to reflect that.

6. Figure 4 A,D,E: Can a qualitative explanation be provided for the way the white region size varies between these cases?

The white region is defined by the threshold γc above which clearance is stable. This threshold increases with *ψ*, hence the smaller white zone in Figure 4A, where *ψ* is decreased from 2 to 0.5, compared to Figure 4D. Conversely, γc does not depend on parameters involved in defense activation, i.e. on parameters that define function *F*. This explains why the white area in Figure 4E, where only *u* has been changed, is the same than in Figure 4A.

7. Line 483: Please spell out GLMM and explain notations (df etc.).

Abbreviations designing particular statistical models are now all spelled out.

8. In Eq. S1-1 I believe that eta/xi should be to the power l and not l+1.

Reviewer 3 is right: there was a typo in Eq. S1-1, a bit more intricate than just an extra +1, though… This is now corrected.

9. Line 893: "A necessary condition for clearance to be stable is that dx/dt(0,yh,zh)<0": this is problematic because equilibrium implies that dx/dt(0,yh,zh)=0. Do the authors mean dx/dt(epsilon,yh,zh)<0? Please clarify this.

. Reviewer 3 is right: by definition dx/dt,(0,y~h,z~h) is zero. We reformulated the sentence as follows: ”A necessary condition for clearance to be stable is that dx/dt,(x,y~h,z~h) is negative when *x* tends towards zero, which in turn requires that y~h>1/δ.”

10. Line 912: I believe that the last > should actually read <.

Reviewer 3 is right: if *ψ >* 0 and *θ >* 0, then z¯∂G/∂z<0˙ and the system can be bistable.

11. Figure S2-2: I believe that this corresponds to a stable equilibrium. It would be good to specify it.

Figure S2-2 indeed describes a stable SPPL. We now clarify this in the legend.

12. Figure S3-1: Please explain what the various curves and lines are.

We now explain in figure legend that black curves delimit the parameter region for which the system is bistable. The dark-red curves enclose a sub-region in which the SPPL is transient in the conditions of the simulation. The red curve is the value of *α* below which a high load infection will always kill the host. This explanation is provided for each single elasticity analysis given in Supplementary Material.

13. Line 1062: f and g should read F and G.

This is now fixed.

14. Figure S4-1: I believe that blue and red indicate increasing and decreasing PLUD, not higher and lower.

This is correct: the figure represents partial derivatives, and therefore variations, not values. We have modified the figure legend to clarify this point.

References

Duneau, D., Ferdy, J.-B., Revah, J., Kondolf, H., Ortiz, G. A., Lazzaro, B. P., and Buchon, N. (2017). Stochastic variation in the initial phase of bacterial infection predicts the probability of survival in *D. melanogaster*. *eLife*, 6, e28298. https://doi.org/10.7554/*eLife*.28298

Ellner, S. P., Buchon, N., D¨orr, T., and Lazzaro, B. P. (2021). Host-pathogen immune feedbacks can explain widely divergent outcomes from similar infections. Proceedings of the Royal Society B, 288(1951), 20210786. https://doi.org/10. 1098/rspb.2021.0786

Faucher, C., Mazana, V., Kardacz, M., Parthuisot, N., Ferdy, J.-B., and Duneau, D. (2021). Step-specific adaptation and trade-off over the course of an infection by gasp mutation small colony variants (P. Keim, Ed.). mBio, 12(1). https://doi.org/10.1128/mBio.01399-20

Gilchrist, M. A., and Coombs, D. (2006). Evolution of virulence: Interdependence, constraints, and selection using nested models. Theoretical population biology, 69(2), 145–153.

Hanson, M., Dost´alov´a, A., Ceroni, C., Poidevin, M., Kondos, S., and Lemaitre, B. (2019). Synergy and remarkable specificity of antimicrobial peptides in vivo using a systematic knockout approach. *eLife*, 8, e44341. https://doi.org/10.

7554/*eLife*.44341

Lin, Fulzele, A., Cohen, L. B., Bennett, E. J., and Wasserman, S. A. (2020). Bombardier enables delivery of short-form bomanins in the *Drosophila* toll response. Frontiers in Immunology, 10, 3040. https://doi.org/10.3389/fimmu.

2019.03040

Souto-Maior, C., Sylvestre, G., Dias, F. B. S., Gomes, M. G. M., and Maciel-de-Freitas, R. (2018). Model-based inference from multiple dose, time course data reveals Wolbachia effects on infection profiles of type 1 dengue virus in Aedes aegypti. PLoS Neglected Tropical Diseases, 12(3), e0006339. https://doi.org/10.1371/journal.pntd.0006339

van Leeuwen, A., Budischak, S. A., Graham, A. L., and Cressler, C. E. (2019). Parasite resource manipulation drives bimodal variation in infection duration. Proc. R. Soc. B, 286, 20190456. https://doi.org/https://doi.org/10.1098/rspb.

2019.0456

Yu, G., Hu, Y., Wang, S., Han, X., Du, X., Xu, H., Zeng, X., Steiner, U., and Rolff, J. (2021). Bistable host-pathogen interaction explains varied infection outcomes. bioRxiv. https://doi.org/10.1101/2021.04.13.439629

Zhang, Z. (2016). Mathematical model of a bacteria-immunity system with the influence of quorum sensing signal molecule. Journal of Applied Mathematics and Physics, 4(5).

[Editors’ note: what follows is the authors’ response to the second round of review.]

The manuscript has been improved, but there are some remaining issues that need to be addressed, as outlined below:Reviewer #1 (Recommendations for the authors):1) Experimental data setI fully disagree with response 3. My concerns of Bomanins affecting tolerance have now been validated by further studies since this manuscript was first submitted (Xu et al. 2023 EMBO Rep). This supports signals already seen in Lin et al. (the authors should review Lin et al. Fig5bbd-early vs late and Fig6). In the new experiments, the authors use of Drs is inappropriate. Drs is involved in hemocyte recruitment to cancerous tissue and Drs OE suppresses JNK activation (Krautz et al. 2020 eLife). Drs is further implicated in other anti-cancer or traumatic brain injury responses (papers by Inoue lab 2019 and multiple recently by Wassarman lab). Both Bom and Drs can reasonably affect tolerance, and any mutation might have unintended consequences. A wild-type control is essential, and there is no justifiable reason not to include one.

We can understand that our approach may seem unusual. Ideally, we would like to have a control with a genetic background that perfectly matches that of our mutant. However, there is generally no such perfect control. At the time this experiment was done, the iso *w^1118^* background control had a problem of high and unusual mortality compared to all of our lines, the cause of which was unknown (since then, it has been communicated that the iso *w^1118^* background had indeed a viral chronic infection). Therefore, we considered the best alternative and thought that it might be even better to use a mutant line created or backcrossed in the same background, but with a mutation known to have no phenotype or a much smaller phenotype than the focal mutant. The mutants we decided to use as controls are defective for 10 of the 12 bomanins (*Bom*^∆55*C*^) and for Drosomycin (*Drs*), two mutants which compared to Diptericins have much less effect on resistance to *P. rettgeri* infections. But reviewer #1 has pointed out nice studies showing that both of these mutated genes could have some impact on disease tolerance. We nevertheless think that this does not invalidate our approach. First, concerning Bomanins, the most recent study investigate tolerance to fungi toxins, not bacteria. It therefore does not translate easily to our experiments. In addition, the reviewer reports a lower PLUD for the bbd mutant, while we have used *Bom*^∆55*C*^. In this study, the latter had no phenotype in the time frame corresponding to our study. Second, regarding Drs, the studies listed by reviewer #1 show that Drs does more than being involved in resistance. But, to our knowledge, they do not suggest anything related to tolerance to bacterial infections. As much as we appreciate and understand the comment of the reviewer, and we can’t exclude that Drs has some role in disease tolerance, we don’t think that our reasoning was as wrong as they seem to suggest. We have challenged our hypothesis in three different ways, including a RNAi experiment, and even if there are uncertainties, they all lead to the same conclusions. We think however that being clear on the limits of our experiments is important, so we now mention it in the manuscript (lines 586-590)

The RNAi experiments are appreciated, and useful. However they raise some concerns as somehow the PLUD of Cat-IR is 10^2 higher than of Dpt-IR. PLUD is not a metric that should be so sensitive to inter-experiment variation, so these data are difficult to reconcile, and their meaning is difficult to trust. Also, what is the control? No detail is given for "mock-RNAi", and in general RNAi is best used as supporting evidence due to the need to mix genetic backgrounds that could affect results in cryptic ways.

The second part of this paragraph points out a ”mistake” that may explain the skepticism of reviewer 1: we forgot to give the information in the supplementary methods about the new lines and new crosses that we have used. We thank greatly the reviewer for pointing this out. Instead of briefly mentioning this in the figure, we now add more details on these crosses in the supplementary method section (S7.1). For each RNAi experiment, we used attP2 (for DiptA) and attP40 (for Catalase) match background control lines. These are the recommended RNAi background match control of the TRiP genetic RNAi panel, from which the RNAi of Diptericin and Catalase are chosen.

The reviewer #1 also pointed out that these two control lines reach very different PLUD values. This is not entirely surprising, because they are from different backgrounds and we know that attP2 and attP40 have very different phenotypes regarding infection (in part because attP2 is inserted in non coding sequences while attP40 is in *msp300*, see PLoS One, 2022; 17(12):e0278598, and unpublished data from Buchon lab). We also realized that we have used misleading language. We should not have said that the control lines were ”mock-RNAi”. The two control lines indeed have the same docking insertions than the lines they were compared to, but they do not express any RNAi. We corrected this in the text, and now mention that they are the background match controls (lines 590-594; 616-618; S7.1; captions of Figure 8 and Figure 9).

2) Regarding my previous point, apologies if this was not clear, but reflecting 2 years later perhaps I can frame this concern better: lower resistance increasing PLUD is misleading phrasing. The way Figure 5 is presented reflects theoretical space. But PLUD is something defined by a biological limit of the host carrying capacity with a physical volume restriction as a theoretical maximum. As shown previously by Duneau et al. 2017, max PLUD of Dmel individuals in Figure 2 of Duneau was log2(25) to log2(26) across all *D. melanogaster* studied. What is different across strains here (and in Duneau et al. 2017 to some extent) is variance of PLUD. Thus why it's odd to frame it as "PLUD increases by loss of resistance," because in fact the maximum PLUD in Fig7 is pretty consistent across all genotypes and treatments, and no increase is really possible (physical/biological limits). Instead, loss of resistance leads to more consistent microbial growth, faster, and reflected by more consistent mortality outcomes. Thus you get more consistent PLUDS near the maximum physical PLUD. In a wild-type host, resistance creates more complex dynamics, and opens the door for tolerance to impact the outcome and the PLUD. The response supplementary data support this concern exactly. Here, seemingly in 3of4 cases, the PLUD data have an intrinsic survivor bias: blue data points with bimodal survival outcomes have a right skew, while red data points lacking diverse survival outcomes skew to the left, and may even show less diverse ranges (certainly true of 630 and 559).

There are several things here that we will try to unfold. First of all, in our theoretical framework the PLUD cannot really be defined as a biological or physical limit. In fact, as shown by the dashed line in Figure 3D-F, the time to death and the corresponding PLUD are established by how fast the infection causes the maximum level of damage the host can endure. Consequently, the bacterial load continues changing after the host’s death, and in most, if not all cases, it eventually exceeds the PLUD. Beyond theory, an experimental confirmation of this can be found in Faucher et al. mBio 2021 where we studied the proliferation of *Xenorhabdus nematophila* during infection and after death. We have now added a sentence making this explicit in the legend of figure 3. That said, physical constraints – such as the amount of resources a pathogen can extract from its host – should influence the pathogen’s proliferation rate, which in turn should determine the PLUD. However, our simulations reveal that the PLUD is influenced by more subtle factors. For instance, we found that a defect in resistance could result in a higher PLUD because the bacteria proliferate so quickly that the damage they cause ”lags” behind. In fact, most of the tolerance and resistance effects we predict in our theoretical study arise from similar time-lag phenomena. In summary, we argue that robust predictions cannot be achieved through verbal models alone, as lagging and nonlinear infection dynamics are too complex. Indeed, it was precisely our inability to reconcile the diverse experimental results from Duneau et al. (2017) with our own verbal models that led us to develop a mathematical model. Reviewer #1 also suggests that a defect in resistance should influence the variance in PLUD. While we have not specifically investigated this, we can address it through our sensitivity analysis. In particular, Figure S4-1 shows how sensitive the PLUD is to the initial infection conditions. We expect that high sensitivity to initial conditions – such as injected dose or host condition, which cannot be fully controlled in experiments – would lead to high variance in PLUD. Figure S4-1 demonstrates that low resistance (i.e., low values of *α* and/or *γ*) generally results in high sensitivity, likely translating to high variance in PLUD. This prediction is directly opposed to that of reviewer #1. We now develop this argument at the end of the section S4.2. Finally, reviewer #1 is correct in noting that in Duneau et al., *eLife* 2017, the PLUD is often around 2^25^ or 2^26^. However, Figure 2A of that study, which combines data across the entire study, shows that the PLUD can vary up to 2^28^. Note that 2^28^ is equivalent to 10^8.4^, which is much higher than the PLUD observed in our current study. Interestingly, the RNAi experiment for DiptA does have a lower PLUD than the RNAi experiment for Catalase, with the latter displaying a high PLUD similar to that in Duneau et al., *eLife* 2017. As we mentioned, this may be due to differences in genotypes. But it may reflect that Duneau et al., *eLife* 2017, and the Catalase RNAi experiment were both conducted at Cornell University, while the other experiments in our current study were carried out at Toulouse University. Overall, this supports the reproducibility of our results, especially when experiments are repeated in the same location years later.

3) The theory of the model, to this reviewer, seems exceptionally detailed, consistent, and logical. This study has merits and contributes to a body of literature that is seeking to formalize host-pathogen interactions in a mathematical biology framework. I remain concerned with the application of this model to empirical data, which appears to be complex to interpret.

We are pleased that reviewer #1, whom we dare guess is a Drosophilist, found the theoretical part of our study clear and important for the field. We hope that our response and the clarifications addressing some of their misunderstandings will now convince them that our study clarifies infection parameters which were previously defined only by verbal models and affected by misconceptions. We also hope that they will agree that developing these methods will allow the community to test fundamental hypotheses relevant in other animal models, which are sometimes experimentally inaccessible.

Reviewer #2 (Recommendations for the authors):General appreciation: The authors put a lot of effort in responding to all the reviewer comments, which I think greatly improved the manuscript.1) There is from my point of view only one issue remaining, which has not been sufficiently addressed. I apologize in case my previous comment on that matter was not clear enough. I had remarked that potential variation in host background mortality should be controlled for in empirical analyses. The authors addressed this issue in the context of their PLUD analyses by removing outliers. However, variation in host background mortality could also be a serious issue for the survival analyses. It seems that currently the implicit assumption in the conducted survival analyses is that there is no variation in background mortality among the compared strains or treatments. Thus, any inferred survival difference is attributed to different infection dynamics. If the possibility of different background mortalities is considered, then a correct interpretation of the survival analyses seems to require the analysis of appropriate non-infection controls. In the simplest case, it might be sufficient to show that there are no apparent survival differences among strains or treatments in non-infection controls. If there are any differences, they would need to be somehow controlled for in the survival analyses of the infected individuals. In case the authors disagree with my argumentation, it would be useful to provide a corresponding explanation in the manuscript why it is not necessary to include non-infection controls in the conducted survival analyses.

This is a valid point, it is particularly important when we are studying survival over a long period of time. However, in our infection experiments, most deaths occur within the first and second days, and young flies like ours do not die within this time frame if they are not infected. We generally inject a small number of flies (10) with PBS (as mentioned in line 1443), especially when survival are kept for a longer period of time, just to make sure that nothing major and unexpected happened. This data are enough to safely say that background mortality is essentially zero during the two days of our experiment, but they are too scarce to be included in our main analyses. It is very likely that readers may have the same concern than reviewer #2 and missed what we wrote, so we now write it more explicitly (line 1441).

Reviewer #3 (Recommendations for the authors):General appreciation: The authors have addressed my comments thoroughly, and I thank them for this. The manuscript is improved as a result. I still have two points about the model.1) I still find the use of the term "stochastic simulations" misleading in the legend of Figure 3. I recommend that the authors explicitly specify "Dots corresponds to results of stochastic simulations where the initial pathogen load is randomly drawn (…) and then the deterministic equations of the model are solved numerically."

We agree that specifying that the deterministic equations of the model are solved numerically after initial conditions are randomly drawn makes the description of our simulations much clearer. This addition has now been included in the caption of Figure 3.

2) I got a bit worried by the authors' response regarding parameter values and robustness to varying them. Indeed they state: "As often with models, duration can be changed almost at will by adjusting parameters or initial conditions!" One would hope that if the parameters are varied in a physiological range, the conclusions do not vary "at will"… This said, I understand the difficulty of precisely determining each parameter.

We understand that our statement suggesting conclusions can vary ”at will” might sound concerning! What we meant to convey is that our theoretical work was not intended to provide quantitatively precise predictions, but rather qualitative insights that we can test. Naturally, we would have preferred to provide quantitative predictions, and this will likely be our next step. Achieving this requires fitting the model to experimental data, which in turn necessitates the development of ad hoc statistical methods. We believe this is a topic deserving of a separate manuscript.

Overall, I find that the theory-experiment comparison is an important strength of this manuscript.